# Learning Efficient Positional Encodings with Graph Neural Networks

**Charilaos I. Kanatsoulis[1]\*, Evelyn Choi[1], Stephanie Jegelka[2,3], Jure Leskovec[1]†,
Alejandro Ribeiro[4]†**
[1] Stanford University   [2] MIT   [3] Technical University of Munich   [4] University of Pennsylvania

## Abstract

Positional encodings (PEs) are essential for effective graph representation learning because they provide position awareness in inherently position-agnostic transformer architectures and increase the expressive capacity of Graph Neural Networks (GNNs). However, designing powerful and efficient PEs for graphs poses significant challenges due to the absence of canonical node ordering and the scale of the graph. In this work, we identify four key properties that graph PEs should satisfy: stability, expressive power, scalability, and genericness. We find that existing eigenvector-based PE methods often fall short of jointly satisfying these criteria. To address this gap, we introduce PEARL, a novel framework of learnable PEs for graphs. Our primary insight is that message-passing GNNs function as nonlinear mappings of eigenvectors, enabling the design of GNN architectures for generating powerful and efficient PEs. A crucial challenge lies in initializing node features in a manner that is both expressive and permutation equivariant. We tackle this by initializing GNNs with random node inputs or standard basis vectors, thereby unlocking the expressive power of message-passing operations, while employing statistical pooling functions to maintain permutation equivariance. Our analysis demonstrates that PEARL approximates equivariant functions of eigenvectors with linear complexity, while rigorously establishing its stability and high expressive power. Experimental evaluations show that PEARL outperforms lightweight versions of eigenvector-based PEs and achieves comparable performance to full eigenvector-based PEs, but with one or two orders of magnitude lower complexity. Our code is available at `https://github.com/ehejin/Pearl-PE`.

## 1 Introduction

Positional encodings (PEs) are fundamental components of graph representation learning and play a key role in the design of effective Graph Transformers (Dwivedi & Bresson, 2021; Rampášek et al., 2022) and Graph Neural Networks (GNNs) (Kipf & Welling, 2016; Hu et al., 2020). Transformer architectures (Vaswani, 2017) are inherently agnostic to structure and node identities, and PEs provide a powerful mechanism to incorporate positional and structural information. On the other hand, message-passing GNNs often struggle with low expressiveness, especially when node features exhibit the same symmetries as the graph structure (Xu et al., 2019; Morris et al., 2019; Kanatsoulis & Ribeiro, 2024). By integrating structural and positional information, PEs enhance GNNs' capacity to capture patterns that would otherwise be difficult to learn and generalize.

Several graph PE methods have been proposed in the literature, which can broadly be categorized into two main types: absolute PEs and relative PEs. Absolute PEs assign an embedding to each node in the graph, reflecting the node's role within the graph structure. Common approaches include Laplacian eigenvectors (Dwivedi & Bresson, 2021), substructure encodings (Tahmasebi et al., 2020; You et al., 2021; Bouritsas et al., 2022), random walk (RW) encodings (Rampášek et al., 2022), and eigenvector-based methods (Kreuzer et al., 2021a; Lim et al.; Huang et al.). Relative PEs, on the other hand, assign representations to pairs of nodes and typically utilize measures such as shortest-path and resistance distances (Ying et al., 2021; Zhang et al., 2023), as well as RW matrices (Ma et al.,

---

\*Correspondance to: charilaos@cs.stanford.edu. †Equal senior authorship.

2023; Geisler et al., 2023). A thorough comparison between absolute and relative PEs can be found in (Black et al., 2024).

In this paper, we study absolute PEs for graphs based on four key criteria: expressive power, scalability, stability under perturbations, and generality. We find that PEs based on eigenvectors of graph Laplacian or other graph operators often struggle to satisfy all these criteria simultaneously. To better understand this, we divide eigenvector-based approaches into two categories: those that compute the full set of eigenvectors and those that only consider the $K$ largest. Full eigenvector approaches offer high expressive power but come with a computational complexity of $\mathcal{O}(N^3)$ and memory complexity of $\mathcal{O}(N^2)$, which is prohibitive for even medium-sized graphs. The full set of eigenvectors can also be used to learn spectral graph filters (Huang et al.), which result in stable PEs. Note that stability is particularly crucial for out-of-distribution generalization.

However, when only a subset of eigenvectors is computed, several limitations arise. First, this introduces an inductive bias, as different graphs encode different information across eigenvalues, especially when they differ in size. Second, the expressive power and stability are reduced, becoming dependent on the eigengap between the selected eigenvalues. These methods also face challenges in terms of stability and generalization when applied to different or unseen graph structures. Consequently, this approach often leads to significantly poorer performance. Substructure-based encodings face similar challenges: while generally stable, they also introduce inductive bias and highly expressive versions require combinatorial complexity. The aforementioned challenges raise a critical research question:

**Question:** Can we learn generic PEs that are simultaneously expressive, stable, and scalable?

In this work, we provide an affirmative answer by proposing PEARL, a powerful and efficient framework for learnable PEs, entirely generated via message-passing GNNs. We begin by showing that message-passing GNNs can be understood as nonlinear mappings of eigenvectors of the graph Laplacian or other graph shift operators. This insight enables the computation of eigenvector-based PEs efficiently with linear or quadratic complexity, leveraging message-passing operations. A central challenge in developing effective PEs with GNNs lies in initializing node features to ensure both expressiveness and permutation equivariance. We address this by initializing each node with a set of $M$ random samples, effectively breaking the symmetries between the graph structure and node features. Each sample is processed independently by a GNN, and the output samples are aggregated via statistically-designed pooling functions to guarantee permutation equivariance. Our analysis demonstrates that PEARL is basis universal, surpasses the expressiveness of the Weisfeiler-Leman (WL) test (Weisfeiler & Leman, 1968), and generalizes the ability of counting key graph substructures. PEARL is also provably stable, inheriting the stability guarantees of GNNs (Gama et al., 2020), which are independent of the eigenvalue gap.

We further analyze the sample complexity of PEARL and show that the number of samples required for effective encoding is independent of graph size. This enables the generation of powerful eigenvector-based PEs for large graphs with linear complexity. For smaller graphs, where the number of samples is comparable to the graph size, we propose an alternative model that initializes node features with basis vectors. This approach approximates the PEs in (Huang et al.) with significantly lower computational and memory complexity. We evaluate the proposed PEARL on graph classification and regression tasks on molecular graphs and social network datasets, and compare it against eigen-based and structure-based absolute PEs. The results demonstrate that PEARL consistently outperforms structure-based PEs and lightweight variants of eigenvector-based PEs, achieving significant improvement on graph classification tasks. In comparison to full eigenvector-based PEs, which have a computational complexity of $\mathcal{O}(N^3)$, PEARL delivers comparable performance with significantly reduced complexity, scaling at $\mathcal{O}(N)$ or $\mathcal{O}(N^2)$.

## 2 PRELIMINARIES

A graph $\mathcal{G} := (\mathcal{V}, \mathcal{E})$, is represented by a set of vertices $\mathcal{V} = \{1, \ldots, N\}$, a set of edges $\mathcal{E} = \{(v, u)\}$, and a graph shift operator (GSO) $\boldsymbol{S} \in \mathbb{R}^{N \times N}$. The GSO is typically sparse, with common choices including the adjacency matrix, the Laplacian matrix, their normalized variants, or the RW transition matrix. The nodes (vertices) in the graph are often associated with node signals $\boldsymbol{x}_v \in \mathbb{R}^d$, each with $d$ features, while edges can carry edge attributes $\boldsymbol{x}_{(u,v)} \in \mathbb{R}^{d_e}$ with $d_e$ features. An important operation in graphs is the spectral decomposition of the graph and refers to the eigenvalue decomposition to the

GSO, $\boldsymbol{S} = \boldsymbol{V}\boldsymbol{\Lambda}\boldsymbol{V}^T$. Matrix $\boldsymbol{V} = [\boldsymbol{v}_1, \ldots, \boldsymbol{v}_n]$ is the orthonormal matrix of eigenvectors, and $\boldsymbol{\Lambda}$ is the diagonal matrix of eigenvalues $\{\lambda_n\}_{n=1}^N$. When $\boldsymbol{S}$ represents the Laplacian matrix, $\boldsymbol{V}$ are the Laplacian eigenvectors that are commonly used as node features or PEs for GNN architectures. In this paper, we study standard message-passing GNNs, defined by the following recursive formula:

$$\boldsymbol{x}_v^{(l)} = g^{(l-1)}\left(\boldsymbol{x}_v^{(l-1)}, f^{(l-1)}\left(\left\{\boldsymbol{x}_u^{(l-1)} : u \in \mathcal{N}(v)\right\}\right)\right). \tag{1}$$

Here, $\mathcal{N}(v)$ represents the 1-hop neighborhood of vertex $v$. The function $f^{(l)}$ aggregates information from the multiset of signals coming from neighboring vertices, while $g^{(l)}$ combines the signal of each vertex with the aggregated information from its neighbors. Common choices for $f^{(l)}$ and $g^{(l)}$ include the single- and multi-layer perceptron (MLP), the linear function, and the summation function.

## 3 OUR WORK: LEARNABLE, EFFICIENT, AND POWERFUL PEs WITH GNNs

### 3.1 GNNs ARE NONLINEAR FUNCTIONS OF GSO EIGENVECTORS

Our first observation is that message-passing GNNs are nonlinear functions of eigenvectors. To see this, let $f^{(l)}$ be one of the following aggregation functions:

$$\sum_{u \in \mathcal{N}(v)} \boldsymbol{x}_u, \quad d_v \cdot \boldsymbol{x}_v - \sum_{u \in \mathcal{N}(v)} \boldsymbol{x}_u, \quad \sum_{u \in \mathcal{N}(v)} \frac{\boldsymbol{x}_u}{\sqrt{d_v d_u}}, \quad \boldsymbol{x}_u - \sum_{u \in \mathcal{N}(v)} \frac{\boldsymbol{x}_u}{\sqrt{d_v d_u}}, \quad \sum_{u \in \mathcal{N}(v)} \frac{\boldsymbol{x}_u}{d_u} \tag{2}$$

where $d_v$ is the degree of node $v$. Then Eq. (1) can be written as $\boldsymbol{X}^{(l)} = g^{(l-1)}\left(\boldsymbol{X}^{(l-1)}, \boldsymbol{S}\boldsymbol{X}^{(l-1)}\right)$, where $\boldsymbol{S}$ represents the adjacency matrix, the Laplacian matrix, the normalized adjacency, the normalized Laplacian, and the RW matrix, for the five choices of $f^{(l)}$ in 2 respectively, and $\boldsymbol{X}^{(l)} \in \mathbb{R}^{N \times F_l}$ represents the signals of all vertices at layer $l$. Now let $g^{(l)}$ be an equivariant MLP operating on each node independently. Note that the MLP is a common choice for function $g^{(l)}$ for the majority of effective GNN architectures due to its expressiveness properties. Then Eq. (1) can be cast as:

$$\boldsymbol{X}^{(l)} = \sigma\left(\sum_{k=0}^{K-1} \boldsymbol{S}^k \boldsymbol{X}^{(l-1)} \boldsymbol{H}_k^{(l)}\right), \tag{3}$$

where $K = 2$, $\boldsymbol{H}_k \in \mathbb{R}^{F_{l-1} \times F_l}$ are the trainable parameters, and $\sigma$ is a point-wise nonlinear activation function. Note that Eq. (3) defines a single-layer graph perceptron, but it can be easily generalized to a multi-layer graph perceptron by letting $\sigma$ represent an equivariant MLP acting on the node signals. Additionally, while we set $K = 2$ here, higher values of $K$ can be considered for more generalized GNN layers. It is worth emphasizing that $\boldsymbol{S}^k$ is never explicitly instantiated; instead, $\boldsymbol{S}^k \boldsymbol{X}^{(l-1)}$ is computed using recursive message-passing operations, as outlined in Eq. (2).

**Proposition 3.1 (GNNs are nonlinear functions of eigenvectors)** *A GNN defined in Eq. (1) with $f^{(l)}$ being one of the functions in Eq. (2) and $g^{(l)}$ being a multi-layer perceptron, operates as a nonlinear function of the GSO eigenvectors, i.e., $\boldsymbol{x}_v^{(l)} = \mathtt{MLP}\left(\boldsymbol{v}^{(v)}\right)$, $\boldsymbol{v}^{(v)} = \boldsymbol{V}[v,:]^T$. The trainable parameters of the first $\mathtt{MLP}$ layer are not independent but depend on the eigenvalues $\{\lambda_n\}_{n=1}^N$ and eigenvectors $\{\boldsymbol{v}_n\}_{n=1}^N$ of the GSO, as well as the node features $\boldsymbol{X}$ of the graph:*

$$\boldsymbol{x}_v^{(l)} = \mathtt{MLP}\left(\boldsymbol{v}^{(v)}\right) = \mathtt{MLP}^{(-1)}\left(\sigma\left(\boldsymbol{W}^T \boldsymbol{v}^{(v)}\right)\right) \tag{4}$$

$$\boldsymbol{W}[n, f] = \sum_{i=1}^{F_{l-1}} \sum_{k=0}^{K-1} \lambda_n^k \boldsymbol{H}_k^{(l)}[i, f] \langle \boldsymbol{\alpha}_n, \boldsymbol{X}^{(l-1)}[:, i] \rangle, \tag{5}$$

*where $\boldsymbol{\alpha}_n = \boldsymbol{v}_n$ when the GSO is symmetric and $\boldsymbol{\alpha}_n = \boldsymbol{V}^{-1}[n,:]$ when it is not. $\mathtt{MLP}^{(-1)}$ denotes all the layers of the $\mathtt{MLP}$ except the first layer.*

The proof is provided in Appendix B. According to Proposition 3.1, the update for node $v$, defined by the function $g \circ f : \left(\mathcal{G}, \mathbb{R}^{F_{l-1}}\right) \to \mathbb{R}^{F_l}$, can be interpreted as a nonlinear mapping ($\mathtt{MLP}$) applied to $\boldsymbol{V}[v,:]$, but the weights of the first layer of this mapping are also mappings, i.e., $\mathbf{W}[n, f]$ :

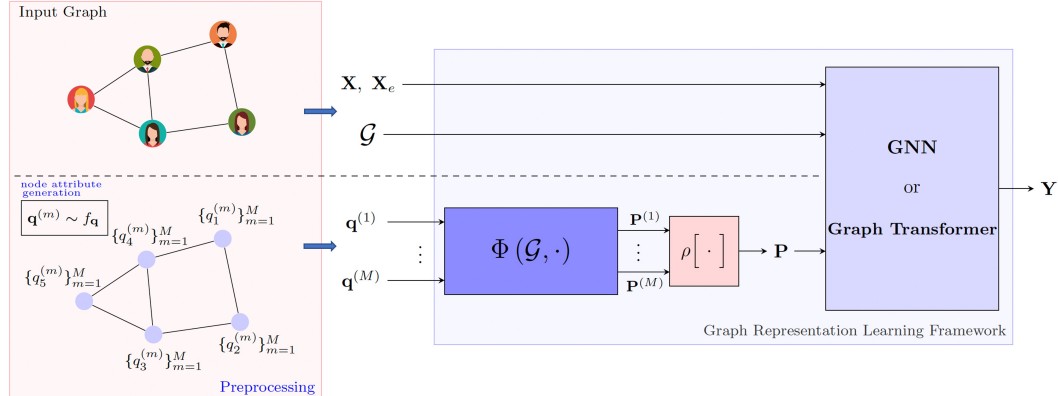

Figure 1: PEARL framework: The input graph undergoes anonymization by removing its node and edge attributes. For each node, a set of $M$ random or basis attributes is generated. Each sample is then independently processed by a message-passing GNN, and a pooling function $\rho$ is applied to produce equivariant PEs. The graph structure, together with the generated PEs and any node or graph attributes, is subsequently processed using either a GNN or a Graph Transformer.

$\left(\mathcal{G}, \mathbb{R}^{F_{l-1}}\right) \to \mathbb{R}$. The degrees of freedom in the first layer of MLP are $KF_lF_{l-1}$ (as described in Eq. (5)), rather than $F_lN$, which would be the case for independent weights $\boldsymbol{W}$. Furthermore, the dot product $\langle \boldsymbol{\alpha}_n, \boldsymbol{X}^{(l-1)}[:,i]\rangle$ depends on the eigenvectors and, for the update of node $v$, it only involves the components $\boldsymbol{X}^{(l-1)}[u,i]$, $u \in \mathcal{N}_v$. Proposition 3.1 is applicable to most message-passing GNN models, including, but not limited to, Graph Convolutional Networks (GCNs) (Kipf & Welling, 2016), Graph Isomorphism Networks (GINs) (Xu et al., 2019), and GraphSAGE (Hamilton et al., 2017).

## 3.2 PEARL: EXPRESSIVE AND EQUIVARIANT POSITIONAL ENCODING NETWORKS

Following the derivation of Proposition 3.1, a critical question arises: what is the optimal choice of node features that allow a GNN to compute expressive and equivariant functions of the eigenvectors? Equivariant structural features augment GNNs with valuable information, but they come at the cost of increased computational complexity and inductive bias. Moreover, these features share the same symmetries as the graph structure, limiting the expressiveness of message-passing GNNs. Alternatively, unique identifiers, such as random node features, can break the structural symmetries and improve expressiveness but at the expense of permutation equivariance, which limits the model's generalization capability. To address this trade-off between equivariance and expressiveness, we propose to momentarily break the structural symmetries by initializing each node with a set of $M$ unique identifiers, while maintaining permutation equivariance in the model output via the use of statistical pooling functions. The proposed PE framework (PEARL) is illustrated in Fig. 1 and, as we see next, ensures both high expressiveness and strong generalization.

Consider a graph $\mathcal{G} = (\mathcal{V}, \mathcal{E})$ with $N$ nodes. For each node $v \in \mathcal{V}$ in $\mathcal{G}$, we design a set of $M$ $1-$dimensional node signals $\left\{q_v^{(1)}, q_v^{(2)}, \ldots, q_v^{(M)}\right\}$, where each $q_v^{(m)}$ operates as a unique identifier. Graph $\mathcal{G}$ is now associated with a set of $M$ independent initial node features represented as $\left\{\boldsymbol{q}^{(m)}\right\}_{m=1}^{M}, \boldsymbol{q}^{(m)} \in \mathbb{R}^N$. Each pair of $\left\{\mathcal{G}, \boldsymbol{q}^{(m)}\right\}$ is independently processed via a GNN $\Phi(\cdot)$, which is described by Eq. (1) or (3), to produce a set of $M$ independent outputs:

$$\boldsymbol{P}^{(m)} = \Phi\left(\mathcal{G}, \boldsymbol{q}^{(m)}\right) \in \mathbb{R}^{N \times d_P}, \quad m = 1, \ldots, M \tag{6}$$

Since $\left\{\boldsymbol{q}^{(m)}\right\}_{m=1}^{M}$ operate as unique identifiers, they break the structural symmetries and unlock the expressive power of message-passing operations. However, each $\boldsymbol{P}^{(m)}$ is not permutation equivariant, and thus is not generalizable. To address this, we design an equivariant function $\rho$, that involves a pooling operation over the independent outputs $\{\boldsymbol{P}^{(m)}\}_{m=1}^{M}$, to generate the final PE for each node:

$$\boldsymbol{P} = \rho\left[\Phi\left(\mathcal{G}, \boldsymbol{q}^{(1)}\right), \ldots, \Phi\left(\mathcal{G}, \boldsymbol{q}^{(M)}\right)\right] \in \mathbb{R}^{N \times d_P} \tag{7}$$

The PEARL framework can universally approximate any continuous basis invariant function.

**Theorem 3.1 (Basis Universality)** *Let $\mathcal{G}$ be a graph with GSO $\boldsymbol{S} = \boldsymbol{V}\boldsymbol{\Lambda}\boldsymbol{V}^T$, and $f$ be a continuous function such that $f(\boldsymbol{V}) = f(\boldsymbol{V}\boldsymbol{Q})$, $\boldsymbol{Q} \in \mathcal{O}\left(diag\left(\boldsymbol{\Lambda}\right)\right)$, for any eigenvalues $\boldsymbol{\Lambda}$. Then there exist GNN $\Phi$ and a continuous equivariant function $\rho$, such that $f(\boldsymbol{V}) = \rho\left[\Phi\left(\mathcal{G}, \boldsymbol{q}^{(1)}\right), \ldots, \Phi\left(\mathcal{G}, \boldsymbol{q}^{(M)}\right)\right]$.*

The proof can be found in Appendix C. Theorem 3.1 can be extended to handle multiple graphs by considering $\mathcal{G}$ as a collection of graphs represented as disconnected components. In this case, the GSO takes a block diagonal form, where each block corresponds to the GSO of an individual graph.

In the following sections, we explore options for the initial node features $\left\{\boldsymbol{q}^{(m)}\right\}_{m=1}^M$ and pooling functions $\rho$. A key aspect of PEARL is designing $M$ independent initial attributes for each node, which enables permutation equivariance at the model's output through the use of pooling functions. This stands in contrast to classical methods, which typically assign a single unique identifier per node.

# 4 OUR WORK: RANDOM POSITIONAL ENCODING NETWORK (R-PEARL)

Next, we present our Random PE Network (R-PEARL). In R-PEARL we define inital node features $\left\{\boldsymbol{q}^{(m)}\right\}_{m=1}^M$ by sampling them randomly from a probability distribution. Specifically, let $\boldsymbol{q} = [q_{v_1}, q_{v_2}, \ldots, q_{v_N}]^T$, where $v_n \in \mathcal{V}$, be a random vector with joint distribution $f_{\boldsymbol{q}}(t_1, \ldots, t_N)$. The set $\left\{\boldsymbol{q}^{(m)}\right\}_{m=1}^M$ consists of $M$ independent $N$-dimensional realizations of $\boldsymbol{q}$, drawn from $f_{\boldsymbol{q}}$. In our experiments and analysis, $\boldsymbol{q}$ is either a set of independent and identically distributed (i.i.d.) Gaussian random variables or a set of i.i.d. random variables with $\mathbb{E}[q_i] = 0$ and $\mathbb{E}[q_i^p] = 1$, where $p \geq 2$.

When these samples are processed by a GNN $\Phi$, the result is $M$ $(N \times d_p)$-dimensional samples of the random matrix output $\Phi(\mathcal{G}, \boldsymbol{q})$. To practically preserve permutation equivariance, we note that the distribution of $\Phi(\mathcal{G}, \boldsymbol{q})$ is itself permutation equivariant, as are any statistics derived from it. Therefore, the function $\rho$ can be any empirical statistic computed from the samples $\left\{\Phi(\mathcal{G}, \boldsymbol{q}^{(m)})\right\}_{m=1}^M$, each capturing different characteristics. For instance, $\rho$ could represent any statistical moment, such as the mean or variance, or other measures as the empirical mode or median. In this paper, we choose $\rho$ to be the empirical mean due to its favorable convergence and stability properties, as well as its simplicity in implementation and low computational and memory complexity:

$$\boldsymbol{P} = \hat{\mathbb{E}}\left[\Phi\left(\mathcal{G}, \boldsymbol{q}^{(1)}\right), \ldots, \Phi\left(\mathcal{G}, \boldsymbol{q}^{(M)}\right)\right] = \frac{1}{M}\sum_{m=1}^M \Phi\left(\mathcal{G}, \boldsymbol{q}^{(m)}\right) = \frac{1}{M}\sum_{m=1}^M \boldsymbol{P}^{(m)} \quad (8)$$

## 4.1 SAMPLE COMPLEXITY

In this section, we analyze the number of samples required such that $\frac{1}{M}\sum_{m=1}^M \boldsymbol{P}^{(m)}$ approximates $\mathbb{E}\left[\Phi\left(\mathcal{G}, \boldsymbol{q}\right)\right]$ with negligible error. To that end, we make the following two assumptions.

**Assumption 4.1** *The pointwise nonlinearity $\sigma$ is Lipschitz continuous with Lipschitz constant $C_\sigma$.*

This is a common assumption in deep learning and is satisfied by the widely used nonlinearities. In most cases, such as the Rectified Linear Unit (ReLU), hyperbolic tangent, and sigmoid, it holds that $C_\sigma = 1$. Before introducing the second assumption, we first need to examine Eq. (3) more closely. Notice that its linear component involves $F_L \cdot F_{l-1}$ graph filters of the form $\sum_{k=0}^{K-1} h_k \boldsymbol{S}^k$, which is also explicitly shown in Appendix E, Eq. (53).

**Assumption 4.2** *The linear operators $\boldsymbol{H}\left(\boldsymbol{S}\right) = \sum_{k=0}^{K-1} h_k \boldsymbol{S}^k$ involved in the projection of Eq. (3) are bounded, i.e., $\|\boldsymbol{H}\left(\boldsymbol{S}\right)\| \leq \beta$.*

This is another common assumption in deep learning, where the value of $\beta$ varies depending on the architecture and task. We can now present Theorem 4.3, which characterizes the number of samples $M$ needed for our approach to converge to the true $\mathbb{E}\left[\Phi\left(\mathcal{G}, \boldsymbol{q}\right)\right]$.

**Theorem 4.3 (Sample Complexity)** *Let $\boldsymbol{P}$ denote the output of the architecture described in Eq. (8), for a graph $\mathcal{G}$ with i.i.d. initial node features with unit variance. Also let $\Phi$ be an $L-$layer GNN*

*described by Eq. (3), with F hidden dimensions at each layer. If $C_\sigma = 1$ and $\beta = 1/F$, the number of samples M required such that:*

$$\left| \frac{1}{M} \sum_{m=1}^{M} \Phi\left(\mathcal{G}, \boldsymbol{q}^{(m)}\right) - \mathbb{E}\left[\Phi\left(\mathcal{G}, \boldsymbol{q}\right)\right] \right| < \epsilon, \text{ with probability at least } 1 - \delta, \quad (9)$$

*satisfies:*

$$M \leq \frac{1}{\delta \cdot \epsilon^2}. \quad (10)$$

It is worth noting that the above bound is independent of the size of the graphs, which suggests that our proposed PE framework is well-suited for large-scale graphs. In practice, we have observed that $10 \sim 100$ samples are typically sufficient.

## 4.2 Expressive Power

In this section, we establish the expressive power of our proposed R-PEARL.

**Corollary 4.4 (Structure Counting)** *Let $\boldsymbol{q} = [q_1, \ldots, q_N]$ be a set of N i.i.d. random variables such that $\mathbb{E}[q_i] = 0$ and $\mathbb{E}[q_i^p] = 1$ for $p \geq 2$. Then, there exists a parametrization $\Phi$, defined by Eq. (3), such that $\mathbb{E}\left[\Phi\left(\cdot, \boldsymbol{q}\right)\right]$ counts the number of 3-, 4-, 5-, 6-, and 7-node cycles for any given graph.*

Corollary 4.4 not only highlights the expressive power of the R-PEARL framework but also provides valuable insights into its generalization ability. Essentially, the R-PEARL framework can learn not just the number of cycles each node in a given graph participates in, but also a counting function that generalizes this capability to any node in any graph. The proof can be found in Appendix F, and is based on the results in (Kanatsoulis & Ribeiro). In Corollary 4.5 we characterize the expressive power of message-passing GNNs with our proposed PEs with respect to the folklore-Weisfeiler-Leman (FWL) test (Cai et al., 1992; Morris et al., 2019; Huang & Villar, 2021).

**Corollary 4.5 (Expressive Power)** *A GNN defined in Eq. (1), with PEs produced by Eq. (8) is strictly more powerful than the 1-FWL test, when $f$, $g$ are injective functions.*

The proof of Corollary 4.5 is a consequence of Corollary 4.4 and the analysis in (Xu et al., 2019). Note that the previous results can be improved (e.g., count cycles and cliques of higher order, go beyond 2-FWL test) when the samples $\left\{\boldsymbol{q}^{(m)}\right\}_{m=1}^{M}$ are drawn from a structurally aware distribution, but this will increase the number of computations and is outside of the scope of this paper.

## 4.3 Stability

The proposed PEs are purely generated by GNN architectures and as a result they inherit favorable stability properties of GNNs. Any stability results for GNNs hold for R-PEARL as well. For instance, let $\tilde{\mathcal{G}}$ be a perturbed version of $\mathcal{G}$ such that $\tilde{\boldsymbol{S}} = \boldsymbol{S} + \boldsymbol{E}$. We can use the stability results in (Gama et al., 2020) and derive the following proposition.

**Corollary 4.6 (Stability)** *Let $\tilde{\mathcal{G}}$ be a perturbed version of $\mathcal{G}$ such that $\tilde{\boldsymbol{S}} = \boldsymbol{S} + \boldsymbol{E}$ with $\|\boldsymbol{E}\| \leq \varepsilon$. Let $\Phi$ be an $L-$layer GNN described by Eq. (3), where each layer consists of $F^2$ Lipschitz continuous filters [cf. Eq. (G.2)] with constant C. Under assumptions 4.1 and 4.2 with $C_\sigma = 1$ and $\beta = 1/F$, the following holds:*

$$\left\| \frac{1}{M} \sum_{m=1}^{M} \Phi\left(\mathcal{G}, \cdot\right)[:, f] - \frac{1}{M} \sum_{m=1}^{M} \Phi\left(\tilde{\mathcal{G}}, \cdot\right)[:, f] \right\|_{\mathcal{P}} \leq \left(1 + 8\sqrt{N}\right) L\varepsilon + \mathcal{O}(\varepsilon^2) \quad (11)$$

*where $\|\cdot\|_{\mathcal{P}}$ is the distance modulo permutation [cf. G.1], and M is the number of samples.*

We can further normalize the proposed PEs by $\sqrt{N} \cdot L$ to improve the stability bound. Notably, our result remains independent of the eigengap $\delta_\lambda$, which is the difference between consecutive eigenvalues of the GSO. However, this independence does not hold for the stability of eigenvectors. According to the Davis-Kahan Theorem (Davis & Kahan, 1970), even a small perturbation in the

graph can lead to arbitrarily large differences between the eigenvector encodings of the original and perturbed GSOs. This limitation also applies to the eigenvector-based PEs in (Lim et al.). The stability bound of the PEs in (Huang et al.) is inversely proportional to the eigengap $\delta_\lambda$ between the $d$-th and $(d+1)$-th smallest eigenvalues when using the first $d$ eigenvectors. This dependence is mitigated if all eigenvectors are computed, but doing so requires $\mathcal{O}(N^3)$ complexity, which is impractical for large graphs. Further details on stability results are provided in Appendix G.

## 4.4 COMPUTATIONAL COMPLEXITY

To implement R-PEARL, we process each initial random attribute independently using a message-passing GNN. Consequently, the computational complexity of the feed-forward pass is equivalent to that of a message-passing GNN multiplied by the number of samples, i.e., $\Theta\left(MNF^2 + M|\mathcal{E}|F\right)$, where $F$ represents the hidden dimension of each GNN layer. The memory complexity of a serial implementation is $\Theta\left(NF\right)$, while for a parallel implementation, it is $\Theta\left(MNF\right)$.

## 5 OUR WORK: BASIS POSITIONAL ENCODING NETWORKS (B-PEARL)

The previous approach, R-PEARL, is particularly advantageous for large graphs where the number of samples is much smaller than the number of nodes and edges, resulting in approximately linear computational and memory complexity. However, for smaller-scale graphs, such as molecular graphs, the computational complexity becomes quadratic. In these cases, we propose using standard basis vectors $\{e_m\}_{m=1}^N$ as the initial node features, where $e_m[m] = 1$ and $e_m[i \neq m] = 0$, thus setting $M = N$. Similar to the previous approach, when these samples are processed by a GNN $\Phi$, the result is $N$ $(N \times d_p)$-dimensional outputs. To maintain permutation equivariance, any equivariant function $\rho$ can be applied, but in this paper, we choose the summation pooling for $\rho$. Overall, B-PEARL is cast as:

$$\boldsymbol{P} = \rho\left[\Phi\left(\mathcal{G}, \boldsymbol{e}_1\right), \dots, \Phi\left(\mathcal{G}, \boldsymbol{e}_N\right)\right] = \sum_{m=1}^N \Phi\left(\mathcal{G}, \boldsymbol{e}_m\right) = \sum_{m=1}^N \boldsymbol{P}^{(m)} \tag{12}$$

**Remark 5.1 (Expressivity, Stability)** *B*-PEARL *admits the same expressivity and stability properies as R*-PEARL*, i.e., Corollaries 4.5, and 4.6 also apply, and B*-PEARL *can count the number of 3-, 4-, 5-, 6-, and 7-node cycles, for any given graph.*

## 5.1 RELATION TO EIGENVECTOR BASED ENCODINGS

The B-PEARL framework is highly related to the SPE encodings (Huang et al.), defined as $\text{SPE}\left(\boldsymbol{V}, \boldsymbol{\Lambda}\right) = \rho\left(\left[\boldsymbol{V}\text{diag}\left(\alpha_1\left(\boldsymbol{\Lambda}\right)\right)\boldsymbol{V}^T, \dots, \boldsymbol{V}\text{diag}\left(\alpha_F\left(\boldsymbol{\Lambda}\right)\right)\boldsymbol{V}^T\right]\right)$, where $\{\alpha_i\}_{i=1}^F$ are continuous functions and $\rho$ is an equivariant function. The suggested SPE implementation is $\text{SPE}\left(\boldsymbol{V}, \boldsymbol{\Lambda}\right) = \sum_{n=1}^N \Phi\left(\left[\boldsymbol{V}\text{diag}\left(\alpha_1\left(\boldsymbol{\Lambda}\right)\right)\boldsymbol{V}[n], \dots, \boldsymbol{V}\text{diag}\left(\alpha_M\left(\boldsymbol{\Lambda}\right)\right)\boldsymbol{V}[n]\right]\right)$, where $\Phi$ is a GNN, and $\{\alpha_i\}_{i=1}^F$ are MLPs. The computational and memory complexity of SPE is $\mathcal{O}\left(N^3\right)$, and $\mathcal{O}\left(N^2\right)$ respectively.

**Remark 5.2** *When $\{\alpha_i\}_{i=1}^F$ in SPE $\left(\boldsymbol{V}, \boldsymbol{\Lambda}\right)$ are pointwise analytic functions, the SPE architecture is equivalent to the proposed B*-PEARL *architecture in Eq. (12). The proof can be found in Appendix H.*

## 5.2 COMPUTATIONAL COMPLEXITY

To implement B-PEARL, we process each initial basis encoding independently using a message-passing GNN. As a result, the computational complexity is $\Theta\left(N^2 F^2 + N|\mathcal{E}|F\right)$, where $F$ represents the hidden dimension of each GNN layer. The memory complexity for a serial implementation is $\Theta\left(NF\right)$, while for a parallel implementation, it increases to $\Theta\left(N^2 F\right)$.

## 6 EXPERIMENTS

In this section, we assess the performance of PEARL on graph classification, graph regression and recommendation tasks. All experiments were conducted on a Linux server with NVIDIA A100 GPU.

Table 1: Graph classification accuracy on REDDIT-B and REDDIT-M (OOM stands for out-of-memory). R-PEARL outperforms all baselines in both REDDIT-B and REDDIT-M. R-PEARL has much lower complexity than the well-performing eigen-based methods.

| Method | Computational Complexity | Memory Complexity | REDDIT-B | REDDIT-M |
|---|---|---|---|---|
| GCN | $\mathcal{O}(N)$ | $\mathcal{O}(N)$ | $50.0 \pm 0.0$ | $20.0 \pm 0.0$ |
| GIN | $\mathcal{O}(N)$ | $\mathcal{O}(N)$ | $91.8 \pm 1.0$ | $56.9 \pm 2.0$ |
| GIN + rand id | $\mathcal{O}(N)$ | $\mathcal{O}(N)$ | $91.8 \pm 1.6$ | $57.0 \pm 2.1$ |
| GSN with cliques | $\mathcal{O}(N^2)$ | $\mathcal{O}(N)$ | $91.1 \pm 1.8$ | $56.2 \pm 1.8$ |
| SignNet-8S | $\mathcal{O}(N^3)$ | $\mathcal{O}(N)$ | $92.4 \pm 1.1$ | $57.8 \pm 0.8$ |
| SignNet-8L | $\mathcal{O}(N)$ | $\mathcal{O}(N)$ | $79.5 \pm 12.3$ | $41.4 \pm 2.7$ |
| SignNet-full | $\mathcal{O}(N^3)$ | $\mathcal{O}(N^2)$ | OOM | OOM |
| BasisNet | $\mathcal{O}(N^3)$ | $\mathcal{O}(N^2)$ | OOM | OOM |
| SPE | $\mathcal{O}(N^3)$ | $\mathcal{O}(N^2)$ | OOM | OOM |
| R-PEARL(ours) | $\mathcal{O}(MN)$ | $\mathcal{O}(N)/\mathcal{O}(MN)$ | $\mathbf{93.0 \pm 1.3}$ | $\mathbf{59.4 \pm 1.0}$ |

## 6.1 ARCHITECTURES

To generate the proposed PE, $\Phi$ is an $L-$layer message-passing GNN with batch normalization layers and skip connections, where $L \in \{7, 8, 9\}$. The first layer of $\Phi$ is a generalized GNN layer, as described in Eq. (3), and $K$ can be greater than two. All the remaining layers in $\Phi$ are GIN layers (Xu et al., 2019). When $K = 2$ we omit this generalized GNN layer, and solely use GIN layers. We denote as R-PEARL the architecture with random samples, described in Eq. (8), and B-PEARL the architecture with basis vectors, described in Eq. (12). In all experiments we evaluated our model on selected values of $K$ ranging from 2 to 18, as well as different sample sizes ranging from 10 to 200, and selected the best model accordingly. For the graph-level tasks, the R-PEARL and B-PEARL encodings are fed to a GINE (Hu et al., 2020) architecture, which is a message-passing GNN that processes node and edge attributes, as well as the graph structure and PEs. For the recommendation tasks, the R-PEARL and B-PEARL PEs are further processed via heterogeneous identity-aware GNNs (You et al., 2021).

## 6.2 BASELINES

The baseline models for comparison are grouped into four categories: i) **GNNs without PEs**: GCN (Kipf & Welling, 2016), GIN (Xu et al., 2019); ii) **GNNs with unique identifiers**: GIN with random IDs (Xu et al., 2019; Abboud et al., 2021; Sato et al., 2019); iii) **GNNs with structural PEs**: GSN with cycles, GSN with cliques (Bouritsas et al., 2022); iv) **GNNs with eigenvector-based PEs**: SignNet, BasisNet (Lim et al.), PEG (Wang et al., 2022), SPE (Huang et al.).

In addition, we implement SignNet-8S, BasisNet-8S and SPE-8S which are variants of the full SignNet, BasisNet, and SPE models. These variants employ the eigenvectors corresponding to the 8 smallest eigenvalues of the normalized Laplacian that still need $\mathcal{O}(N^3)$ computational complexity. In SPE-8S and BasisNet-8S the memory complexity remains $\mathcal{O}(N^2)$, but in SignNet-8S it reduces from $\mathcal{O}(N^2)$ to $\mathcal{O}(N)$. Furthermore, we implement SignNet-8L, utilizing only the 8 largest eigenvectors, which reduces both the memory and computational complexity to $\mathcal{O}(N)$.

## 6.3 GRAPH CLASSIFICATION ON SOCIAL NETWORKS

We first evaluate our architecture on graph classification tasks using the REDDIT-B (2,000 graphs, 2 classes, 429.6 average nodes) and REDDIT-M (5,000 graphs, 5 classes, 508.5 average nodes) datasets (Yanardag & Vishwanathan, 2015). Each graph represents an online discussion thread, with nodes representing different users, and edges indicating whether one user responded to another's comment. In both datasets, the task is to predict the subreddit to which a particular discussion graph belongs. To train the GNN models, we conduct 10-fold cross-validation. Table 1 summarizes the mean and standard deviation of classification accuracy over the 10 folds. We report the best performance observed during 350 epochs of training, as is the standard practice for this dataset. The results are presented in Table 1.

We observe that R-PEARL outperforms all baselines on REDDIT-B and REDDIT-M. SignNet-8S achieves the second best performance, but with one to two orders of magnitude higher computational

Table 2: logP Prediction in ZINC. B-PEARL ouperforms all the baselines both in MAE and Generalization gap. It is notable that B-PEARL achieves these results with quadratic complexity compared to the second best (SPE) that operates with cubic complexity.

| PE Method | #PEs | Test MAE | Training MAE | General. Gap |
|---|---|---|---|---|
| No PE | N/A | $0.1772 \pm 0.0040$ | $0.1509 \pm 0.0086$ | $\textbf{0.0263} \pm \textbf{0.0113}$ |
| GCN | N/A | $0.469 \pm 0.002$ | – | – |
| GIN | N/A | $0.209 \pm 0.018$ | – | – |
| GSN with cycles | 10 | $0.115 \pm 0.012$ | – | – |
| GIN + rand id | 1 | $0.279 \pm 0.023$ | – | – |
| SignNet-8S | 8 | $0.1034 \pm 0.0056$ | $0.0418 \pm 0.0101$ | $0.0602 \pm 0.0112$ |
| SignNet | Full | $0.0853 \pm 0.0026$ | $0.0349 \pm 0.0078$ | $0.0502 \pm 0.0103$ |
| BasisNet-8S | 8 | $0.1554 \pm 0.0048$ | $0.0513 \pm 0.0053$ | $0.1042 \pm 0.0063$ |
| BasisNet | Full | $0.1555 \pm 0.0124$ | $0.0684 \pm 0.0202$ | $0.0989 \pm 0.0258$ |
| SPE-8S | 8 | $0.0736 \pm 0.0007$ | $0.0324 \pm 0.0058$ | $0.0413 \pm 0.0057$ |
| SPE | Full | $0.0693 \pm 0.0040$ | $0.0334 \pm 0.0054$ | $0.0359 \pm 0.0087$ |
| R-PEARL(ours) | N/A | $0.0696 \pm 0.004$ | $0.0319 \pm 0.007$ | $0.0377 \pm 0.008$ |
| B-PEARL(ours) | N/A | $\textbf{0.0644} \pm \textbf{0.001}$ | $\textbf{0.0290} \pm \textbf{0.003}$ | $0.0353 \pm 0.002$ |

complexity. Notably, SignNet-full, BasisNet, and SPE are unable to handle these datasets due to their quadratic memory complexity.

## 6.4 Graph regression on Molecular Graphs

We also evaluate our model on the task of predicting the penalized water-octanol partition coefficient (logP) for molecules from the ZINC dataset (Irwin et al., 2012; Dwivedi et al., 2023). We use the standard split for this dataset, which entails 10,000 molecules for training, 1,000 for validation, and another 1,000 for testing. We report the mean and standard deviation of the MAE for the model achieving the highest validation accuracy, averaged over 4 different seeds. The results can be found in Table 2. We observe that B-PEARL achieves the best results, and also the best generalization gap between the competing methods. It is also notable that R-PEARL and B-PEARL also outperform all the remaining competing methods.

We conduct experiments on the DrugOOD dataset (Ji et al., 2022). The dataset evaluates models on out-of-distribution (OOD) generalization, focusing on three specific types of domain shifts: Assay, Scaffold, and Size. The Assay, Scaffold, and Size splits test the ability to generalize to different bioassays, molecules with different structures, and molecules of different sizes, respectively. The results are presented in Table 3. We observe that stable methods generally work better than SignNet and BasisNet, which have reduced stability. B-PEARL improves the AUC by $9.5\%$ compared to BasisNet, and $3.8\%$ compared to SignNet in Size OOD generalization. B-PEARL achieves a $4.6\%$ improvement in AUC over both SignNet and BasisNet in Scaffold OOD generalization. Furthermore, B-PEARL demonstrates an advantage over SPE in size OOD generalization. This improvement is likely attributed to the linear scaling of SPE's stability bound with graph size, compared to the square-root scaling of PEARL's stability bound, which enhances its size generalization capabilities.

## 6.5 Large-scale Link Prediction on Relational Databases (RelBench)

We also test the performance of the proposed PEARL on large-scale link prediction using the Stack Exchange Q&A Website Database. To that end we utilize the rel-stack dataset for the relational deep learning benchmark (RelBench) (Fey et al.; Robinson et al., 2024). Rel-stack is a temporal and heterogeneous graph with approximately 38 million nodes. We consider two different tasks: i) `user-post-comment`, where we predict a list of existing posts that a user will comment on in the next two years, and ii) `post-post-related`, where we predict a list of existing posts that users will link a given post to in the next two years. The results for the two tasks can be found in Table 4.

The backbone model for this RelBench task is a heterogeneous identity-aware GNN (You et al., 2021) and all methods are trained with a batch size of 20. From Table 4 we observe that PEARL has an $11\%$ benefit over the identity-aware backbone model with no PE on the `user-post-comment` task and a $2\%$ benefit on the `post-post-related` task. PEARL works similarly to SignNet-8S but with lower complexity. It also works similarly to SignNet-8L on the `user-post-comment` task, while achieving $5\%$ better performance than SignNet-8L on the `post-post-related` task.

Table 3: Binding Affinity AUROC results (5 random seeds) on DrugOOD. PEARL outperforms SignNet and BasisNet, and performs comparably to SPE, while maintaining lower computational complexity.

| Domain | PE Method | ID-Val (AUC) | ID-Test (AUC) | OOD-Val (AUC) | OOD-Test (AUC) |
|---|---|---|---|---|---|
| **Assay** | No PE | 92.92±0.14 | 92.89±0.14 | 71.02±0.79 | 71.68±1.10 |
| | PEG | 92.51±0.17 | 92.57±0.22 | 70.86±0.44 | 71.98±0.65 |
| | SignNet | 92.26±0.21 | 92.43±0.27 | 70.16±0.56 | **72.27±0.97** |
| | BasisNet | 88.96±1.35 | 89.42±1.18 | 71.19±0.72 | 71.66±0.05 |
| | SPE | 92.84±0.20 | 92.94±0.15 | 71.26±0.62 | **72.53±0.66** |
| | SPE-8S | 92.36±0.18 | 92.62±0.10 | 70.71±0.47 | 71.72 ± 0.71 |
| | R-PEARL(ours) | 92.71±0.10 | 92.92±0.12 | 70.57±0.72 | **72.24±0.30** |
| | B-PEARL(ours) | 90.54±0.89 | 90.81 ±0.79 | 70.53±0.67 | 71.22±0.42 |
| **Scaffold** | No PE | 96.56±0.10 | 87.95±0.20 | 79.07±0.97 | 68.00±0.60 |
| | PEG | 95.65±0.29 | 86.20±0.14 | 79.17±0.97 | 69.15±0.75 |
| | SignNet | 95.48±0.34 | 86.73±0.56 | 77.81±0.70 | 66.43±1.06 |
| | BasisNet | 85.80±3.75 | 78.44±2.45 | 73.36±1.44 | 66.32±5.68 |
| | SPE | 96.32±0.28 | 88.12±0.41 | 80.03±0.58 | **69.64±0.49** |
| | SPE-8S | 96.44±0.079 | 87.88±0.45 | 79.34±0.50 | 68.72±0.63 |
| | R-PEARL(ours) | 96.09±0.32 | 88.01±0.43 | 78.72±0.02 | 69.20±1.00 |
| | B-PEARL(ours) | 96.06±0.29 | 87.56±0.81 | 79.86±0.58 | **69.51±0.62** |
| **Size** | No PE | 93.78±0.12 | 93.60±0.27 | 82.76±0.04 | **66.04±0.70** |
| | PEG | 92.46±0.35 | 92.67±0.23 | 82.12±0.49 | 66.01±0.10 |
| | SignNet | 93.30±0.43 | 93.20±0.39 | 80.67±0.50 | 64.03±0.70 |
| | BasisNet | 86.04±4.01 | 85.51±4.04 | 75.97±1.71 | 60.79±3.19 |
| | SPE | 92.46±0.35 | 92.67±0.23 | 82.12±0.49 | 66.02 ± 0.49 |
| | SPE-8S | 93.68±0.20 | 93.86±0.12 | 83.04±0.63 | 65.74 ± 2.2 |
| | R-PEARL(ours) | 93.32±0.34 | 93.92±0.20 | 82.09±0.44 | 65.89 ± 1.30 |
| | B-PEARL(ours) | 93.18 ± 0.45 | 93.29 ± 0.46 | 83.14 ± 0.37 | **66.58 ± 0.67** |

Table 4: Validation and test mean average precision (MAP) on large-scale RelBench recommendation tasks. PEARL has an 11% benefit over the backbone model with no PE on the `user-post-comment` task and a 2% benefit on the `post-post-related` task.

| Task | Evaluation | No PE | SignNet-8L | SignNet-8S | B-PEARL(ours) | R-PEARL(ours) |
|---|---|---|---|---|---|---|
| `User-post-comment` | Val. MAP | 15.20 | 15.33 | **15.47** | 15.13 | 15.24 |
| | Test MAP | 12.47 | 13.76 | 13.77 | 13.80 | **13.87** |
| `Post-post-related` | Val. MAP | 8.10 | 7.90 | 7.70 | 8.00 | **8.40** |
| | Test MAP | 10.73 | 10.39 | 10.86 | **10.94** | 10.86 |

# 7 RELATED WORK

The works that are mostly relevant to our work can be grouped in 4 categories: i) Eigenvector-based Positional Encodings, e.g., (Dwivedi & Bresson, 2021; Rampášek et al., 2022; Kreuzer et al., 2021a; Mialon et al., 2021; Feldman et al., 2022; Huang et al.; Zhang et al.); ii) Graph Neural Networks with unique node identifiers, e.g., (Loukas, 2019; Abboud et al., 2021; Sato et al., 2021; Abboud et al., 2021; Sato et al., 2021; Eliasof et al., 2023); iii) Graph Representation Learning with Structural Encodings, e.g., (Li et al., 2020; Ying et al., 2021; You et al., 2019; 2021; Dwivedi et al.; Ma et al., 2023; Kanatsoulis & Ribeiro); iv) (Wang et al., 2022; Srinivasan & Ribeiro; Murphy et al., 2018). A detailed discussion can be found in Appendix A.

# 8 CONCLUSION

In this paper, we proposed a novel framework for learnable positional encodings (PEs) that addresses key limitations in existing eigenvector-based methods, particularly in terms of stability, expressive power, scalability, and genericness. By leveraging message-passing GNNs as nonlinear mappings of eigenvectors, we designed efficient PEs that maintain permutation equivariance through the use of statistical pooling functions. Our approach not only ensures high expressiveness and stability but also significantly reduces computational complexity. Experimental results demonstrate that our method consistently outperforms lightweight eigenvector-based PEs and matches the performance of full eigenvector-based methods, all while offering substantial improvements in computational efficiency. These findings open new avenues for developing scalable, expressive, and robust graph representation techniques, paving the way for advancements in graph-based learning tasks.

ACKNOWLEDGMENTS

We gratefully acknowledge the support of NSF under Nos. OAC-1835598 (CINES), CCF-1918940 (Expeditions), DMS-2327709 (IHBEM), IIS-2403318 (III); Stanford Data Applications Initiative, Wu Tsai Neurosciences Institute, Stanford Institute for Human-Centered AI, Chan Zuckerberg Initiative, Amazon, Genentech, GSK, Hitachi, SAP, and UCB.

The content is solely the responsibility of the authors and does not necessarily represent the official views of the funding entities.

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

## A  RELATED WORK

**Eigenvector-based Positional Encodings:** Positional encodings are a crucial component in applying transformers to graph data and further integrating structural information in graph neural networks (GNNs). A notable approach for such positional encodings (PEs) is the use of Laplacian eigenvectors. These eigenvector-based PEs have been shown to enhance performance in transformers on graph-related tasks, as demonstrated in (Dwivedi & Bresson, 2021) and (Rampášek et al., 2022). Additionally, they can be incorporated in attention mechanisms as seen in (Kreuzer et al., 2021a), (Mialon et al., 2021), and (He et al., 2023). Laplacian eigenvectors can also be used to improve performance in the context of GNNs (Kim et al., 2022).

However, eigenvector-based positional encodings face challenges with stability and sign ambiguity. Small structural changes in graphs can cause significant change in eigenvectors and their corresponding positional encodings. In addition, the sign ambiguity of eigenvectors can introduce unwanted inconsistencies in these positional encodings. Works such as (Lim et al.) and (Huang et al.) address these issues by designing sign-invariant or basis-invariant models to produce these PEs, or by making the PEs more robust and stable. (Zhang et al.) introduced the concept of the expressive power of spectral invariant GNNs, which are GNN architectures augmented with invariant spectral features, and provided a unified theoretical framework to analyze them. Feldman et al. (2022) used eigenvector-based heat kernels to generate node embeddings to overcome the limitations of the WL test. Geisler et al. (2024) combine spatial and spectral graph filters in a unified GNN architecture.

**Randomized Graph Neural Networks** Initializing GNNs with unique node identifiers to enhance the expressive power has been first proposed by (Loukas, 2019; Abboud et al., 2021; Sato et al., 2021). In particular, (Abboud et al., 2021) and (Sato et al., 2021) used random node features as inputs to GNNs, leading to enhanced function approximation, though at the expense of permutation equivariance, a key property in graph learning. Eliasof et al. (2023) proposed a method for generating PEs in graph neural networks by leveraging random feature propagation, inspired by the power iteration method and its generalizations. The core of their approach involves concatenating several intermediate steps to compute the dominant eigenvectors of a propagation matrix. Dupty et al. proposed a randomization method that approximates the individualization-refinement technique through particle filtering. The particle filtering GNN (PF-GNN) employs a 1-WL-based initialization method, which is subsequently refined with particle filtering sampling to overcome the 1-WL limitations.

**Graph Representation Learning with Structural Encodings:** Structural encodings are also important in capturing aspects of a graph's structure, such as connectivity and neighborhood information. (Li et al., 2020) uses distance PEs for GNNs (distance from an anchor node) using shortest paths and random walks. One approach is using distance-based information between nodes through methods like shortest paths or random walks, to captural structural information for transformers (Ying et al., 2021) (You et al., 2019) (You et al., 2021). Other methods learn structural PEs directly. For instance, (Dwivedi et al.) learn embeddings that are initialized with Laplacian eigenvectors or random walks. Similarly, (Ma et al., 2023) learn a linear combination of the Laplacian for creating relative PEs.

**Node Embedding Methods:** One foundational approach to capturing meaningful graph representations is through node embeddings. DeepWalk (Perozzi et al., 2014) and node2vec (Grover & Leskovec, 2016) are early instances of these approaches and leverage random walk strategies to learn node embeddings on graphs. Although these methods show the significance of capturing structural information, they lack expressivity and do not incorporate many learnable components.

**Equivariant pooling:** Similar techniques to ours have been introduced in (Wang et al., 2022), which generate PEs by applying transformations on the Laplacian. On the other hand, Kanatsoulis & Ribeiro recently analyzed the capability of GNNs to count substructures using expectation pooling functions. Srinivasan & Ribeiro explored the equivalence between node embeddings and structural representations, showing that the expectation of node embeddings can serve as structural representations of the graph, and proposed methods to sample informative node embeddings for enhanced graph representation learning. Finally, Murphy et al. (2018) investigated models of permutation-invariant functions as averages of permutation-sensitive functions applied to all reorderings of a group.

## A.1    COMPARING PEARL TO STRUCTURAL PEs

The proposed PEARL framework can provably count important substructures in any graph, such as cycles, cliques, and quasi-cliques. More importantly, it can generalize the counting function to graphs not seen during training, demonstrating the robust generalization ability of PEARL. This naturally invites comparison with methods that explicitly compute these substructures independently. Below, we summarize the key comparison points with such methods:

**Expressivity**: PEARL is not limited to pre-defined motifs, such as cycles or cliques. It can compute other potentially important substructures, such as dense subgraphs, chordal cycles, or combinations of motifs, that explicit counting methods might omit simply because they are not pre-specified. Notably, the number of possible motifs in a graph grows combinatorially, highlighting the flexibility and breadth of PEARL.

**Complexity**: Explicitly counting high-order motifs, especially at the node level, can be computationally expensive. PEARL bypasses this challenge by learning to capture these structures implicitly, making it more scalable to large and complex graphs.

**Bias**: Predetermining which motifs to count introduces bias into the model. For example, molecular graphs often benefit from detecting cycles, while social networks emphasize cliques or dense subgraphs. In contrast, PEARL is task-agnostic and allows the data to guide which motifs are most relevant, adapting to the specific requirements of the application. On the flip side, when the training data have small sizes, learning can benefit from the specific biases that structural PEs admit.

## B    PROOF OF PROPOSITION 3.1

Under the assumptions of Proposition 3.1 the GNN has the following recursive formula:

$$\boldsymbol{X}^{(l)} = \text{MLP}\left(\boldsymbol{X}^{(l-1)}, \boldsymbol{S}\boldsymbol{X}^{(l-1)}\right) = \text{MLP}^{(-1)}\left(\sigma\left(\sum_{k=0}^{K-1}\boldsymbol{S}^k\boldsymbol{X}^{(l-1)}\boldsymbol{H}_k^{(l)}\right)\right), \qquad (13)$$

where $\text{MLP}^{(-1)}$ denotes all the layers of MLP except the first layer, and $K = 2$. We compute the eigenvalue decomposition $\boldsymbol{S}^k = \boldsymbol{V}\boldsymbol{\Lambda}^k\boldsymbol{V}^T$, and use some extra algebraic manipulations.

$$\boldsymbol{X}^{(l)} = \text{MLP}^{(-1)}\left(\sigma\left(\sum_{k=0}^{K-1}\boldsymbol{V}\boldsymbol{\Lambda}^k\boldsymbol{V}^T\boldsymbol{X}^{(l-1)}\boldsymbol{H}_k^{(l)}\right)\right) \qquad (14)$$

$$= \text{MLP}^{(-1)}\left(\sigma\left(\sum_{k=0}^{K-1}\sum_{n=1}^{N}\lambda_n^k\boldsymbol{v}_n\boldsymbol{v}_n^T\boldsymbol{X}^{(l-1)}\boldsymbol{H}_k^{(l)}\right)\right) \qquad (15)$$

$$= \text{MLP}^{(-1)}\left(\sigma\left(\sum_{k=0}^{K-1}\sum_{n=1}^{N}\lambda_n^k\boldsymbol{v}_n\left[\boldsymbol{v}_n^T\boldsymbol{X}^{(l-1)}[:, 1], \ldots, \boldsymbol{v}_n^T\boldsymbol{X}^{(l-1)}[:, F_{l-1}]\right]\boldsymbol{H}_k^{(l)}\right)\right), \quad (16)$$

where $\boldsymbol{V}[:, n] = \boldsymbol{v}_n$ and $\boldsymbol{\Lambda}[n, n] = \lambda_n$. We now focus on the output of the first MLP layer $\boldsymbol{X}^{(l,1)}$, $\boldsymbol{X}^{(l)} = \text{MLP}^{(-1)}\left(\boldsymbol{X}^{(l,1)}\right)$. Then each feature of $\boldsymbol{X}^{(l,1)}$ can be written as follows:

$$\boldsymbol{X}^{(l,1)}[:, f] = \sigma\left(\sum_{k=0}^{K-1}\sum_{n=1}^{N}\lambda_n^k \boldsymbol{v}_n\left[\boldsymbol{v}_n^T\boldsymbol{X}^{(l-1)}[:, 1], \ldots, \boldsymbol{v}_n^T\boldsymbol{X}^{(l-1)}[:, F_{l-1}]\right]\boldsymbol{H}_k^{(l)}[:, f]\right) \quad (17)$$

$$= \sigma\left(\sum_{k=0}^{K-1}\sum_{n=1}^{N}\lambda_n^k \boldsymbol{v}_n \sum_{i=1}^{F_{l-1}}\boldsymbol{H}_k^{(l)}[i, f]\boldsymbol{v}_n^T\boldsymbol{X}^{(l-1)}[:, i]\right) \quad (18)$$

$$= \sigma\left(\sum_{n=1}^{N}\sum_{i=1}^{F_{l-1}}\sum_{k=0}^{K-1}\lambda_n^k \boldsymbol{H}_k^{(l)}[i, f] < \boldsymbol{v}_n, \boldsymbol{X}^{(l-1)}[:, i] > \boldsymbol{v}_n\right) \quad (19)$$

$$= \sigma\left(\sum_{n=1}^{N}\boldsymbol{W}[n, f]\boldsymbol{v}_n\right), \quad (20)$$

where

$$\boldsymbol{W}[n, f] = \sum_{i=1}^{F_{l-1}}\sum_{k=0}^{K-1}\lambda_n^k \boldsymbol{H}_k^{(l)}[i, f] < \boldsymbol{v}_n, \boldsymbol{X}^{(l-1)}[:, i] > . \quad (21)$$

As a result $\boldsymbol{X}^{(l)} = \sigma\left(\boldsymbol{V}\boldsymbol{W}\right)$. When $\boldsymbol{S}$ is not symmetric we can replace $\boldsymbol{v}_n$ with $\boldsymbol{V}^{-1}[n, :]$. This concludes the proof.

## C  BASIS UNIVERSALITY OF PEARL

We consider the general form of PEARL:

$$\boldsymbol{P} = \rho\left[\Phi\left(\mathcal{G}, \boldsymbol{q}^{(1)}\right), \ldots, \Phi\left(\mathcal{G}, \boldsymbol{q}^{(M)}\right)\right] \in \mathbb{R}^{N \times d_P}, \quad (22)$$

where $\rho$ is an unrestricted equivariant function and $\Phi$ is a message-passing GNN with skip connections. We let $\boldsymbol{q}^{(m)} = \boldsymbol{e}_m$ and $M = N$. From Proposition 3.1 we have:

$$\boldsymbol{X}^{(l)} = \text{MLP}\left(\boldsymbol{V}\right) = \text{MLP}^{(-1)}\left(\sigma\left(\boldsymbol{V}\boldsymbol{W}\right)\right) \quad (23)$$

$$\boldsymbol{W}[n, f] = \sum_{i=1}^{F_{l-1}}\sum_{k=0}^{K-1}\lambda_n^k \boldsymbol{H}_k^{(l)}[i, f]\langle\boldsymbol{v}_n, \boldsymbol{X}^{(l-1)}[:, i]\rangle. \quad (24)$$

We omit the nonlinearities from the GNN, and for $\boldsymbol{X}^{(0)} = \boldsymbol{e}_m$ we have:

$$\boldsymbol{X}^{(K)} = \boldsymbol{V}\boldsymbol{W}, \quad \boldsymbol{W}[n, f] = \langle\boldsymbol{v}_n, \boldsymbol{e}_m\rangle\sum_{k=0}^{K-1}\boldsymbol{h}_k[f]\lambda_n^k, \quad (25)$$

As a result, $\boldsymbol{W}[n, f]$ is a polynomial on the eigenvalues $\tilde{h}_f\left(\lambda_n\right) = \sum_{k=0}^{K-1}\boldsymbol{h}_k[f]\lambda_n^k$ scaled by $\langle\boldsymbol{v}_n, \boldsymbol{e}_m\rangle$. We will now use the following lemma.

**Lemma C.1** *Let $\mathcal{G}$ be a graph with $N$ nodes and GSO $\boldsymbol{S} \in \mathbb{R}^{N \times N}$. Let $\mathcal{S} = \{\{\lambda_1, \ldots, \lambda_N\}\}$ be the multiset of eigenvalues of $\boldsymbol{S}$; $\mathcal{S}$ can have repeated elements (eigenvalues). In addition, let $\mathcal{M} = \{\mu_1, \ldots, \mu_q\}$ be the ordered set of all distinct (non-repeated) eigenvalues of $\boldsymbol{S}$. We can always design a poynomial filter such that*

$$\tilde{h}\left(\lambda\right) = \begin{cases} \gamma\left(\mu_f\right), & \text{if } \lambda = \mu_f \\ 0, & \text{if } \lambda = \mu_j \neq \mu_f \end{cases} \quad (26)$$

*Proof: Let*

$$\begin{bmatrix} \tilde{h}\left(\mu_1\right) \\ \tilde{h}\left(\mu_2\right) \\ \vdots \\ \tilde{h}\left(\mu_q\right) \end{bmatrix} = \begin{bmatrix} 1\ \mu_1\ \mu_1^2 \dots \mu_1^{K-1} \\ 1\ \mu_2\ \mu_2^2 \dots \mu_2^{K-1} \\ \vdots \\ 1\ \mu_q\ \mu_q^2 \dots \mu_q^{K-1} \end{bmatrix} \begin{bmatrix} h_0 \\ h_1 \\ \vdots \\ h_{K-1} \end{bmatrix} = \boldsymbol{B}\boldsymbol{h} \tag{27}$$

$\boldsymbol{B}$ *is a Vandermonde matrix and when* $K = q$, *the determinant of* $\boldsymbol{B}$ *takes the following form:*

$$det\left(\boldsymbol{B}\right) = \prod_{1 \le i < j \le q} \left(\mu_i - \mu_j\right). \tag{28}$$

*Since the values* $\mu_i$ *are distinct,* $\boldsymbol{B}$ *has full column rank and there exists a polynomial* $\tilde{h}$ *with unique parameters* $\boldsymbol{h} = \boldsymbol{B}^{-1}\boldsymbol{e}_i\gamma\left(\mu_f\right)$ *such that* $\tilde{h}\left(\lambda\right) = \gamma\left(\mu_f\right)$ *if* $\lambda = \mu_f$, *and* $\tilde{h}\left(\lambda\right) = 0$ *if* $\lambda = \mu_j \ne \mu_f$.

Using Lemma C.1, we can design $\tilde{h}_f\left(\lambda_n\right) = \sum_{k=0}^{K-1} \boldsymbol{h}_k[f]\lambda_n^k$ such that

$$\tilde{h}_f\left(\lambda_n\right) = \begin{cases} 1, & \text{if } \lambda_n = \mu_f \\ 0, & \text{if } \lambda_n = \mu_j \ne \mu_f \end{cases} \tag{29}$$

Under this parametrization, $\boldsymbol{X}^{(K)}$ takes the form:

$$\boldsymbol{X}^{(K)} = \left[\boldsymbol{V}_{\mu_1}\boldsymbol{V}_{\mu_1}^T\boldsymbol{e}_m, \dots, \boldsymbol{V}_{\mu_q}\boldsymbol{V}_{\mu_q}^T\boldsymbol{e}_m\right] \in \mathbb{R}^{N \times q}, \tag{30}$$

where $\boldsymbol{V}_{\mu_f}$ is the eigenspace (orthogonal space of the eigenvectors) corresponding to eigenvalue $\mu_f$. Since we independently feed $\boldsymbol{e}_1, \dots, \boldsymbol{e}_N$ to the PEARL architecture, we will have $N$ output samples for each output feature, i.e., $N$ samples for $\boldsymbol{X}^{(K)}[:, f]$. We represent the $m-$th sample as $\boldsymbol{X}^{(K)}[:, f, m]$, and for the $f-$th output feature, we have the following samples:

$$\boldsymbol{X}^{(K)}[:, f, :] = \left[\boldsymbol{V}_{\mu_f}\boldsymbol{V}_{\mu_f}^T\boldsymbol{e}_1, \dots, \boldsymbol{V}_{\mu_f}\boldsymbol{V}_{\mu_f}^T\boldsymbol{e}_N\right] = \boldsymbol{V}_{\mu_f}\boldsymbol{V}_{\mu_f}^T, \tag{31}$$

We process the output samples of each feature via an equivariant function $\rho$, to get the final output embedding:

$$\boldsymbol{Y} = \rho\left(\boldsymbol{V}_{\mu_1}\boldsymbol{V}_{\mu_1}^T, \dots, \boldsymbol{V}_{\mu_q}\boldsymbol{V}_{\mu_q}^T\right). \tag{32}$$

We can choose $\rho$ to operate on each feature independently i.e.,

$$\boldsymbol{Y} = \rho\left(g^{(1)}\left(\boldsymbol{V}_{\mu_1}\boldsymbol{V}_{\mu_1}^T\right), \dots, g^{(q)}\left(\boldsymbol{V}_{\mu_q}\boldsymbol{V}_{\mu_q}^T\right)\right), \tag{33}$$

where $\left\{g^{(i)}\right\}_{i=1}^f$ is a universally approximating permutation equivariant or invariant function, such as a high-order tensor IGN (Maron et al., 2018). Equation (33) is the definition of BasisNet (Lim et al.). BasisNet universally approximates all continuous basis invariant functions, which proves that PEARL is also a universal approximator of basis invariant functions.

## D  PEARL IS A UNIVERSAL FUNCTION OF EIGENVALUES

**Theorem D.1 (Eigenvalue Universality)** *Let* $\mathcal{G}$ *be a graph with GSO* $\boldsymbol{S} = \boldsymbol{V}\boldsymbol{\Lambda}\boldsymbol{V}^T$, *and* $f$ *be any continuous function of eigenvalues* $f\left(diag\left(\boldsymbol{\Lambda}\right)\right)$. *Then there exists a GNN* $\Phi$ *and a continuous invariant function* $\rho$, *such that* $f\left(diag\left(\boldsymbol{\Lambda}\right)\right) = \rho\left[\Phi\left(\mathcal{G}, \boldsymbol{q}^{(1)}\right), \dots, \Phi\left(\mathcal{G}, \boldsymbol{q}^{(M)}\right)\right]$.

*Proof*: Going back to Eq. 33 we can set $\rho \circ g^{(i)} = \boldsymbol{1}^T diag\left(\cdot\right)$, which is an invariant function that performs graph pooling. Then the output $\boldsymbol{y} \in \mathbb{R}^q$ takes the form:

$$\boldsymbol{y} = \left[\boldsymbol{1}^T diag\left(\boldsymbol{V}_{\mu_1}\boldsymbol{V}_{\mu_1}^T\right), \dots, \boldsymbol{1}^T diag\left(\boldsymbol{V}_{\mu_q}\boldsymbol{V}_{\mu_q}^T\right)\right] = \left[\boldsymbol{1}^T\left(\boldsymbol{V}_{\mu_1}\right).^2\boldsymbol{1}, \dots, \boldsymbol{1}^T\left(\boldsymbol{V}_{\mu_q}\right).^2\boldsymbol{1}\right] \tag{34}$$

$$= \left[\text{mult}\left(\mu_1\right), \dots, \text{mult}\left(\mu_q\right)\right], \tag{35}$$

where $\text{mult}\left(\mu_i\right)$ is the multiplicity of eigenvalue $\mu_i$.

Using Lemma C.1, we can design $\tilde{h}_f(\lambda_n) = \sum_{k=0}^{K-1} \boldsymbol{h}_k[f]\lambda_n^k$ such that

$$\tilde{h}_f(\lambda_n) = \begin{cases} \mu_f, & \text{if } \lambda_n = \mu_f \\ 0, & \text{if } \lambda_n = \mu_j \neq \mu_f \end{cases} \tag{36}$$

As a result we can design a set of $2q$ filters $\left\{\tilde{h}_f\right\}_{f=1}^{2q}$ such that

$$\tilde{h}_{2f-1}(\lambda_n) = \begin{cases} \mu_f, & \text{if } \lambda_n = \mu_f \\ 0, & \text{if } \lambda_n = \mu_j \neq \mu_f \end{cases} \tag{37}$$

$$\tilde{h}_{2f}(\lambda_n) = \begin{cases} 1, & \text{if } \lambda_n = \mu_f \\ 0, & \text{if } \lambda_n = \mu_j \neq \mu_f \end{cases} \tag{38}$$

Under this parametrization, $\boldsymbol{X}^{(K)}$ takes the form:

$$\boldsymbol{X}^{(K)} = \left[\boldsymbol{V}_{\mu_1}\mu_1\boldsymbol{V}_{\mu_1}^T\boldsymbol{e}_m, \boldsymbol{V}_{\mu_1}\boldsymbol{V}_{\mu_1}^T\boldsymbol{e}_m, \ldots, \boldsymbol{V}_{\mu_q}\mu_q\boldsymbol{V}_{\mu_q}^T\boldsymbol{e}_m, \boldsymbol{V}_{\mu_q}\boldsymbol{V}_{\mu_q}^T\boldsymbol{e}_m\right] \in \mathbb{R}^{N \times 2q}, \tag{39}$$

and according to previous analysis we have

$$\boldsymbol{Y} = \rho\left(\boldsymbol{V}_{\mu_1}\mu_1\boldsymbol{V}_{\mu_1}^T, \boldsymbol{V}_{\mu_1}\boldsymbol{V}_{\mu_1}^T, \ldots, \boldsymbol{V}_{\mu_q}\mu_q\boldsymbol{V}_{\mu_q}^T, \boldsymbol{V}_{\mu_q}\boldsymbol{V}_{\mu_q}^T\right). \tag{40}$$

We can choose $\rho$ to be an invariant function that involves graph pooling operations:

$$\rho = \text{MLP}\left[\boldsymbol{1}^T\text{diag}(\cdot), \ldots, \boldsymbol{1}^T\text{diag}(\cdot)\right] \tag{41}$$

Then the output $\boldsymbol{y} \in \mathbb{R}^{2q}$ takes the form:

$$\begin{aligned} \boldsymbol{y} &= \text{MLP}\left[\boldsymbol{1}^T\text{diag}\left(\boldsymbol{V}_{\mu_1}\mu_1\boldsymbol{V}_{\mu_1}^T\right), \boldsymbol{1}^T\text{diag}\left(\boldsymbol{V}_{\mu_1}\boldsymbol{V}_{\mu_1}^T\right), \ldots, \boldsymbol{1}^T\text{diag}\left(\boldsymbol{V}_{\mu_q}\mu_q\boldsymbol{V}_{\mu_q}^T\right), \boldsymbol{1}^T\text{diag}\left(\boldsymbol{V}_{\mu_q}\boldsymbol{V}_{\mu_q}^T\right)\right] \\ &= \text{MLP}\left[\boldsymbol{1}^T\left(\boldsymbol{V}_{\mu_1}\right)^{.2}\boldsymbol{1}\mu_1, \boldsymbol{1}^T\left(\boldsymbol{V}_{\mu_1}\right)^{.2}\boldsymbol{1}, \ldots, \boldsymbol{1}^T\left(\boldsymbol{V}_{\mu_q}\right)^{.2}\boldsymbol{1}\mu_q, \boldsymbol{1}^T\left(\boldsymbol{V}_{\mu_q}\right)^{.2}\boldsymbol{1}\right] \tag{42} \\ &= \text{MLP}\left[\text{mult}(\mu_1)\mu_1, \text{mult}(\mu_1)\ldots, \text{mult}(\mu_q)\mu_q, \text{mult}(\mu_q)\right], \tag{43} \end{aligned}$$

which is a universal function of eigenvalues. This concludes our proof.

## E    SAMPLE COMPLEXITY

To prove Theorem 4.3 and characterize the sample complexity of our approach, we will use this version of Chebychef's inequality (Boucheron et al., 2003):

$$P\left(\frac{1}{M}\left|\sum_{m=1}^M\left(\boldsymbol{P}^{(m)} - \mathbb{E}[\Phi(\mathcal{G}, \boldsymbol{q})]\right)\right| \geq \epsilon\right) \leq \frac{\text{var}(\Phi(\mathcal{G}, \boldsymbol{q}))}{M \cdot \epsilon^2}. \tag{44}$$

To establish a bound for the variance of the output $\Phi(\mathcal{G}, \boldsymbol{q})$, we begin by analyzing how pointwise nonlinearity affects the variance of a random variable. Let $X$ be a random variable with variance $\text{Var}(X)$, and let $\sigma$ be a Lipschitz continuous function with Lipschitz constant $C_\sigma$. Our goal is to examine the impact of applying $\sigma$ to $X$, specifically focusing on how it influences the variance of the transformed variable $\sigma(X)$.

### E.1    EFFECT OF POINTWISE ACTIVATION TO THE VARIANCE OF A RANDOM VARIABLE

Since $\sigma$ is Lipschitz continuous with constant $C_\sigma$, for any values of $X$ and $\mathbb{E}[X]$, we can apply the Lipschitz condition:

$$|\sigma(X) - \sigma(\mathbb{E}[X])| \leq C_\sigma|X - \mathbb{E}[X]|.$$

Squaring both sides, we have

$$(\sigma(X) - \sigma(\mathbb{E}[X]))^2 \leq C_\sigma^2(X - \mathbb{E}[X])^2.$$

Taking the expectation of both sides gives us the following:

$$\mathbb{E}[(\sigma(X) - \sigma(\mathbb{E}[X]))^2] \leq C_\sigma^2\mathbb{E}[(X - \mathbb{E}[X])^2].$$

Since $\mathbb{E}[(X - \mathbb{E}[X])^2] = \mathrm{Var}\,(X)$, this simplifies:

$$\mathbb{E}[(\sigma\,(X) - \sigma\,(\mathbb{E}[X]))^2] \leq C_\sigma^2 \mathrm{Var}\,(X).$$

Now we expand the left-hand side of the previous equation.

$$\mathbb{E}[(\sigma\,(X) - \sigma\,(\mathbb{E}[X]))^2] = \mathbb{E}\left[(\sigma\,(X) - \mathbb{E}[\sigma\,(X)] + \mathbb{E}[\sigma\,(X)] - \sigma\,(\mathbb{E}[X]))^2\right] \tag{45}$$

$$= \mathbb{E}[(\sigma\,(X) - \mathbb{E}\,[\sigma\,(X)])^2] + (\mathbb{E}[\sigma\,(X)] - \sigma\,(\mathbb{E}[X]))^2 \tag{46}$$

$$+ 2\mathbb{E}[< (\sigma\,(X) - \mathbb{E}[\sigma\,(X)]), (\mathbb{E}\,[\sigma\,(X)] - \sigma\,(\mathbb{E}[X])) >] \tag{47}$$

$$= \mathbb{E}[(\sigma\,(X) - \mathbb{E}\,[\sigma\,(X)])^2] + (\mathbb{E}[\sigma\,(X)] - \sigma\,(\mathbb{E}[X]))^2 \tag{48}$$

$$= \mathrm{Var}\,(\sigma\,(X)) + (\mathbb{E}[\sigma\,(X)] - \sigma\,(\mathbb{E}[X]))^2, \tag{49}$$

Let $\mu = (\mathbb{E}[\sigma\,(X)] - \sigma\,(\mathbb{E}[X]))$, then the variance of $\sigma(X)$ is bounded:

$$\mathrm{Var}(\sigma(X)) \leq C_\sigma^2 \mathrm{Var}(X) - \mu^2 \leq C_\sigma^2 \mathrm{Var}(X). \tag{50}$$

This shows that the Lipschitz constant $C_\sigma$ acts as a scaling factor on the variance of the random variable. If $C_\sigma$ is large, the variance of $\sigma(X)$ can be significantly larger, and if $C_\sigma$ is small, it can shrink the variance accordingly. For the majority of nonlinearities used in deep learning, such as ReLU, sigmoid, and hyperbolic tangent, $C_\sigma = 1$, and $\mathrm{Var}(\sigma(X)) \leq \mathrm{Var}(X)$.

## E.2 Effect of Graph Convolution to the variance of a random node signal

The next step is to study the effect of graph convolution (linear message-passing) operations to a set of node features. In particular, let $\boldsymbol{X}^{(l)} \in \mathbb{R}^{N \times F_{l-1}}$ be the node input to the $l-th$ GNN layer. Then we define $\boldsymbol{Z}^{(l)} \in \mathbb{R}^{N \times F_l}$ as the following:

$$\boldsymbol{X}^{(l)} = \sigma\left(\boldsymbol{Z}^{(l)}\right), \ \boldsymbol{Z}^{(l)} = \sum_{k=0}^{K-1} \boldsymbol{S}^k \boldsymbol{X}^{(l-1)} \boldsymbol{H}_k \tag{51}$$

After some algebraic manipulations, we have

$$\boldsymbol{Z}^{(l)} = \sum_{k=0}^{K-1} \boldsymbol{S}^k \sum_{f=1}^{F_{l-1}} \boldsymbol{X}^{(l-1)}[:, f] \boldsymbol{H}_k[f, :]^T = \sum_{f=1}^{F_{l-1}} \sum_{k=0}^{K-1} \boldsymbol{S}^k \boldsymbol{X}^{(l-1)}[:, f] \boldsymbol{H}_k[f, :]^T, \tag{52}$$

and each feature of $\boldsymbol{Z}^{(l)}$ can be cast as the following:

$$\boldsymbol{Z}^{(l)}[:, d] = \sum_{f=1}^{F_{l-1}} \sum_{k=0}^{K-1} \boldsymbol{H}_k[f, d] \boldsymbol{S}^k \boldsymbol{X}^{(l-1)}[:, f], \ \ d \in \{1, \ldots, F_l\}. \tag{53}$$

The above equation implies that each feature $\boldsymbol{Z}^{(l)}[:, d]$ is generated by a summation over $F_{l-1}$ features of the following type:

$$\boldsymbol{z} = \sum_{k=0}^{K-1} \boldsymbol{h}_k \boldsymbol{S}^k \boldsymbol{x} = \boldsymbol{H}\,(\boldsymbol{S})\,\boldsymbol{x} \tag{54}$$

We assume that the norm of $\boldsymbol{H}\,(\boldsymbol{S}) = \sum_k h_k \boldsymbol{S}^k$ is bounded, i.e., $\|\boldsymbol{H}\,(\boldsymbol{S})\| \leq \beta$.

As a result, we will first analyze the variance of $\boldsymbol{z}$ when the input $\boldsymbol{x}$ has the following covariance matrix:

$$\mathbb{E}\left[(\boldsymbol{x} - \mathbb{E}\,[\boldsymbol{x}])(\boldsymbol{x} - \mathbb{E}\,[\boldsymbol{x}])^T\right] = \boldsymbol{C} \tag{55}$$

The covariance of $\boldsymbol{z}$ is therefore written as follows:

$$\mathbb{E}\left[(\boldsymbol{z} - \mathbb{E}\,[\boldsymbol{z}])(\boldsymbol{z} - \mathbb{E}\,[\boldsymbol{z}])^T\right] = \boldsymbol{H}\,(\boldsymbol{S})\,\boldsymbol{C}\boldsymbol{H}\,(\boldsymbol{S}) = \sum_{k=0}^{K-1} h_k \boldsymbol{S}^k \boldsymbol{C} \sum_{m=0}^{K-1} h_m \boldsymbol{S}^m \tag{56}$$

$$= \sum_{k=0}^{K-1} \sum_{m=0}^{K-1} h_k h_m \boldsymbol{S}^k \boldsymbol{C} \boldsymbol{S}^m \tag{57}$$

and the variance for each individual variable $\boldsymbol{z}[i]$ is

$$\text{var}\left(\boldsymbol{z}\left[i\right]\right) = \sum_{k=0}^{K-1}\sum_{l=0}^{K-1} h_k h_l \boldsymbol{S}^k[i,:]^T \boldsymbol{Q} \boldsymbol{S}^l[:,i] \tag{58}$$

$$= \sum_{k=0}^{K-1}\sum_{l=0}^{K-1} h_k h_l \sum_{m\in\mathcal{N}_i^{(k)}}\sum_{n\in\mathcal{N}_i^{(l)}} \boldsymbol{S}^k[i,m]\boldsymbol{S}^l[i,n]\text{cov}\left(\boldsymbol{x}\left[m\right],\boldsymbol{x}\left[n\right]\right) \tag{59}$$

$$\leq \sum_{k=0}^{K-1}\sum_{l=0}^{K-1} h_k h_l \sum_{m\in\mathcal{N}_i^{(k)}}\sum_{n\in\mathcal{N}_i^{(l)}} \boldsymbol{S}^k[i,m]\boldsymbol{S}^l[i,n] \left|\text{cov}\left(\boldsymbol{x}\left[m\right],\boldsymbol{x}\left[n\right]\right)\right| \tag{60}$$

$$\leq \sum_{k=0}^{K-1}\sum_{l=0}^{K-1} h_k h_l \sum_{m\in\mathcal{N}_i^{(k)}}\sum_{n\in\mathcal{N}_i^{(l)}} \boldsymbol{S}^k[i,m]\boldsymbol{S}^l[i,n] \max_i\left(\text{var}\left(\boldsymbol{x}\left[i\right]\right)\right) \tag{61}$$

$$\leq \sum_{k=0}^{K-1}\sum_{l=0}^{K-1} h_k h_l \text{deg}_{\max}^k \text{deg}_{\max}^l \max_i\left(\text{var}\left(\boldsymbol{x}\left[i\right]\right)\right) \tag{62}$$

$$\leq \sum_{k=0}^{K-1} h_k \text{deg}_{\max}^k \sum_{l=0}^{K-1} h_l \text{deg}_{\max}^l \max_i\left(\text{var}\left(\boldsymbol{x}\left[i\right]\right)\right) \leq \beta^2 \max_i\left(\text{var}\left(\boldsymbol{x}\left[i\right]\right)\right), \tag{63}$$

where $\text{deg}_{\max}^k$ is the maximum degree of $\boldsymbol{S}^k$, and is equal to $\text{deg}_{\max}^k = 1$, when $\boldsymbol{S}$ is the normalized adjacency matrix or the random walk matrix. The inequality in (61) follows from the Cauchy-Schwartz inequality, the inequality in (62) follows from the definition of $\boldsymbol{S}^k$, and the last inequality in (63) is due the boundedness of the operator $\boldsymbol{H}\left(S\right)$.

Overall,

$$\text{var}\left(\boldsymbol{z}\left[i\right]\right) \leq \beta^2 \max_i\left(\text{var}\left(\boldsymbol{x}\left[i\right]\right)\right). \tag{64}$$

The final step is to analyze the the variance of a random variable that is a sum of dependent random variables, $\boldsymbol{z}[i] = \sum_{f=1}^{F_{l-1}} \boldsymbol{z}_f[i]$. Then

$$\text{var}\left(\boldsymbol{z}\left[i\right]\right) = \mathbb{E}\left[\left(\sum_{f=1}^{F_{l-1}} \boldsymbol{z}_f[i]\right)^2\right] = \sum_{f=1}^{F_{l-1}}\sum_{g=1}^{F_{l-1}} \mathbb{E}\left[\boldsymbol{z}_f[i],\boldsymbol{z}_g[i]\right] \leq \sum_{f=1}^{F_{l-1}}\sum_{g=1}^{F_{l-1}} \left|\mathbb{E}\left[\boldsymbol{z}_f[i],\boldsymbol{z}_g[i]\right]\right| \tag{65}$$

$$\leq F_{l-1}^2 \max_f\left(\text{var}\left(\boldsymbol{z}_f\left[i\right]\right)\right), \tag{66}$$

where the last inequality in (61) follows from the Cauchy-Schwartz inequality, and is quadratic with respect to the length of the GNN layer. Combining Eq. (50), (64), and (65, we conclude:

$$\text{var}\left[\boldsymbol{X}^{(l)}\right] \leq C_\sigma^2 \beta^2 F_{l-1}^2 \max\left(\text{var}\left[\boldsymbol{X}^{(l-1)}\right]\right), \tag{67}$$

If we assume that $F_l = F$ for all hidden layers, we have

$$\text{var}\left[\boldsymbol{X}^{(L)}\right] \leq \left(C_\sigma \beta F\right)^{2L} \max\left(\text{var}\left[\boldsymbol{X}\right]\right). \tag{68}$$

In our proposed approach $\max\left(\text{var}\left[\boldsymbol{X}\right]\right) = 1$. If we further assume that $C_\sigma = 1$, which is typically the case in practice, and that $\beta = 1/F$, implying that the magnitude of trainable parameters is inversely proportional to the number of hidden dimensions in each layer, then we obtain:

$$\text{var}\left[\boldsymbol{X}^{(L)}\right] \leq 1. \tag{69}$$

This concludes the proof for Theorem 4.3, which is repeated below:

**Theorem E.1 (Sample Complexity)** *Let $\boldsymbol{P}$ denote the output of the architecture described in Eq. (8), for a graph $\mathcal{G}$ with i.i.d. initial node features with unit variance. Also let $\Phi$ be an $L-$layer GNN*

*described by Eq. (3), with F hidden dimensions at each layer. If $C_\sigma = 1$ and $\beta = 1/F$, the number of samples required such that:*

$$\left| \frac{1}{M} \sum_{m=1}^{M} \Phi\left(\mathcal{G}, \boldsymbol{q}^{(m)}\right) - \mathbb{E}\left[\Phi\left(\mathcal{G}, \boldsymbol{q}\right)\right] \right| < \epsilon, \text{ with probability at least } 1 - \delta, \tag{70}$$

*satisfies:*

$$M \leq \frac{1}{\delta \cdot \epsilon^2}. \tag{71}$$

Theorem 4.3 bounds the number of samples required for the output of R-PEARL to approximate the expected value $\mathbb{E}\left[\Phi(\mathcal{G}, \mathbf{q})\right]$, where $\mathbf{q} \in \mathbb{R}^N$ is a random vector. It also describes the number of samples required for R-PEARL to preserve equivariance. To be more precise, since $\mathbf{q}$ is processed using an equivariant GNN $\Phi(\mathcal{G}, \cdot)$, the resulting distribution of $\Phi(\mathcal{G}, \mathbf{q})$ is permutation equivariant. Consequently, the expected value $\mathbb{E}\left[\Phi(\mathcal{G}, \mathbf{q})\right]$ is also permutation equivariant.

R-PEARL computes the empirical mean $\hat{\mathbb{E}}\left[\Phi(\mathcal{G}, \mathbf{q})\right]$, enabling it to approximate $\mathbb{E}\left[\Phi(\mathcal{G}, \mathbf{q})\right]$ with high precision. In essence, R-PEARL achieves equivariance with increasing accuracy as the number of samples $M$ grows. For very small sample sizes, R-PEARL is not equivariant; however, in practice, a modest number of samples is sufficient to render it effectively equivariant. This is evident in our experiments. In Tables 1 and 2, the "GIN + rand id" model corresponds to R-PEARL with a single sample, which is not equivariant. We observe that "GIN + rand id" performs significantly worse—more than four times worse in logP prediction for molecular graphs—compared to R-PEARL.

## F    PROOF OF COROLLARY 4.4

To prove Corollary 4.4, we assume that the pointwise nonlinearities $\sigma$ are elementwise power functions, i.e., $\sigma(\cdot) = (\cdot)^p$ for integer values of $p \geq 2$. Using this assumption, we can apply Theorem K.1 from (Kanatsoulis & Ribeiro) to establish the result. For approximate results, classical smooth nonlinearities such as the hyperbolic tangent, the sigmoid, or the Swish function can be employed and analyzed via their Taylor series expansion around zero:

$$\sigma(x) = \sum_{p=0}^{K-1} \frac{\sigma^{(p)}(0)}{p!} x^p, \tag{72}$$

where $\sigma^{(p)}$ represents the $p$-th derivative of $\sigma(x)$ evaluated at 0. It is straightforward to observe that elementwise power functions appear in this expansion, enabling approximate cycle counting. In any case, the proposed PEs will contain rich information relevant to cycle counts.

B-PEARL achieves the same level of expressivity as R-PEARL. Specifically, there exists a parametrization $\Phi$, defined by Eq. (3), such that B-PEARL can count the number of 3-, 4-, 5-, 6-, and 7-node cycles in which each node participates for any given graph. This equivalence holds because B-PEARL can be thought as a special case of R-PEARL, where the input $\boldsymbol{x}$ has higher-order moments:

$$\mathbb{E}\left[\boldsymbol{x} \circ \boldsymbol{x} \circ \cdots \circ \boldsymbol{x}\right] = \boldsymbol{I} \circ \boldsymbol{I} \circ \cdots \circ \boldsymbol{I}.$$

The structural expressivity of R-PEARL depends on $\mathbb{E}\left[\boldsymbol{x} \circ \boldsymbol{x} \circ \cdots \circ \boldsymbol{x}\right]$, which generate tensor product operations, as shown in (Kanatsoulis & Ribeiro).

## G    STABILITY ANALYSIS

Let $\tilde{\mathcal{G}}$ be a perturbed version of graph $\mathcal{G}$, with GSOs $\tilde{\boldsymbol{S}}$ and $\boldsymbol{S}$, respectively. We consider two perturbation models, i.e., additive and relative perturbation:

$$\text{Additive perturbation model:} \quad \tilde{\boldsymbol{S}} = \boldsymbol{S} + \boldsymbol{E} \tag{73}$$

$$\text{Relative perturbation model:} \quad \tilde{\boldsymbol{S}} = \boldsymbol{S}\boldsymbol{E} + \boldsymbol{S}\boldsymbol{E} \tag{74}$$

In order to measure the distance between $\tilde{\boldsymbol{S}}$ and $\boldsymbol{S}$, as well as the GNN outputs when the input graphs are perturbed versions of each other we define the distance modulo permutation:

**Definition G.1 (Linear operator distance modulo permutation)** *(Gama et al., 2020) Given linear operators $\boldsymbol{A}$ and $\tilde{\boldsymbol{A}}$ we define the operator distance modulo permutation as*

$$\left\| \boldsymbol{A} - \tilde{\boldsymbol{A}} \right\|_{\mathcal{P}} = \min_{\boldsymbol{\Pi}} \max_{\boldsymbol{x}:\|\boldsymbol{x}\|=1} \left\| \boldsymbol{\Pi}^T(\boldsymbol{A}\boldsymbol{x}) - \tilde{\boldsymbol{A}}(\boldsymbol{\Pi}^T\boldsymbol{x}) \right\|, \tag{75}$$

*where $\boldsymbol{\Pi}$ is a permutation matrix.*

### G.1 LIPSCHITZ AND INTEGRAL LIPSCHITZ FILTERS

Next, we need to define the notion of Lipschitz and Integral Lipschitz filters. First, we note that graph filters are pointwise operators in the graph frequency domain, i.e.,

$$\boldsymbol{H}(\boldsymbol{S}) = \sum_{k=0}^{K-1} h_k \boldsymbol{S}^k = \sum_{k=0}^{K-1} h_k \boldsymbol{V}\Lambda^k \boldsymbol{V}^T = \boldsymbol{V}\left(\sum_{k=0}^{K-1} h_k \boldsymbol{\Lambda}^k\right) \boldsymbol{V}^T. \tag{76}$$

We can therefore define the graph frequency response of the filter as:

$$h(\lambda) = \sum_{k=0}^{K-1} h_k \lambda^k. \tag{77}$$

To continue our analysis we define the following filter types.

**Definition G.2 (Lipschitz Filter)** *(Gama et al., 2020) Given a filter $\boldsymbol{h} = \{h_k\}_{k=0}^{K-1}$ its frequency response $h(\lambda)$ is given by equation 77. We say the filter is Lipschitz if there exists a constant $C > 0$ such that for all $\lambda_1$ and $\lambda_2$,*

$$\left| h(\lambda_2) - h(\lambda_1) \right| \leq C \left| \lambda_2 - \lambda_1 \right|. \tag{78}$$

**Definition G.3 (Integral Lipschitz Filter)** *(Gama et al., 2020) Given a filter $\boldsymbol{h} = \{h_k\}_{k=0}^{K-1}$ its frequency response $h(\lambda)$ is given by equation 77. We say the filter is integral Lipschitz if there exists a constant $C > 0$ such that for all $\lambda_1$ and $\lambda_2$,*

$$|h(\lambda_2) - h(\lambda_1)| \leq C \frac{|\lambda_2 - \lambda_1|}{|\lambda_1 + \lambda_2|/2}. \tag{79}$$

### G.2 STABILITY BOUNDS FOR RANDOM AND BASIS PEs

We can use any stability bounds for GNNs. To see this, let

$$\left\| \Phi(\mathcal{G}, \cdot)[:, f] - \Phi(\tilde{\mathcal{G}}, \cdot)[:, f] \right\|_{\mathcal{P}} \leq \Gamma \tag{80}$$

Then

$$\left\| \frac{1}{M}\sum_{m=1}^{M} \Phi(\mathcal{G}, \cdot)[:, f] - \frac{1}{M}\sum_{m=1}^{M} \Phi(\tilde{\mathcal{G}}, \cdot)[:, f] \right\|_{\mathcal{P}} \leq \frac{1}{M}\left\| \sum_{m=1}^{M} \Phi(\mathcal{G}, \cdot)[:, f] - \Phi(\tilde{\mathcal{G}}, \cdot)[:, f] \right\|_{\mathcal{P}}$$

$$\leq \frac{1}{M}\sum_{m=1}^{M} \left\| \Phi(\mathcal{G}, \cdot)[:, f] - \Phi(\tilde{\mathcal{G}}, \cdot)[:, f] \right\|_{\mathcal{P}}$$

$$\leq \Gamma \tag{81}$$

Using the previous definitions and Eq. (81) we can now use the analysis in (Gama et al., 2020) to establish the stability of the proposed PEs.

**Proposition G.1 (Stability to additive perturbations)** *Let $\tilde{\mathcal{G}}$ be a perturbed version of $\mathcal{G}$ such that $\tilde{\boldsymbol{S}} = \boldsymbol{S} + \boldsymbol{E}$ with $\|\boldsymbol{E}\| \leq \varepsilon$. Let $\Phi$ be an $L-$layer GNN described by Eq. (3), where each layer consists of Lipschitz filters with constant $C$. Under assumptions 4.1 and 4.2 with $C_\sigma = 1$ and $\beta = 1/F$, the following holds:*

$$\left\| \frac{1}{M}\sum_{m=1}^{M} \Phi(\mathcal{G}, \cdot)[:, f] - \frac{1}{M}\sum_{m=1}^{M} \Phi(\tilde{\mathcal{G}}, \cdot)[:, f] \right\|_{\mathcal{P}} \leq \left(1 + 8\sqrt{N}\right) L\varepsilon + \mathcal{O}(\varepsilon^2) \tag{82}$$

**Proposition G.2 (Stability to relative perturbations)** *Let $\tilde{\mathcal{G}}$ be a perturbed version of $\mathcal{G}$ such that $\tilde{\boldsymbol{S}} = \boldsymbol{S} + \boldsymbol{S}\boldsymbol{E} + \boldsymbol{E}\boldsymbol{S}$ with $\|\boldsymbol{E}\|_{\mathcal{P}} \leq \varepsilon$. Let $\Phi$ be an $L-$layer GNN described by Eq. (3), where each layer consists of Integral Lipschitz filters with constant $C$. Under assumptions 4.1 and 4.2 with $C_\sigma = 1$ and $\beta = 1/F$, the following holds:*

$$\left\| \frac{1}{M} \sum_{m=1}^{M} \Phi\left(\mathcal{G}, \cdot\right)[:, f] - \frac{1}{M} \sum_{m=1}^{M} \Phi\left(\tilde{\mathcal{G}}, \cdot\right)[:, f] \right\|_{\mathcal{P}} \leq 2\left(1 + 8\sqrt{N}\right) L\varepsilon + \mathcal{O}(\varepsilon^2) \quad (83)$$

# H  SPECTRAL FILTERS WITH GRAPH FILTERS

The suggested implementation in (Huang et al.) is:

$$\text{SPE}\left(\boldsymbol{V}, \boldsymbol{\Lambda}\right) = \sum_{n=0}^{N-1} \rho\left(\left[\boldsymbol{V}\text{diag}\left(\alpha_1\left(\boldsymbol{\Lambda}\right)\right)\boldsymbol{V}[n]^T, \ldots, \boldsymbol{V}\text{diag}\left(\alpha_M\left(\boldsymbol{\Lambda}\right)\right)\boldsymbol{V}[n]^T\right]\right) \quad (84)$$

$$= \sum_{n=0}^{N-1} \rho\left(\left[\boldsymbol{V}\text{diag}\left(\alpha_1\left(\boldsymbol{\Lambda}\right)\right)\boldsymbol{V}^T\boldsymbol{e}_n, \ldots, \boldsymbol{V}\text{diag}\left(\alpha_M\left(\boldsymbol{\Lambda}\right)\right)\boldsymbol{V}^T\boldsymbol{e}_n\right]\right), \quad (85)$$

where $\rho$ represents multiple GIN layers. If we assume that $\alpha_m$ are analytic element wise functions, then we can take the taylor series expansion and represent $\alpha_m$ as a polynomial. Then SPE can be cast as:

$$\text{SPE}\left(\boldsymbol{V}, \boldsymbol{\Lambda}\right) = \sum_{n=0}^{N-1} \rho\left(\left[\boldsymbol{V}\text{diag}\left(\sum_{k=0}^{K-1} h_k^1 \boldsymbol{\Lambda}^k\right)\boldsymbol{V}^T\boldsymbol{e}_n, \ldots, \boldsymbol{V}\text{diag}\left(\sum_{k=0}^{K-1} h_k^M \boldsymbol{\Lambda}^k\right)\boldsymbol{V}^T\boldsymbol{e}_n\right]\right) \quad (86)$$

$$= \sum_{n=0}^{N-1} \rho\left(\left[\boldsymbol{V}\sum_{k=0}^{K-1} h_k^1 \boldsymbol{\Lambda}^k\boldsymbol{V}^T\boldsymbol{e}_n, \ldots, \boldsymbol{V}\sum_{k=0}^{K-1} h_k^M \boldsymbol{\Lambda}^k\boldsymbol{V}^T\boldsymbol{e}_n\right]\right) \quad (87)$$

$$= \sum_{n=0}^{N-1} \rho\left(\left[\sum_{k=0}^{K-1} h_k^1 \boldsymbol{V}\boldsymbol{\Lambda}^k\boldsymbol{V}^T\boldsymbol{e}_n, \ldots, \sum_{k=0}^{K-1} h_k^M \boldsymbol{V}\boldsymbol{\Lambda}^k\boldsymbol{V}^T\boldsymbol{e}_n\right]\right) \quad (88)$$

$$= \sum_{n=0}^{N-1} \rho\left(\left[\sum_{k=0}^{K-1} h_k^1 \boldsymbol{A}^k\boldsymbol{e}_n, \ldots, \sum_{k=0}^{K-1} h_k^M \boldsymbol{A}^k\boldsymbol{e}_n\right]\right) = \sum_{n=0}^{N-1} \rho\left(\sum_{k=0}^{K-1} \boldsymbol{A}^k\boldsymbol{e}_n\boldsymbol{h}_k^T\right) \quad (89)$$

The expression in Eq. (89) coincides with the B-PEARL architecture, which concludes our proof.

# I  IMPLEMENTATION DETAILS

All results for the SPE, SignNet, and BasisNet models using only 8 eigenvectors were either sourced from their original papers, when available, or obtained by retraining the original models with 8 eigenvectors corresponding to the 8 largest or smallest eigenvalues. All other baseline results were sourced from their original papers. For both the R-PEARL and B-PEARL models, batch normalization is applied within the $\Phi$ layers. Additionally, when $K > 2$, the output of the first $\Phi$ layer is passed through a shallow MLP consisting of 1 or 2 layers before continuing through the remaining layers.

For the REDDIT datasets, we train R-PEARL with 30-40 samples and $K = 2$, omitting the first layer described in Eq. (3). During execution we use 40-200 samples. R-PEARL uses 8 GIN layers with 64 hidden dimensions, each followed by batch normalization, to generate the PEs. For SignNet, we used 4 GIN layers with 128 hidden dimensions and batch normalization. We experimented with both 4 and 8 layers for SignNet and found that using 4 layers yielded the best results. The PEs are further processed by a base model consisting of 6 additional GIN layers. In R-PEARL, skip connections are applied across the GIN layers, followed by a linear layer at the end. SignNet uses residual connections instead, and also uses MLP encoders for the eigenvectors, as well as a Set Transformer (Lee et al., 2019). For both models, we train with a batch size of 128 for REDDIT-BINARY and REDDIT-MULTI.

Table 5: Estimated runtime per epoch in Hours:Minutes for different models on RelBench.

| Task | No PE | SignNet-largest | SignNet-smallest | B-PEARL | R-PEARL |
|------|-------|-----------------|------------------|---------|---------|
| user-post-comment | 00:03 | 00:07 | 01:22 | 00:13 | 00:17 |
| post-post-related | 00:01 | 00:08 | 00:26 | 00:05 | 00:01 |

On the ZINC datasets, R-PEARL, B-PEARL, and SPE use a batch size of 128. The base model for each is a 4-layer GINE. Similar to the SPE model, we inject the original positional encoding into every layer by passing it through an MLP and adding it to the layer's input. Notably, our model employs 8 GIN layers with 40 hidden units for $\Phi$, whereas SPE uses an 8-layer GIN with 128 hidden units, in addition to 3 MLPs. For R-PEARL we use 50-120 samples and $K = 12$, while for B-PEARL we use $K = 4$.

For the DrugOOD datasets, R-PEARL, B-PEARL, SPE, and SignNet all use 4-layer GINE base models. Both R-PEARL and B-PEARL use a 3-layer GIN for $\Phi$. To process positional encodings, in addition to a 3-layer GIN, SPE uses 16 3-layer MLPs on the Scaffold and Size splits, while SignNet uses a 3-layer MLP across all splits. At each layer of the base model, all models concatenate the original positional encodings with the input features. In the Assay and Size splits we use R-PEARL with $K = 14$ and 80 samples, while for the Scaffold split, we use $K = 16$ and 200 samples.

For the RelBench tasks, both R-PEARL and B-PEARL models use $K = 7$. The R-PEARL model employs a 5-layer GIN with 40 hidden units and 120 samples. The B-PEARL model uses either a 5-layer or a 7-layer GIN, depending on the task: 5 layers for post-post-related and 7 layers for user-post-comment, both with the same number of hidden units. For the SignNet models, we use an 8-layer GIN with batch normalization to generate positional encodings. The positional encodings from the models are incorporated as additional node features for each node. The original node features are generated by a Tabular ResNet model, which learns representations over the various node features. These combined features are then fed into the base GNN model. All models follow the same setup as in RelBench for the base model, which employs a 2-layer ID-GNN. Training is conducted with a batch size of 20.

Table 5 presents the runtime of our end-to-end PE models on the RelBench tasks. Notably, our R-PEARL and B-PEARL achieve shorter runtimes compared to SignNet-smallest across both tasks.

For the Peptides-struct dataset we use an 8-layer GatedGCN base model for B-PEARL, and a 6-layer GINE model for R-PEARL. We use a 9-layer GIN to generate the positional encodings for both models, with $K = 4$ for R-PEARL and $K = 1$ for B-PEARL.

For our experiments and model training pipeline we follow the codebases of (Huang et al.) and (Lim et al.), using Python, PyTorch (Paszke et al., 2019), and the PyTorch Geometric (Fey & Lenssen, 2019) libraries. Our code can be found here: `https://github.com/ehejin/Pearl-PE`.

## J ADDITIONAL EXPERIMENTS

### J.1 EXPERIMENTS ON GRAPH ISOMORPHISM

We conduct experiments on the Circular Skip Link (CSL) dataset (Murphy et al., 2019) which is the golden standard when it comes to benchmarking GNNs for graph isomorphism (Dwivedi et al., 2023). CSL contains 150 4-regular graphs, where the edges form a cycle and contain skip-links between nodes. Each graph consists of 41 nodes and 164 edges and belongs to one of 10 classes. Message-passing GNNs with WL-related PEs fail to classify these graphs and classification is completely random. This is due to the inability of the WL algorithm to handle regular graphs.

The proposed PEARL architectures, however, have no issue in processing regular graphs and achieve $100\%$ classification accuracy. In particular, let $\Phi$ be a two-layer GNN, where each layer is defined by Eq. (3) with $K = 5$, $F_0 = F_1 = 1$, and $\sigma(\cdot) = \texttt{ReLU}(\cdot)$. The generated node PE $\boldsymbol{P}$ is processed by a summation graph pooling function to produce a scalar embedding for each graph. Then, both B-PEARL and R-PEARL can perfectly classify the CSL graphs with $100\%$ classification accuracy, for any randomly generated trainable weights. This means that B-PEARL and R-PEARL can

Table 6: B-PEARL PE for every class of the CSL graphs. B-PEARL can perfectly classify the CSL graphs with $100\%$ classification accuracy.

| | | | | | CLASS | | | | |
|---|---|---|---|---|---|---|---|---|---|
| 0 | 1 | 2 | 3 | 4 | 5 | 6 | 7 | 8 | 9 |
| 0 | 27351.6 | 8800.2 | 25779.9 | 20458.4 | 17197.2 | 15861.3 | 24055.6 | 4106.8 | 17667.0 |

perfectly classify the CSL graphs without any training. For example let $\Phi$ consist of two identical layers with parameters $(h_0, h_1, h_2, h_3, h_4) = (0, 1, -\frac{1}{2}, \frac{1}{3}, -\frac{1}{4})$. The output $\mathbf{1}^T \mathbf{P}$ of B-PEARL is presented in Table 6. The output remains the same for all graphs within the same class, and differs distinctly for graphs belonging to different classes. Consequently, perfect classification accuracy can be achieved by feeding the B-PEARL encoding into a simple linear classifier or even a linear assignment algorithm.

## J.2 EXPERIMENTS ON PEPTIDES-STRUCT DATASET

Table 7: Test MAE on the Peptides-struct dataset. B-PEARL achieves the second-best performance, behind Graph ViT, within a 500k parameter budget. Graph ViT utilizes additional random walk information, while B-PEARL should learn this information via training.

| R-PEARL | B-PEARL | GPS | SAN+LapPE | SAN+RWSE | GNN-AK+ | SUN | Graph ViT |
|---|---|---|---|---|---|---|---|
| $0.247_{\pm 0.001}$ | $0.248_{\pm 0.001}$ | $0.252_{\pm 0.001}$ | $0.268_{\pm 0.004}$ | $0.255_{\pm 0.001}$ | $0.274_{\pm 0.00}$ | $0.250_{\pm 0.001}$ | $0.245_{\pm 0.002}$ |

We conduct experiments on the Peptides-struct dataset from the Long Range Graph Benchmark dataset (Dwivedi et al., 2022). This dataset comprises over 15,000 graphs containing more than 2 million nodes in total, with each graph ranging from 8 to 444 nodes. It is designed to evaluate a model's ability to capture long-range interactions. The metric used is Mean Absolute Error (MAE), and the task involves regressing on the 3D structure of peptides to predict properties such as their length. We compare the performance of R-PEARL and B-PEARL against state-of-the-art models within a 500k parameter budget: GPS (Rampášek et al., 2022), SAN (Kreuzer et al., 2021b), SUN (Frasca et al., 2022), GNN-AK (Zhao et al., 2022), and ViT (He et al., 2023). The results can be found in Table 7.

We observe that B-PEARL achieves the second-best performance, trailing Graph ViT by only 0.002. However, it is important to note that Graph ViT leverages additional random walk information, whereas B-PEARL must learn this information during training. This distinction can be particularly significant, especially given that both models operate within the same constrained 500k parameter budget.

# K ABLATION STUDIES

## K.1 ABLATION ON DIFFERENT BACKBONE MODELS

Table 8: Test MAE of B-PEARL and R-PEARL with different backbones within a 500k parameter budget on the ZINC dataset.

| Model | GINE | GatedGCN | PNA |
|---|---|---|---|
| B-PEARL | $0.0679 \pm 0.0026$ | $0.0765 \pm 0.0018$ | $0.0740 \pm 0.0010$ |
| R-PEARL | $0.0721 \pm 0.0045$ | $0.0810 \pm 0.0039$ | $0.0946 \pm 0.0032$ |

We conduct additional base model ablations for our model, using PNA and GatedGCN in addition to GINE on the ZINC dataset (Corso et al., 2020; Li et al., 2016). Our results are shown in Table 8. We observe that PEARL demonstrates strong performance across various backbones. B-PEARL consistently outperforms equivalent SignNet models with the same backbones, while R-PEARL outperforms SignNet with the GatedGCN backbone (Lim et al.). All models in the table were kept within the 500k parameter budget.

Table 9: logP Prediction in ZINC with different R-PEARL $K$ values.

| PE Method | #PEs | K | Test MAE | Training MAE | General. Gap |
|---|---|---|---|---|---|
| SignNet-8S | 8 | N/A | $0.1034 \pm 0.0056$ | $0.0418 \pm 0.0101$ | $0.0602 \pm 0.0112$ |
| SignNet | Full | N/A | $0.0853 \pm 0.0026$ | $0.0349 \pm 0.0078$ | $0.0502 \pm 0.0103$ |
| BasisNet-8S | 8 | N/A | $0.1554 \pm 0.0048$ | $0.0513 \pm 0.0053$ | $0.1042 \pm 0.0063$ |
| BasisNet | Full | N/A | $0.1555 \pm 0.0124$ | $0.0684 \pm 0.0202$ | $0.0989 \pm 0.0258$ |
| SPE-8S | 8 | N/A | $0.0736 \pm 0.0007$ | $0.0324 \pm 0.0058$ | $0.0413 \pm 0.0057$ |
| SPE | Full | N/A | $0.0693 \pm 0.0040$ | $0.0334 \pm 0.0054$ | $0.0359 \pm 0.0087$ |
| R-PEARL(ours) | N/A | 1 | $0.0699 \pm 0.002$ | $0.0366 \pm 0.006$ | $0.0333 \pm 0.007$ |
| R-PEARL(ours) | N/A | 2 | $0.0831 \pm 0.005$ | $0.0725 \pm 0.0125$ | $0.0106 \pm 0.008$ |
| R-PEARL(ours) | N/A | 12 | $0.0699 \pm 0.002$ | $0.0366 \pm 0.006$ | $0.0333 \pm 0.007$ |
| B-PEARL(ours) | N/A | 1 | $0.0644 \pm 0.001$ | $0.0290 \pm 0.003$ | $0.0353 \pm 0.002$ |
| B-PEARL(ours) | N/A | 4 | $0.0680 \pm 0.0023$ | $0.0381 \pm 0.004$ | $0.0299 \pm 0.0033$ |
| B-PEARL(ours) | N/A | 12 | $0.0676 \pm 0.0016$ | $0.0403 \pm 0.0104$ | $0.0273 \pm 0.0090$ |

### K.2 ABLATION ON $K$

We also report our results on the ZINC dataset (Irwin et al., 2012) with alternate values of $K$ for R-PEARL and B-PEARL . In the case of $K = 2$ we omit the first layer described in Eq. (3), using solely an 8-layer GIN for $\phi$. These results are shown in Table9. We observe that even with $K = 2$ our model outperforms SignNet. Furthermore, even with a low $K$ value of 4, B-PEARL outperforms SPE.

### K.3 ABLATION ON NUMBER OF SAMPLES $M$

We conduct ablation studies on the number of samples used by R-PEARL across the REDDIT datasets to examine the impact of sample size on model performance. These results are illustrated in Fig. 2. For each dataset, we evaluate the original model with 200 samples on a single test fold, varying the number of samples from 1 to 200. The test accuracy is then plotted, and Monte Carlo simulation-based smoothing is applied to generate the plots. Notably, we observe that model performance begins to converge with as few as 10 samples—an order of magnitude lower than the graph size.

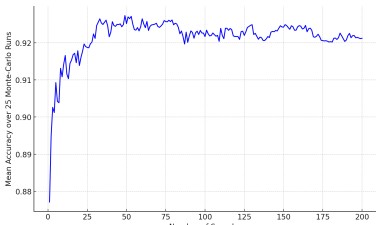
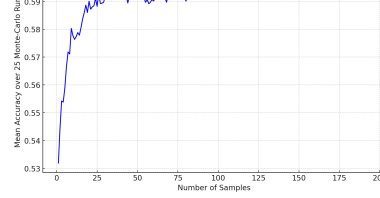

(a) Sample Ablation on REDDIT-BINARY      (b) Sample Ablation on REDDIT-MULTI-5K

Figure 2: Ablation studies on the sample size for R-PEARL; it converges with only a few samples.

### K.4 ABLATION ON THE NUMBER OF GNN LAYERS IN PEARL

Table 10: Ablation on the number of GIN layers in B-PEARL for the ZINC Dataset over 4 seeds.

| # GIN Layers | 3 | 5 | 7 | 9 |
|---|---|---|---|---|
| Test MAE | $0.0701 \pm 0.005$ | $0.067 \pm 0.003$ | $0.069 \pm 0.001$ | $0.0644 \pm 0.001$ |

The results in Table 10 show that B-PEARL achieves strong performance on the ZINC dataset even with less layers in the GNN producing the positional encodings. With only 5 layers, B-PEARL outperforms SPE.

### K.5 ABLATION ON DIFFERENT PARAMETER SIZE

Table 11 presents our ablation study on the number of parameters for the ZINC dataset. Notably, all models outperform SignNet even within the 500k parameter budget. Furthermore, B-PEARL achieves superior performance compared to SPE on full eigenvectors, even with fewer parameters.

Table 11: logP Prediction in ZINC over number of parameters.

| PE Method | #Parameters | Test MAE | Training MAE | General. Gap |
|-----------|-------------|----------|--------------|--------------|
| SignNet | 487k | $0.0853 \pm 0.0026$ | $0.0349 \pm 0.0078$ | $0.0502 \pm 0.0103$ |
| SPE | 650k | $0.0693 \pm 0.0040$ | $0.0334 \pm 0.0054$ | $0.0359 \pm 0.0087$ |
| R-PEARL | 644k | $0.0699 \pm 0.002$ | $0.0366 \pm 0.006$ | $0.0333 \pm 0.007$ |
| B-PEARL | 644k | $0.0644 \pm 0.001$ | $0.0290 \pm 0.003$ | $0.0353 \pm 0.002$ |
| R-PEARL | 487k | $0.0696 \pm 0.004$ | $0.0319 \pm 0.007$ | $0.0377 \pm 0.008$ |
| B-PEARL | 487k | $0.0655 \pm 0.004$ | $0.0252 \pm 0.005$ | $0.0403 \pm 0.004$ |

