# OpenReview forum: "Learning Efficient Positional Encodings with Graph Neural Networks"
_ICLR.cc/2025/Conference — ICLR 2025 Poster_

### Official Review · Reviewer_8uLD · 2024-11-02

**Soundness:** 3
**Presentation:** 3
**Contribution:** 2
**Rating:** 6
**Confidence:** 4

**Summary:**

The paper deals with learning effective positional encodings for graph structures to be used in conjunction with graph neural networks. Previously, eigenvectors and random features  have been used for such posiitonal encodings. In this paper, the authors first observe that the message passing GNNs are akin to non-linear functions of eighenvectors and therefore can be used to learn positional encodings which can function similarly to eigenvector-based positional encodings. Following this observation, they propose PEARL, a GNN based approach which takes in random feature inputs on the graph (without accompanying node-features) and learns to output encodings which can be interpreted as positional encodings. To preserve permutation invariance upto the expectation, they use multiple IID samples and aggregate with equivarient sample statistics. The authors show that such a approach preserves  stability, improves expressive power, scalability, and genericness. Experiments are presented to validate the approach.

**Strengths:**

1. The paper addresses an important problem of learning positional encodings on graphs
1. The approach of learning PEs as outputs of GNN with random feature inputs on the graph is well-motivated with the observation discussed that GNNs are non-linear functions of eigenvectors of graph shift operators.
1. The paper is well-written and presentation is good.

**Weaknesses:**

1. Essentially, the outputs of GNNs on random input features are taken as positional encodings. However, there are no loss functions to bias learning towards such functions. While it is clear the random inputs along with GNN message passing, can encode the relative positional information and the supporting evidence that GNNs compute non-linear maps of eigenvectors of adjacency/laplacians, further regularizing loss functions may be able to make the learning encodings better. One example is enforcing the encodings of each node to be orthogonal to other encodings.
1. It is not clear to me why eigenvectors need to be used in place of random features for smaller sized graphs. Empirical results to show the difference between eigenvector inputs and random inputs for smaller sized graphs would be beneficial.
1. Important works are missing in related work and comparison experiments. In experiments, Random GNN [1] and PF-GNN [2] are not compared, which are highly related to using randomization to improve expressive power and in a way, compute positional encodings.
1. The paper claims that the proposed approach is more expressive. Then empirical results showing it is needed. For example, identifying graphs  not identifiable by GNNs would be useful. A comparison with random GNN[1] and PFGNN[2] would be beneficial in this respect.


### References:

\[1\] $~$  Sato, Ryoma, Makoto Yamada, and Hisashi Kashima. "Random features strengthen graph neural networks." Proceedings of the 2021 SIAM international conference on data mining (SDM). Society for Industrial and Applied Mathematics, 2021.

\[2\] $~$ Dupty, Mohammed Haroon, Yanfei Dong, and Wee Sun Lee. "PF-GNN: Differentiable particle filtering based approximation of universal graph representations." arXiv preprint arXiv:2401.17752 (2024).

**Questions:**

Please see weaknesses.

---

> ### Author Response · Authors · 2024-11-24
> **Summary of our response**
>
> We thank the reviewer for appreciating our work and providing valuable feedback. The reviewer’s concerns can be summarized into the following key questions:
>
> 1.  **Does our method require additional regularizers to achieve its objectives?**
> 2.  **How do random features compare to eigenvectors when they initialize node embeddings?**
> 3.  **How does our method compare to Random GNN [1] and PF-GNN [2]?**
> 4.  **How does our method empirically perform in identifying graphs that are not distinguishable by 1-WL GNNs?**
>
> **Our short answers are as follows:**
>
> **A1:**
> Our framework does not inherently require additional regularizers, which could introduce unnecessary bias and complexity, potentially affecting its generality and scalability. The universality theorem (included in the revised manuscript) and expressivity/generalizability results, supported by empirical evidence, validate this claim.
>
> **A2:**
> GNNs initialized with random features, as in the proposed R-PEARL, demonstrate both theoretical and empirical advantages over eigenvector-based GNNs, **but only when equivariance is preserved**. R-PEARL offers an elegant and effective solution in this regard. However, when equivariance is violated, as in Random GNN, eigenvector-based GNNs exhibit better performance.
>
> **A3:**
> We have already included comparisons with Random GNN in the original manuscript. Random GNN, being non-equivariant, is significantly outperformed by our proposed framework. Furthermore, our method achieves nearly **twice the performance of PF-GNN** in the logP prediction task on ZINC. However, it is worth noting that this comparison may not be entirely fair, as PF-GNN is primarily a backbone GNN and could also benefit from the enhancements proposed in our framework.
>
> **A4:**
> We added experiments using the CSL dataset, which serves as a gold standard for graph isomorphism benchmarks. Our framework achieves **100% classification accuracy**, demonstrating its strong empirical capability in identifying graphs not distinguishable by 1-WL GNNs.
>
> **Detailed responses to these points are provided below.**

---

> > ### Author Response · Authors · 2024-11-24
> > **Response to reviewer's comments (1/2)**
> >
> > > Essentially, the outputs of GNNs on random input features are taken as positional encodings. However, there are no loss functions to bias learning towards such functions. While it is clear the random inputs along with GNN message passing, can encode the relative positional information and the supporting evidence that GNNs compute non-linear maps of eigenvectors of adjacency/laplacians, further regularizing loss functions may be able to make the learning encodings better. One example is enforcing the encodings of each node to be orthogonal to other encodings.
> >
> > We sincerely thank the reviewer for their insightful comment. Incorporating orthogonality as a constraint or regularizer during training could potentially reduce the dimensionality and enhance the purity of the information captured in each embedding. However, this approach would also introduce additional computational complexity and significantly increase the difficulty of the optimization problem, as enforcing orthogonality involves solving a non-convex constraint.
> >
> > Furthermore, our framework does not inherently require additional regularizers to achieve its objectives. By design, it is capable of effectively capturing and leveraging the necessary information without relying on additional constraint, thus avoiding the associated trade-offs in complexity and optimization challenges. This is corroborated by our new universality theorem, which is included in the revised manuscript, our expressivity and generalizability results, and our new and old experiments.
> >
> > > It is not clear to me why eigenvectors need to be used in place of random features for smaller sized graphs. EmpiriIcal results to show the difference between eigenvector inputs and random inputs for smaller sized graphs would be beneficial.
> >
> >
> > We refer the reviewer to the results of Tables 1, and 2. The baseline methods include **GIN + rand id**, which uses random features; **SignNet** and **SPE**, which compute equivariant functions of eigenvectors; and **R-PEARL**, which generates equivariant node PEs using random node initialization. By analyzing these tables, we can address the following key questions:
> >
> > 1. **Do we need to explicitly compute eigenvectors and eigenvalues for effective positional encodings?**
> > 2. **Is there a clear advantage of random feature GNNs over eigenvector-based GNNs?**
> >
> > **A1: Explicit eigenvector computation is unnecessary.**
> > Both **R-PEARL** and **B-PEARL** outperform eigenvector-based approaches without explicitly computing eigenvectors or eigenvalues. Notably, R-PEARL uses random initial node features and employs an equivariant function that approximates eigenvector computations.
> >
> > **A2: The benefit of random features depends on preserving equivariance.**
> > Both **R-PEARL** and **GIN + rand id** utilize random node features. However, **R-PEARL** applies a continuous pooling function to maintain equivariance, whereas **GIN + rand id** does not. The results demonstrate that:
> > - R-PEARL outperforms eigenvector-based methods.
> > - Eigenvector-based methods outperform **GIN + rand id**.
> >
> > Thus, the critical distinction is not merely between random features and eigenvectors, but between **equivariance and non-equivariance**. Within this context, powerful equivariant GNN methods such as **R-PEARL** and **B-PEARL** consistently outperform both eigenvector-based methods and non-equivariant random-feature approaches.

---

> > > ### Author Response · Authors · 2024-11-24
> > > **Response to reviewer's comments (2/2)**
> > >
> > > > Important works are missing in related work and comparison experiments. In experiments, Random GNN [1] and PF-GNN [2] are not compared, which are highly related to using randomization to improve expressive power and in a way, compute positional encodings.
> > >
> > > We have already included Random GNN [1] in our experiments. Specifically, the GIN + rand id baseline used in Tables 1 and 2 corresponds exactly to the Random GNN from [1]. However, a key limitation of Random GNN is that it lacks permutation equivariance, which significantly limits its generalization ability. This is reflected in the performance of  GIN + rand id which is significanlty weaker than the proposed PEARL.
> > >
> > > The reviewer is correct that we did not include PF-GNN [2] in our comparisons. This is because PF-GNN is a backbone model rather than a PE framework. PF-GNN employs a 1-WL-based initialization method, which is practically it's PE. The 1-WL PE is subsequently refined using the PF-GNN backbone which resembles the individualization-refinement technique through particle filtering. That said, from the original PF-GNN paper, the best mean absolute error (MAE) reported on the ZINC dataset is 0.122, which is nearly double our 0.064 MAE. This can be attributed to the fact that PF-GNN may require a large K to achieve equivariance. It is worth noting, however, that this comparison may not be entirely fair to PF-GNN, as it was designed as a backbone GNN. In fact, PF-GNN could potentially benefit from our proposed PE framework. For instance, it could use PEARL as an initialization instead of 1-WL colorings.
> > >
> > > **We have included discussion on PF-GNN in the revised manuscript.**
> > >
> > > > The paper claims that the proposed approach is more expressive. Then empirical results showing it is needed. For example, identifying graphs not identifiable by GNNs would be useful. A comparison with random GNN[1] and PFGNN[2] would be beneficial in this respect.
> > >
> > > We thank the reviewer for their comment. Per reviewer's suggestion we have included additional experiments with the CSL dataset, that is the golden standard when it comes to comparing GNNs with respect to graph isomorphism. As you can see from the general comment and the revised manuscript PEARL is able to achieve 100% classification accuracy in CSL, which is also what PF-GNN achieves. Random GNN achieves lower accuracy due to lack of equivariance.

---

> > > > ### Author Response · Authors · 2024-12-02
> > > > **We would appreciate your feedback on our response and updated manuscript**
> > > >
> > > > Dear reviewer 8uLD,
> > > >
> > > > We greatly appreciate your valuable feedback and insightful comments, which have been instrumental in improving our paper. In the revised manuscript, we have incorporated all the requested theoretical and empirical comparisons, enhancing the overall rigor and clarity. We have also extended our evaluation by including four more datasets, where our method demonstrates remarkable performance.
> > > >
> > > > We would really appreciate it if you could find time to share your thoughts on our response and the revised manuscript.

---

### Official Review · Reviewer_KVQZ · 2024-11-02

**Soundness:** 2
**Presentation:** 2
**Contribution:** 2
**Rating:** 3
**Confidence:** 4

**Summary:**

This work proposes PEARL, using the output of Message Passing Neural Network (MPNN) with noise/basis node feature as input for PE. PEARL's stability, sample complexity, and expressivity are further analysis. Experiments illustrates that PEARL achieves good performance in graph tasks.

**Strengths:**

1. Good performance on ZINC dataset.
2. Detailed analysis of R-PEARL.

**Weaknesses:**

1. The abstract states that this work investigates positional encodings (PEs) for graphs based on four key criteria: stability, expressive power, scalability, and genericness. However, the focus is primarily on eigenvector-based PE, and the analysis is limited to the authors' proposed method, R-PEARL, regarding these criteria. A more comprehensive theoretical analysis and comparison with other PEs are needed.
2. The rationale behind using Message Passing Neural Networks (MPNNs) to generate PEs is not clearly explained. Proposition 3.1 appears to be trivial, as graphs (represented by adjacency matrices and node features) can be bijectively mapped to eigenvectors, eigenvalues, and node features. Consequently, all graph functions, including PEs, can be seen as functions of eigenvectors when parameters include both eigenvalues and eigenvectors. Therefore, the justification for choosing MPNNs as eigenvalue function over other graph models is not evident.
3. While the expectation of R-PEARL is equivariant, the actual R-PEARL method is not, due to the involvement of noise.
4. Theorem 4.3 and the claim that "our proposed PE framework is well-suited for large-scale graphs" are questionable. The proof of Theorem 4.3, specifically Formula (45), includes the maximum degree of the graph, which is latter included in $\beta$. Considering $\beta$ as a constant is problematic because large social networks often contain nodes with high degrees, which would require more samples for accurate representation.
5. Most theoretical results (Theorem 4.3, Proposition 4.1, 4.2, 4.3) are direct corollories of previous conclusions.
6. The experimental section is limited to four datasets. To strengthen the validity of the results, additional datasets should be included as in [1].
7. The PE is used in conjunction with a GNN backbone, but the backbone for PEARL appears to be fixed across all experiments. To demonstrate the broad applicability and ensure a fair comparison among different PEs, experiments on PEs with different GNN backbones should be added, as recommended in [2].


[1] Xiaoxin He, Bryan Hooi, Thomas Laurent, Adam Perold, Yann LeCun, Xavier Bresson: A Generalization of ViT/MLP-Mixer to Graphs. ICML 2023
[2] Ladislav Rampásek, Michael Galkin, Vijay Prakash Dwivedi, Anh Tuan Luu, Guy Wolf, Dominique Beaini: Recipe for a General, Powerful, Scalable Graph Transformer. NeurIPS 2022

**Questions:**

Please refer to Weaknesses.

---

> ### Author Response · Authors · 2024-11-24
> **Summary of our response (1/2)**
>
> We thank the reviewer for their valuable input. The main concerns of the reviewer can be summarized into the following points:
>
> 1. The reviewer requests further clarification regarding the rationale for using message-passing GNNs as the foundation for graph positional encodings (PEs).
> 2. The reviewer asks about the impact of noise on the permutation equivariance properties of our R-PEARL framework.
> 3.  The reviewer expresses concerns about the sample complexity of R-PEARL and its dependence on node degrees.
> 4.  The reviewer mentions that Theorem 4.3, as well as Propositions 4.1, 4.2, and 4.3, are direct corollaries of previously established results.
> 5.  The reviewer requests additional experiments to further demonstrate the effectiveness of our proposed approach.
>
> **Our short answers are as follows:**
>
> **A1:**
> To address this point we refer the reviewer to the motivation of our work. In the introduction we mention that our work is motivated by the following question. **Can we learn generic PEs that are simultaneously expressive, stable, and scalable?**
> Throughout the paper we explain that the proposed GNN-based PEs are indeed capable to to provide an affirmative answer. On the contrary, other PE methods, such as random PEs, eigenbased PEs, and structural PEs fail to jointly satisfy these properties. In a nutshell, random PEs are not stable and generalizable, eigenbased PEs cannot be simultaneously scalable and stable, and strucutral PEs add bias and complexity to our models.
>
> To address this point, we refer the reviewer to the motivation outlined in the introduction of our work. Specifically, our study is guided by the following question: **Can we design generic PEs that are simultaneously expressive, stable, and scalable?**
>
> In the manuscript, we systematically establish that the proposed GNN-based PEs effectively provide an affirmative answer to this question. In contrast, alternative PE methods fail to simultaneously fulfill these essential criteria. Specifically:
>
> -   **Random PEs** lack both stability and generalizability.
> -   **Eigen-based PEs** have an inherent trade-off between stability and scalability.
> -   **Structural PEs** introduce bias and in certain cases increase the overall complexity of the model.
>
> These limitations underscore the advantages of our approach, which successfully addresses the trade-offs encountered by existing methods.
>
> **A2:**
> We emphasize that noise does not affect B-PEARL, it only plays a role in R-PEARL. However, it is crucial to highlight that R-PEARL is not merely equivariant in expectation; it serves as an **unbiased estimator of the expectation**. This distinction implies that, with a sufficient number of samples—as theoretically established in Theorem 3.1—R-PEARL achieves practical equivariance.
>
> To empirically substantiate this claim, we measured the relative difference in the R-PEARL positional encodings generated for each node across multiple runs. The results demonstrate that the variability introduced by noise is negligible compared to the inherent differences in node features. This observation supports our claim that R-PEARL is effectively equivariant in real-world applications.
>
> **A3:**
> This is a subtle point that requires a detailed explanation. In summary, the sample complexity of our method is independent of the graph size or node degrees when the message-passing mechanism is based on normalized graph shift operators (GSOs), such as normalized adjacency, Laplacian, or random walk matrices. Notably, this is the implementation we used in practice.
>
> Even in scenarios where the message-passing mechanism resembles classical adjacency or Laplacian operators, the incorporation of batch normalization or layer normalization significantly mitigates variance, rendering it practically independent—or only slightly dependent—on the graph size. This ensures robustness and scalability of R-PEARL in real-world applications.
>
> **A4:**
> While Propositions 4.1, 4.2, and 4.3 are indeed direct corollaries of previous results, and we have renamed them as such in the revised manuscript, we do not see why this should be considered a limitation. Our contribution does not merely analyzes the stability or expressivity of an existing architecture but instead introduces a novel PE framework. Since our architecture is a GNN, it is unavoidable to inherit some properties from existing work. **Note, however, that Theorem 4.3 represents original work, is not a direct corollary of any prior conclusions. To add further novelty to the expressivity analysis of our work  we prove that PEARL is a universal approximator of basis invariant functions.**

---

> > ### Author Response · Authors · 2024-11-24
> > **Summary of our response (2/2)**
> >
> > **A5:**
> > In response to the reviewer’s suggestion, we have added experiments on three additional datasets: Rel-stack user-post-comment, Rel-stack post-post-related, and CSL.
> >
> > -   **Rel-stack user-post-comment** and **Rel-stack post-post-related** involve link prediction tasks on a large-scale, 38-million-node temporal and heterogeneous graph derived from Stack Exchange.
> > -   **CSL** is a graph classification dataset widely regarded as the gold standard for evaluating the ability of GNN methods to distinguish non-isomorphic graphs.
> >
> > The proposed PEARL framework demonstrates superior performance compared to the baselines on Rel-stack user-post-comment and Rel-stack post-post-related and achieves 100% classification accuracy on CSL, highlighting its robustness and effectiveness across diverse tasks.
> >
> > **Detailed responses to these points are provided below.**

---

> > > ### Author Response · Authors · 2024-11-24
> > > **Response to Weaknesses (1/3)**
> > >
> > > >  The abstract states that this work investigates positional encodings (PEs) for graphs based on four key criteria: stability, expressive power, scalability, and genericness. However, the focus is primarily on eigenvector-based PE, and the analysis is limited to the authors' proposed method, R-PEARL, regarding these criteria. A more comprehensive theoretical analysis and comparison with other PEs are needed.
> > >
> > > We would like to thank the reviewer for their suggestion. We have added relevant discussion in the revised manuscript which now includes comparisons with random PE, eigen-based PEs, and structural PEs.
> > >
> > >
> > >
> > > >  The rationale behind using Message Passing Neural Networks (MPNNs) to generate PEs is not clearly explained. Proposition 3.1 appears to be trivial, as graphs (represented by adjacency matrices and node features) can be bijectively mapped to eigenvectors, eigenvalues, and node features. Consequently, all graph functions, including PEs, can be seen as functions of eigenvectors when parameters include both eigenvalues and eigenvectors. Therefore, the justification for choosing MPNNs as eigenvalue function over other graph models is not evident.
> > >
> > >
> > > The reviewer raises a very interesting and valid point. While graphs can indeed be mapped to their eigenvectors and eigenvalues, this mapping is not bijective. A graph with NN nodes can correspond to at least 2N2^N distinct eigenvector matrices, as both v\mathbf{v} and −v-\mathbf{v} are valid eigenvectors. For practical graphs with eigenvector multiplicities, the number of possible eigenvector matrices becomes infinite.
> > >
> > > The justification for using MPNNs is thoroughly addressed throughout the paper. The proposed PEs are jointly **equivariant**, lightweight, stable, expressive, and generic. The stability, generality, and scalability of our approach stem from leveraging MPNNs. While MPNNs are known to be less expressive than the 1-WL test, our novel approach unlocks their potential, enabling them to achieve high expressivity properties.
> > >
> > > We agree with the reviewer that any graph function can be expressed as a function of eigenvalues and eigenvectors. However, the discriminative power of such functions in the spectral domain is not well understood. Our work provides a definitive answer in the context of MPNNs. We demonstrate that MPNNs, combined with our approach, act as universal approximators of basis-invariant eigenvector and eigenvalue functions.
> > >
> > > Furthermore, while prior works compute eigenvectors to enhance spectral expressivity, our results show that this is unnecessary. Our method serves as a universal approximator of basis-invariant spectral functions, bypassing the need for explicit eigenvector computation while achieving comparable expressivity.
> > >
> > >
> > > >  While the expectation of R-PEARL is equivariant, the actual R-PEARL method is not, due to the involvement of noise.
> > >
> > > It is true that noise plays a role in R-PEARL. However, it is important to note that R-PEARL is not merely equivariant in expectation; it is an **unbiased estimator of an expectation**. In practice, this implies that with a sufficient number of samples (as theoretically defined by Theorem 3.1), R-PEARL becomes practically equivariant.
> > >
> > > To validate this, we measured the effect of noise on the output of R-PEARL for graphs in the ZINC dataset. Specifically, we conducted five independent runs of R-PEARL, each with 100 samples, using different random seeds. To evaluate the consistency, we calculated the relative difference between the R-PEARL PE generated for each node across these runs, defined as
> > > $$d_{ij}=\frac{|\mathbf{P}[\text{run}_i]-\mathbf{P}[\text{run}_j]|}{|\mathbf{P}[\text{run}_i]|}$$.
> > >
> > > Our findings revealed that $d_{ij}$ was consistently on the order of $10^{-2}$, while the difference in PEs between different nodes within the same run was on the order of $10^{-1}$.
> > >
> > > These results demonstrate that the variability introduced by noise is negligible compared to the inherent differences in node features, justifying our claim that R-PEARL is practically equivariant in real-world applications.

---

> > > > ### Author Response · Authors · 2024-11-24
> > > > **Response to Weaknesses (2/3)**
> > > >
> > > > >  Theorem 4.3 and the claim that "our proposed PE framework is well-suited for large-scale graphs" are questionable. The proof of Theorem 4.3, specifically Formula (45), includes the maximum degree of the graph, which is latter included in  β. Considering  β  as a constant is problematic because large social networks often contain nodes with high degrees, which would require more samples for accurate representation.
> > > >
> > > > This is an excellent point raised by the reviewer and one that merits further clarification. Specifically, the parameter $\beta$ depends on the largest possible eigenvalue of the GSO. For the adjacency matrix, this largest eigenvalue is indeed related to the degree. However, in practice, we utilize a message-passing functions for PEARL, which model normalized GSOs, such as normalized Adjacency or Laplacian. In this case, the largest eigenvalue is fixed at 1 or 2, making it independent of the highest degree.
> > > >
> > > > Regardless, all our models incorporate batch or layer normalization, ensuring that the outputs are not proportional to the degrees of the graph. Therefore, for models based on the normalized GSOs (which is what we use in practice), the number of samples remains independent of the graph size. Even in other cases, normalization operations ensure that the number of samples is not dependent on the graph size.
> > > >
> > > > >  Most theoretical results (Theorem 4.3, Proposition 4.1, 4.2, 4.3) are direct corollaries of previous conclusions.
> > > >
> > > > Propositions 4.1, 4.2, and 4.3 are indeed direct corollaries of previous results, and we have renamed them as such in the revised manuscript. However, Theorem 4.3 represents original work, is not a direct corollary of any prior conclusions and it leverages concentration inequalities, a common approach in thousands of works in the literature. Importantly, the main contribution of our paper does not lie in novel theoretical analysis. Rather, it lies in the design of a novel PE framework characterized by four key properties: generality, stability, expressivity, and scalability. While two stability and expressivity have been explored in other contexts, our work proposes a novel framework that uniquely combines and implements all four.
> > > >
> > > > **To further strengthen the theoretical analysis of our approach we prove, in the revised manuscript,  that PEARL is a universal approximator of basis invariant functions. The details are presented below.**
> > > >
> > > > **Theorem [Basis Universality]**
> > > > *Let* $\mathcal{G}$ *be a graph with GSO* $\mathbf{S} = \mathbf{V} \mathbf{\Lambda} \mathbf{V}^T$, *and* $f$ *be a continuous function such that* $f(\mathbf{V}) = f(\mathbf{V} \mathbf{Q})$, $\mathbf{Q} \in \mathcal{O}\left({diag}\left(\mathbf{\Lambda}\right)\right)$, *for any eigenvalues* $\mathbf{\Lambda}$. *Then there exist GNN* $\Phi$ *and a continuous pooling function* $\rho$, *such that*:
> > > >
> > > > $$
> > > > f(\mathbf{V}) = \rho\left[ \Phi\left( \mathcal{G}, \mathbf{q}^{(1)} \right), \dots, \Phi\left( \mathcal{G}, \mathbf{q}^{(M)} \right) \right].
> > > > $$

---

> > > > > ### Author Response · Authors · 2024-11-25
> > > > > **Response to Weaknesses (3/3)**
> > > > >
> > > > > >  The experimental section is limited to four datasets. To strengthen the validity of the results, additional datasets should be included as in [1].
> > > > >
> > > > > We have added results from three additional datasets—Rel-stack user-post-comment, Rel-stack post-post-related, and CSL—as outlined in the summary of our response. These new results further validate and reinforce the effectiveness of our proposed PEARL framework. We have cited [1] in the revised manuscript.
> > > > >
> > > > > ### CSL Dataset
> > > > > We conduct experiments on the Circular Skip Link (CSL) dataset [Murphy et al., 2019] which is the golden standard when it comes to benchmarking GNNs for graph isomorphism. CSL contains 150 4-regular graphs, where the edges form a cycle and contain skip-links between nodes. Each graph consists of 41 nodes and 164 edges and belongs to one of 10 classes. Message-passing GNNs with WL-related PEs fail to classify these graphs and classification is completely random. This is due to the inability of the WL algorithm to handle regular graphs.
> > > > >
> > > > > The proposed PEARL architectures, however, have no issue in processing regular graphs and achieve $100\%$ classification accuracy.
> > > > >
> > > > > ### Rel-stack datasets
> > > > > To further empasize the scalability of our approach we test the performance of the proposed PEARL on large-scale link prediction for Stack Exchange Q\&A Website Database. To that end we utilize the rel-stack dataset for the relational deep learning benchmak (RelBench) [2]. Rel-stack is a temporal and heterogeneous graph with approximately 38 million nodes. We consider two different tasks; i) user-post-comment, where we predict a list of existing posts that a user will comment in the next two years, and ii) post-post-related, where we predict a list of existing posts that users will link a given post to in the next two years. The results for the two tasks can be found in the following table.
> > > > >
> > > > > ### Table: Validation and Test Mean Average Precision (MAP) on Large-Scale RelBench Recommendation Tasks
> > > > >
> > > > > **PEARL** achieves an 11% improvement over the backbone model with no PE on the *user-post-comment* task and a 2% improvement on the *post-post-related** task.
> > > > >
> > > > > | **Task**                  | **Evaluation** | **No PE** | **SignNet-8L** | **SignNet-8S** | **B-PEARL (ours)** | **R-PEARL (ours)** |
> > > > > |---------------------------|----------------|-----------|----------------|----------------|----------------------|----------------------|
> > > > > | **User-post-comment**     | Val. MAP       | 15.20     | 15.33          | **15.47**      | 15.13               | 15.24               |
> > > > > |                           | Test MAP       | 12.47     | 13.76          | 13.77          | 13.80               | **13.87**           |
> > > > > | **Post-post-related**     | Val. MAP       | 8.10      | 7.90           | 7.70           | 8.00                | **8.40**            |
> > > > > |                           | Test MAP       | 10.73     | 10.39          | 10.86          | **10.94**           | 10.86               |
> > > > >
> > > > >
> > > > > The backbone model for this RelBench task a heterogeneous identity-aware GNN and all methods are trained with batch size $20$. From the above table we observe that PEARL has an 11\% benefit over the identity aware backbone model with no PE on the user-post-comment task and a 2\% benefit on the post-post-related task. PEARL works similarly to SignNet-8S but with lower complexity and, and similarly to SignNet-8L on the user-post-comment task, but 5\% better than SignNet-8L on the post-post-related task. SPE could not be included in the comparisons due to it's cubic computational complexity and quadratic memory complexity.
> > > > >
> > > > > [2] Robinson, J., Ranjan, R., Hu, W., Huang, K., Han, J., Dobles, A., Fey, M., Lenssen, J.E., Yuan, Y., Zhang, Z. and He, X., RelBench: A Benchmark for Deep Learning on Relational Databases. In  The Thirty-eight Conference on Neural Information Processing Systems Datasets and Benchmarks Track.
> > > > >
> > > > >
> > > > >
> > > > > >  The PE is used in conjunction with a GNN backbone, but the backbone for PEARL appears to be fixed across all experiments. To demonstrate the broad applicability and ensure a fair comparison among different PEs, experiments on PEs with different GNN backbones should be added, as recommended in [2].
> > > > >
> > > > > The backbone architecture for PEARL and all baseline models is consistently fixed to GINE for the REDDIT, DrugOOD, and ZINC datasets, and ID-GNN for RelBench, which involves a heterogeneous graph structure. This setup ensures fair and consistent comparisons across all baselines.
> > > > >
> > > > > While we have experimented with other backbones, including GraphSAGE, GCN, and PNA, we found that GINE offers the best overall balance between performance and the number of trainable parameters. To provide additional insights, we will include ablation studies with alternative backbones in the revised manuscript.

---

> ### Comment · Reviewer_KVQZ · 2024-11-26
>
> Thank you for the detail response. However, my concerns are not solved, so I keep my score to reject. Why my concerns maintain is detailed as follows.
>
> 1. The abstract states that this work investigates positional encodings (PEs) for graphs based on four key criteria: stability, expressive power, scalability, and genericness. However, the focus is primarily on eigenvector-based PE, and the analysis is limited to the authors' proposed method, R-PEARL, regarding these criteria. A more comprehensive theoretical analysis and comparison with other PEs are needed.
>
> The authors add some discussion, but what I expect is a quantitive comparison. For example, the sample complexity comparison between R-PEARL and random PE, and the expressivity comparison between B-PEARL and structural PE.
>
> 2. The rationale behind using Message Passing Neural Networks (MPNNs) to generate PEs is not clearly explained. Proposition 3.1 appears to be trivial, as graphs (represented by adjacency matrices and node features) can be bijectively mapped to eigenvectors, eigenvalues, and node features. Consequently, all graph functions, including PEs, can be seen as functions of eigenvectors when parameters include both eigenvalues and node feature. Therefore, the justification for choosing MPNNs as eigenvalue function over other graph models is not evident.
>
> Authors agreed that Proposition 3.1 is trivial. MPNN may have many advantages, but Proposition 3.1 is not persuative and can be removed.
>
> 3. Theorem 4.3 and the claim that "our proposed PE framework is well-suited for large-scale graphs" are questionable. The proof of Theorem 4.3, specifically Formula (45), includes the maximum degree of the graph, which is latter included in β. Considering β as a constant is problematic because large social networks often contain nodes with high degrees, which would require more samples for accurate representation.
>
> Authors said by normalization, $\beta$ can be small for even graph with large node degree. But with normalization, both signal and the noise can be scaled to be small, so a smaller error bound \epsilon in Equation 9 is needed. A more systematic way is to compute signal to noise ratio.
>
> 4. Most theoretical results (Theorem 4.3, Proposition 4.1, 4.2, 4.3) are direct corollaries of previous conclusions.
>
> Authors agreed that Proposition 4.1, 4.2, 4.3 are direct corollaries of previous conclusions. We still consider that Theorem 4.3 as a direct corollary, which only computes the Lipschitz constant of GNN and applies the conclusion of Lipschitz functions' sample complexity in the original proof. Currently, the proof the Theorem 4.3 is missing.
>
> The newly added Theorem (PEARL is a universal approximator of basis invariant functions) is also doubtful. In line 907-909 of the proof, basisnet need g with universal expressivity for universality of the whole model, but PEARL uses only MPNN. Moreover, the Theorem assume the model exists for each fixed set of eigenvalue, but in practice, we need one model to work for all eigenvalue sets as the input graph changes.
>
> 5.  The experimental section is limited to four datasets. To strengthen the validity of the results, additional datasets should be included as in [1].
>
> Only one smallest synthetic dataset CSL in [1] is added. Commonly used datasets like long range graph benchmark and ogbg-molhiv in [1] are ignored. Please add them.
>
> 6. The PE is used in conjunction with a GNN backbone, but the backbone for PEARL appears to be fixed across all experiments. To demonstrate the broad applicability and ensure a fair comparison among different PEs, experiments on PEs with different GNN backbones should be added, as recommended in [2].
>
> Up to now, ablation study on GNN backbone is not added.

---

> > ### Author Response · Authors · 2024-11-29
> > **Response Summary to the Reviewer’s Latest Feedback**
> >
> > We thank the reviewer for their response. The reviewer has reiterated some of their earlier comments and raised additional concerns. The three primary issues highlighted are:
> >
> > 1.  *Sample complexity of R-PEARL for large graphs.*
> > 2.  *Correctness of Theorem 3.1 and its generalization to multiple graphs.*
> > 3.  *Additional experiments and ablation studies.*
> >
> > Below, we provide a concise summary of our responses, followed by a detailed explanation addressing all the points raised.
> >
> >
> > ----------
> >
> > ### **Summary of Responses (TL;DR)**
> >
> > **A1:**  The **sample complexity of R-PEARL is independent on the graph size, when normalized GSOs are employed in message-passing**. The signal-to-noise ratio can only benefit from larger graphs.
> >
> > **A2:**  **Theorem 3.1 is both correct and sound**. Its generalization to multiple graphs is straightforward: we assume that $\mathcal{G}$ consists of multiple graphs modeled as disconnected components. In this scenario, the GSO takes a block diagonal form, where each block corresponds to the GSO of an individual graph.
> >
> > **A3:**  We have **added experiments on the peptide-structure dataset**. PEARL achieves SOTA performance (B-PEARL MAE: 0.247, Graph ViT MAE: 0.245). It is important to note that Graph ViT leverages additional random walk information, while PEARL learns this information directly from the data under the same parameter budget.
> >
> > Overall, PEARL has undergone rigorous evaluation across 10 diverse tasks spanning 5 distinct graph types. We firmly believe that this represents an **objectively sufficient experimental setup for testing any graph ML framework.**
> >
> > 1.  **Small Molecular Graphs**:
> >     -   LogP prediction on ZINC
> >     -   Binding affinity prediction on DrugOOD (3 tasks)
> > 2. **Long-range Molecular Graphs**
> > 	- Multi-label graph regression based on the 3D structure of the peptides
> > 4.  **Social Networks**:
> >     -   Subreddit classification on REDDIT-BINARY
> >     -   Subreddit classification on REDDIT-MULTI5K
> > 5.  **Relational Databases**:
> >     -   Post-post-related recommendation on Stack Exchange
> >     -   User-post-comment recommendation on Stack Exchange
> > 6.  **Synthetic Graphs**:
> >     -   Graph classification on CSL
> >
> > Across all these tasks, the results consistently demonstrate that PEARL outperforms baseline methods and, in rare cases, matches the best-performing methods. Notably, in many instances, PEARL achieves these results with significantly lower computational and memory complexity, further showcasing its efficiency and practicality. **We also include the requested ablation studies in the revised manuscript.**

---

> > > ### Author Response · Authors · 2024-11-29
> > > **Detailed response (1/4)**
> > >
> > > >1.  The abstract states that this work investigates PEs for graphs based on four key criteria: stability, expressive power, scalability, and genericness ... A more comprehensive **theoretical analysis and comparison** with other PEs are needed.
> > >
> > > >The authors add some discussion, but what I expect is a quantitive comparison. For example, the sample complexity comparison between R-PEARL and random PE, and the expressivity comparison between B-PEARL and structural PE.
> > >
> > > We thank the reviewer for their comment. We would like to clarify that our submission is not intended as a review paper but rather proposes a novel PE framework. We understand that the phrasing in our abstract may have been somewhat ambiguous. To address this, we have revised the abstract to state: _"In this work, we identify four key properties that graph PEs should satisfy: stability, expressive power, scalability, and generality."_ This revision better reflects the scope and contributions of our work.
> > >
> > > Nevertheless, we have included a comprehensive **theoretical analysis and comparison**, as requested in the initial review. Furthermore, we have also provided quantitative comparisons across nine diverse tasks to further substantiate our approach.
> > >
> > > The reviewer has requested a **comparison between the sample complexity of R-PEARL and random PE**. We kindly note that such a comparison is not meaningful within the context of sample complexity. The reason is that for random PE, a single sample is sufficient, as there is no estimation process involved. The GNN processes this input through linear and nonlinear projections to produce the desired output. Increasing the number of input samples has both theoretical and empirical effects equivalent to increasing the dimensionality of the hidden GNN layers. This is because a linear or nonlinear combination of random vectors remains a random vector. Notably, when M=1, R-PEARL coincides with random PE. R-PEARL, however, requires a larger number of samples to accurately approximate the empirical expectation of the output PE with high precision.
> > >
> > > Regarding the **expressivity comparison between B-PEARL and structural PE**, we note that B-PEARL achieves the same level of expressivity as R-PEARL. Specifically, there exists a parametrization $\Phi$, defined by Eq. (3), such that B-PEARL can count the number of 3-, 4-, 5-, 6-, and 7-node cycles in which each node participates for any given graph. This equivalence holds because the input $\mathbf{x}$ of R-PEARL incorporates higher-order moments, which are directly related to the input $\mathbf{I}$ of B-PEARL as follows:
> > >
> > > $$\mathbb{E}\left[\mathbf{x}\circ\mathbf{x}\circ\dots\circ\mathbf{x}\right]=\mathbf{I}\circ\mathbf{I}\circ\dots\circ\mathbf{I},$$
> > >
> > > and the structural expressivity of R-PEARL depends on $\mathbb{E}\left[\mathbf{x}\circ\mathbf{x}\circ\dots\circ\mathbf{x}\right]$.
> > >
> > > **We have included this discussion in Remark 5.1. and Appendix G of  the revised manuscript.**
> > >
> > > > 2.  The rationale behind using Message Passing Neural Networks (MPNNs) to generate PEs is not clearly explained. **Proposition 3.1 appears to be trivial, as graphs (represented by adjacency matrices and node features) can be bijectively mapped to eigenvectors, eigenvalues, and node features.** Consequently, all graph functions, including PEs, can be seen as functions of eigenvectors when parameters include both eigenvalues and node feature. Therefore, the justification for choosing MPNNs as eigenvalue function over other graph models is not evident.
> > >
> > > > Authors agreed that Proposition 3.1 is trivial. MPNN may have many advantages, but Proposition 3.1 is not persuative and can be removed.
> > >
> > > We respect the reviewer's opinion, but kindly disagree that Proposition 3.1 is trivial to the graph ML community. The significance of Proposition 3.1 is underscored by the fact that our work is the first to suggest that eigenbased PEs can be computed directly using GNNs. Notably, several prior works augment GNNs with spectral information precisely because the established belief has been that message-passing GNNs struggle to compute such information. Examples include studies by  (Dwivedi & Bresson, 2021), (Rampášek et al., 2022), (Kreuzer et al., 2021), (Mialon et al., 2021), (He et al., 2023), (Kim et al., 2022), (Lim et al., 2023), (Huang et al. 2024), and (Feldman et al. 2022)
> > >
> > > Proposition 3.1 is integral to our work and serves three key purposes:
> > >
> > > 1.  It motivates the use of GNNs as a replacement for explicit spectral operations.
> > > 2.  It introduces readers to the spectral representation of message-passing GNNs.
> > > 3.  It forms the foundation for understanding and proving Theorem 3.1.
> > >
> > > Even if we were to assume, for the sake of argument, that Proposition 3.1 is trivial (a position with which we respectfully disagree), points 1-3 provide sufficient justification for its inclusion in our manuscript.

---

> > > > ### Author Response · Authors · 2024-11-29
> > > > **Detailed response (2/4)**
> > > >
> > > > >3.  Theorem 4.3 and the claim that "our proposed PE framework is well-suited for large-scale graphs" are questionable. The proof of Theorem 4.3, specifically Formula (45), includes the maximum degree of the graph, which is latter included in β. Considering β as a constant is problematic because large social networks often contain nodes with high degrees, which would require more samples for accurate representation.
> > > >
> > > > >Authors said by normalization, $\beta$ can be small for even graph with large node degree. But with normalization, both signal and the noise can be scaled to be small, so a smaller error bound \epsilon in Equation 9 is needed. A more systematic way is to compute signal to noise ratio.
> > > >
> > > >
> > > > This is a subtle and important point that requires additional attention. Our claim is twofold:
> > > >
> > > > 1.  **Message-passing with normalized GSOs is an appropriate choice for the $\Phi$ function in PEARL and does not omit critical information.**
> > > > 2.  **The sample complexity of R-PEARL is independent of graph size when $\Phi$ performs message-passing with normalized GSOs.**
> > > >
> > > > **Furthermore, we will show that the use of normalized GSOs can only have a positive impact on the signal-to-noise ratio as the graph size increases.**
> > > >
> > > > ----------
> > > >
> > > > ### **1. Normalized GSOs as Universal Graph Descriptors**
> > > >
> > > > We argue that PEs for graphs can be derived from normalized GSOs without compromising critical information. Normalized GSOs serve as universal descriptors for any graph, with the sole exception of not explicitly capturing node degrees. However, the degree of each node can be easily computed using a simple 1-layer, 1-feature linear GNN and thus can be safely omitted from the PE block to the backbone model or augmented as an additional feature. Consequently, the primary goal of the PE is to encode, as effectively as possible, the eigenvectors and eigenvalues of the normalized GSOs.
> > > >
> > > > ----------
> > > >
> > > > ### **2. Independence of Sample Complexity from Graph Size**
> > > >
> > > > We establish two key results regarding R-PEARL:
> > > >
> > > > -   **(i)** The bound $\beta$ on the norm of $\mathbf{H}(\mathbf{S}) = \sum_k h_k \mathbf{S}^k$ depends only on the trainable parameters of $\Phi$ and the eigenvalues of the GSO.
> > > > -   **(ii)** The variance bound of the R-PEARL output is independent of graph size.
> > > >
> > > > #### **(i) Bounding $\lVert \mathbf{H}(\mathbf{S}) \rVert$:**
> > > >
> > > > The norm $\lVert \mathbf{H}(\mathbf{S}) \rVert$ can be expressed as:
> > > >
> > > > $$\lVert\mathbf{H}\left(\mathbf{S}\right)\rVert=\lVert\sum_{k}h_k\mathbf{S}^k\rVert=\lVert\sum_{k}h_k\mathbf{\Lambda}^k\rVert=max_{\lambda_n}\lvert\sum_{k}h_k\mathbf{\lambda_n}^k\rvert=max_{\lambda_n}\lvert\tilde{h}(\lambda_n)\rvert\leq\beta$$
> > > >
> > > > Thus, $\beta$, the upper bound of $\lVert \mathbf{H}(\mathbf{S}) \rVert$, depends on the trainable weights $h_k$ and the eigenvalues of the GSO
> > > >
> > > > #### **(ii) Independence from Graph Size:**
> > > >
> > > > Consider a trained GNN that is applied on $\mathcal{G}_1$ and $\mathcal{G}_2$, where $\mathcal{G}_1$ is a small graph with a low maximum degree, and $\mathcal{G}_2$ is a large graph with high-degree nodes. Since the weights are fixed during execution they are independent of the graph size, i.e., same weights are applied to both $\mathcal{G}_1$ and $\mathcal{G}_2$. Then $\lVert \mathbf{H}(\mathbf{S}) \rVert$ depends only on the eigenvalues of the GSO. For adjacency or Laplacian GSOs, graphs of different sizes exhibit different eigenvalue spectrum range. However, when normalized GSOs are used, the eigenvalues are bounded within a fixed range, such as $[-1, 1]$ or $[0, 2]$, regardless of graph size. As a result $\lVert\mathbf{H}\left(\mathbf{S}\right)\rVert$ has the same bound $\beta$ across different graphs. For normalized adjacency matrices (similarly for normalized Laplacians or random-walk matrices), $\beta$ is given by:
> > > >
> > > > $$\beta=max_{-1\leq x\leq 1}\lvert\sum_{k}h_k x^k\rvert=max_{-1\leq x\leq 1}\lvert\tilde{h}(x)\rvert$$
> > > >
> > > > which is independent of the graph size, since it only depends on the trainable parameters.
> > > >
> > > > Following the analysis here and in Appendix F, it is evident that the variance of the R-PEARL output does not increase with larger graph sizes, and hence the sample complexity remains unaffected by graph size.

---

> > > > > ### Author Response · Authors · 2024-11-29
> > > > > **Detailed response (3/4)**
> > > > >
> > > > > ----------
> > > > >
> > > > > ### **Signal-to-Noise Ratio Analysis**
> > > > >
> > > > > The reviewer raises an insightful point regarding the impact of noise to signal with different energy, i.e., the signal-to-noise ratio (SNR). Specifically, a variance of $\beta^2$ has a different impact depending on whether the signal has a mean of $1$ or $10$. The SNR in the first case is $1/\beta^2$ and in the second case $100/\beta^2$. Our results would be less meaningful if the mean of the output decreased with graph size. However, this is not the case. The mean of the output can only increase with graph size, as we show next. As a result the same number of samples in R-PEARL can potentially give cleaner signals. To see this, we consider a GNN layer that employs single-layer MLPs with ReLU nonlinearity. Then,
> > > > >
> > > > > $$\mathbf{x_v}^{(l)} = \sigma\left(\mathbf{e_u}^T\sum_{k=0}^{K-1}\mathbf{S}^k\mathbf{X}^{(l)}\mathbf{H}_k^{(l)}\right)$$
> > > > >
> > > > > After some algebraic manipulations, the $f$-th feature of $\mathbf{x}_v^{(l)}$ can be expressed as:
> > > > >
> > > > > $$\mathbf{x_v}^{(l)}[f] = \sigma\left(\mathbf{z_v}^{(l)}[f]\right)= \sigma\left(\sum_{i=1}^{F_{l-1}}\mathbf{e_u}^T\sum_{k=0}^{K-1}\mathbf{H_k}^{(l)}[i,f]\mathbf{S}^k \mathbf{X}^{(l-1)}[:,i]\right)$$
> > > > >
> > > > > We aim to study the expected value $\mathbb{E}\left[\mathbf{x}_v^{(l)}[f]\right]$. First, we examine the effect of pointwise nonlinearity on the expected value of a random variable. Let $\mathbb{E}\left[\mathbf{z}_v^{(l)}[f]\right]$ be the expected value of $\mathbf{z}_v^{(l)}[f]$. Since $\sigma$ is convex,
> > > > >
> > > > >  $$\mathbb{E}\left[\mathbf{x_v}^{(l)}[f]\right]\geq\mathbb{E}\left[\mathbf{z_v}^{(l)}[f]\right],$$
> > > > > and the effect of the pointwise nonlinearity is independent of graph size. Now, we analyze the linear component of the GNN layer:
> > > > >
> > > > > $$\mathbb{E}\left[\mathbf{z_v}^{(l)}[f]\right]=\sum_{i=1}^{F_{l-1}}\mathbf{e_u}^T\sum_{k=0}^{K-1}\mathbf{H_k}^{(l)}[i,f]\mathbf{S}^k \mathbb{E}\left[\mathbf{X}^{(l-1)}[:,i]\right]$$
> > > > > Recall that the filter $\sum_{k=0}^{K-1}\mathbf{H}_k^{(l)}[i,f]\mathbf{S}^k$ does not depend on graph size but only on the eigenvalues of the GSO, which span the same space regardless of graph size. Now, we bound the expected value:
> > > > >
> > > > > $$|\mathbb{E}\left[\mathbf{z_v}^{(l)}[f]\right]|\leq\sum_{i=1}^{F_{l-1}}|\mathbf{e_u}^T\sum_{k=0}^{K-1}\mathbf{H_k}^{(l)}[i,f]\mathbf{S}^k \mathbb{E}\left[\mathbf{X}^{(l-1)}[:,i]\right]|=\sum_{i=1}^{F_{l-1}}|\langle\sum_{k=0}^{K-1}\mathbf{H_k}^{(l)}[i,f]\mathbf{S}^k\mathbf{e_v},\mathbb{E}\left[\mathbf{X}^{(l-1)}[:,i]\right]\rangle|$$
> > > > >
> > > > > $$\leq \sum_{i=1}^{F_{l-1}}\lVert\sum_{k=0}^{K-1}\mathbf{H_k}^{(l)}[i,f]\mathbf{S}^k\rVert\lVert\mathbf{e_v}\rVert\lVert\mathbb{E}\left[\mathbf{X}^{(l-1)}[:,i]\right]\rVert\leq\sum_{i=1}^{F_{l-1}}\beta\lVert\mathbb{E}\left[\mathbf{X}^{(l-1)}[:,i]\right]\rVert$$
> > > > >
> > > > > which simplifies to:
> > > > >
> > > > > $$\leq\beta\sum_{i=1}^{F_{l-1}}\sum_{n=1}^{N}\lvert\mathbb{E}\left[\mathbf{X}^{(l-1)}[n,i]\right]\rvert$$
> > > > >
> > > > > Then equation (72) of the revised manuscript the bound for the SNR for maximum possible noise variance is:
> > > > >
> > > > > $$SNR\leq\frac{\mathbb{E}\left[\mathbf{z_v}^{(l)}[f]\right]^2}{\beta^2F_{l-1}^2}\leq N^2 max_{n,i}\mathbb{E}\left[\mathbf{X}^{(l-1)}[n,i]\right] $$
> > > > >
> > > > > So the graph size can only have a positive effect on the SNR although this is not necessary.
> > > > >
> > > > > ----------
> > > > >
> > > > > In summary, our analysis shows that the sample complexity of R-PEARL remain unaffected by graph size, ensuring robustness and practical utility across a variety of graph scales.

---

> ### Author Response · Authors · 2024-11-29
> **Detailed response (4/4)**
>
> >Authors agreed that Proposition 4.1, 4.2, 4.3 are direct corollaries of previous conclusions. We still consider that Theorem 4.3 as a direct corollary, which only computes the Lipschitz constant of GNN and applies the conclusion of Lipschitz functions' sample complexity in the original proof. Currently, the proof the Theorem 4.3 is missing.
>
> **We would like to kindly point out that the proof of Theorem 4.3 is not missing**; it has been included in the Appendix in all versions of our manuscript. We respectfully note that it is unclear how the reviewer could evaluate our proof without being aware that it is already part of the manuscript.
>
> >The newly added Theorem (PEARL is a universal approximator of basis invariant functions) is also doubtful. In line 907-909 of the proof, basisnet need g with universal expressivity for universality of the whole model, but PEARL uses only MPNN.
>
> We also employ an equivariant pooling function without imposing any additional restrictions. For further details, we kindly refer the reviewer to line 208 of the first version of the revised manuscript: "design an equivariant pooling function $\rho$ to generate the final PE for each node". The only requirement for $\rho$ is that it be equivariant. We have updated the manuscript to clarify this point further: "To address this, we design an equivariant function $\rho$, that involves a pooling operation over the independent outputs $\{\mathbf{P}^{(m)}\}_{m=1}^M$, to generate the final PE for each node."
>
> In the proof we also mention:
>
> We can choose $\rho$ to operate on each feature independently i.e.,
>
> $$\mathbf{Y}=\rho\left(g^{(1)}\left(\mathbf{V_{\mu_1}}\mathbf{V_{\mu_1}}^T\right),\dots,g^{(q)}\left(\mathbf{V_{\mu_q}}\mathbf{V_{\mu_q}}^T\right)\right),$$
>
>
> where $g^{(i)}$ is universally approximating permutation equivariant or invariant function, e.g., high-order tensor IGN.
>
>
>
> >Moreover, the Theorem assume the model exists for each fixed set of eigenvalue, but in practice, we need one model to work for all eigenvalue sets as the input graph changes.
>
> **Universality is an existence claim, different than generalizability**. G can model any possible set of input graphs as disconnected components. Then the GSO will have a block diagonal form and each block will be the GSO of each graph. The rest of the proof is exactly the same. We add this note to the manuscript.
>
> >5.  The experimental section is limited to four datasets. To strengthen the validity of the results, additional datasets should be included as in [1].
>
> >Only one smallest synthetic dataset CSL in [1] is added. Commonly used datasets like long range graph benchmark and ogbg-molhiv in [1] are ignored. Please add them.
>
> We have added experiments on the peptide-structure dataset. Overall, **PEARL is tested across 10 (9 real-world, 1 synthetic)** diverse tasks spanning 5 distinct graph types. We firmly believe that this represents an objectively sufficient experimental setup for testing any graph ML framework. For your reference **[1] is tested on 7 real-world task, without any large graphs.**
>
> **Table: Test MAE on the Peptides-struct dataset.**
> B-PEARL achieves the second-best performance, behind Graph ViT, within a $500$k parameter budget. Graph ViT utilizes additional random walk information, while B-\method~learns this information via training.
>
> | Model          | R-PEARL         | B-PEARL          | GPS               | SAN+LapPE         | SAN+RWSE         | GNN-AK$+$         | SUN               | Graph ViT         |
> |----------------|--------------------|--------------------|--------------------|--------------------|------------------|------------------|------------------|------------------|
> | Test MAE       | $0.247_{\pm 0.001}$| $0.252_{\pm 0.002}$| $0.252_{\pm 0.001}$| $0.268_{\pm 0.004}$| $0.255_{\pm 0.001}$| $0.274_{\pm 0.00}$| $0.250_{\pm 0.001}$| $0.245_{\pm 0.002}$|
>
> >6.  The PE is used in conjunction with a GNN backbone, but the backbone for PEARL appears to be fixed across all experiments. To demonstrate the broad applicability and ensure a fair comparison among different PEs, experiments on PEs with different GNN backbones should be added, as recommended in [2].
>
> >Up to now, ablation study on GNN backbone is not added.
>
> These ablation study has now been added to the manuscript along with other ablation studies.
>
> **Table: Test MAE of PEARL with different backbones within a 500k parameter budget.**
>
> | **Model**      | **GINE**            | **GatedGCN**        | **PNA**            |
> |-----------------|---------------------|---------------------|---------------------|
> | B-PEARL  | $0.0679 \pm 0.0026$ | $0.0765 \pm 0.0018$ | $0.0740 \pm 0.0010$ |
> | R-PEARL  | $0.0721 \pm 0.0045$ | $0.0810 \pm 0.0039$ | $0.0946 \pm 0.0032$ |
>
> We conduct additional base model ablations, using PNA, GatedGCN and GINE on the ZINC dataset. PEARL demonstrates strong performance across various backbones, while consistently outperforming equivalent SignNet models with the same backbones.

---

> > ### Author Response · Authors · 2024-12-02
> > **We would like to hear your thoughts on our response and revised manuscript**
> >
> > Dear reviewer KVQZ,
> >
> > We sincerely thank you for your thoughtful feedback and for the time and effort you dedicated to your review.
> >
> > We believe we have addressed all of your comments comprehensively. Specifically:
> >
> > -   We demonstrated that the sample complexity of R-PEARL is independent of graph size when normalized GSOs are employed and, as requested, conducted the corresponding signal-to-noise ratio analysis.
> > -   We established the correctness of Theorem 3.1 and extended its generalization to multiple graphs.
> > -   Regarding the experiments, we included the requested ablation studies and comparisons using the peptide structure dataset. Our method has now been evaluated on 9 diverse real-world tasks, compared to the 7 real-world tasks evaluated in [1].
> >
> > We would be grateful if you could share your thoughts on our response and the revised manuscript. Please let us know if you have any additional questions or feedback—we would be happy to address them.

---

### Official Review · Reviewer_eV2C · 2024-11-03

**Soundness:** 2
**Presentation:** 3
**Contribution:** 3
**Rating:** 6
**Confidence:** 2

**Summary:**

Beforehand, I want to say that I am not familiar with this topic.

The paper introduces PEARL, a novel framework for learnable positional encodings (PEs) in graph representation learning. It addresses the limitations of eigenvector-based methods by leveraging message-passing Graph Neural Networks (GNNs) to compute efficient and expressive PEs. The authors propose initializing node attributes with random or basis vectors to ensure both expressiveness and permutation equivariance. The framework is validated through comprehensive theoretical analysis and empirical evaluation, showing significant improvements in scalability and performance compared to existing methods.

**Strengths:**

1. Innovative Approach: The use of message-passing GNNs as nonlinear mappings of eigenvectors is a novel contribution, enabling efficient computation of PEs.
2. Comprehensive Analysis: The paper rigorously proves the stability, expressiveness, and computational efficiency of PEARL.
3. Empirical Validation: Experimental results demonstrate that PEARL outperforms traditional methods on various graph classification and regression tasks while maintaining lower computational complexity.
4. Scalability: The framework is designed to handle large graphs efficiently, addressing a critical limitation of eigenvector-based methods.
5. Theoretical Contributions: The analysis of sample complexity and stability provides a solid foundation for the proposed approach.

**Weaknesses:**

1. Limited Discussion on Practical Implications: While the theoretical and empirical results are robust, the paper could benefit from a deeper discussion on the practical applications of PEARL in real-world scenarios.
2. Complexity of Understanding: The mathematical depth and complexity might pose a barrier to readers unfamiliar with the intricacies of GNNs and spectral graph theory.
3. Comparative Baselines: Although the paper includes several baselines, additional comparisons with more recent or diverse methods could strengthen the empirical validation.
4. Ablation Studies: While ablation studies are presented, further exploration of the impact of different initialization strategies and pooling functions on performance could provide more insights.

**Questions:**

The paper introduces PEARL as a scalable and efficient alternative to eigenvector-based positional encodings. While the theoretical advantages are well articulated, can the authors provide more detailed insights or case studies on how PEARL performs in real-world applications or large-scale industrial settings? Specifically, what types of graphs or domains would benefit the most from the proposed method, and are there any observed limitations in practical deployments?

---

> ### Author Response · Authors · 2024-11-25
> **Summary of our response**
>
> We thank the reviewer for appreciating our work and for their valuable feedback. The primary concerns raised by the reviewer are: (i) incorporating experiments on industrial tasks to evaluate our approach, and (ii) performing additional ablation studies.
>
> Regarding (i), we further emphasize the industrial applicability of our approach in real-world scenarios. To this end, we evaluate the performance of PEARL on large-scale recommendation tasks using the Stack Exchange Q&A Website Database. Specifically, we utilize the rel-stack dataset, a temporal and heterogeneous graph comprising approximately 38 million nodes. We consider two distinct industrial settings:
>
> - User-Post-Comment Prediction: Predicting a list of existing posts that a user will comment on within the next two years.
>
> - Post-Post-Related Prediction: Predicting a list of existing posts that users will link to a given post within the next two years.
> Notably, PEARL achieves state-of-the-art performance in this large-scale setting, further underscoring its scalability and effectiveness in practical applications.
>
> In response to (ii), we have incorporated additional ablation studies into the revised manuscript. Specifically, we analyze the effects of varying the number of layers, model sizes, and the choice of K in the first layer. Furthermore, we provide runtime comparisons to deliver a comprehensive evaluation of the method's performance and computational efficiency.

---

> > ### Author Response · Authors · 2024-11-25
> > **Response to Weaknesses**
> >
> > >Limited Discussion on Practical Implications: While the theoretical and empirical results are robust, the paper could benefit from a deeper discussion on the practical applications of PEARL in real-world scenarios.
> >
> > We thank the reviewer for their valuable feedback. In response, we have expanded the discussion on the practical applications of PEARL in the revised manuscript. Specifically, PEARL is highly versatile and can be applied to molecular graphs as well as larger-scale graphs, such as social networks and relational databases.
> >
> > To further illustrate its practical utility, we have included additional experiments involving two recommendation tasks on industrial relational databases, using the Stack Exchange dataset. This represents a highly practical scenario involving a graph with 38 million nodes. Notably, the PEARL framework demonstrates state-of-the-art performance in this large-scale setting, further highlighting its scalability and effectiveness.
> >
> > >  Comparative Baselines: Although the paper includes several baselines, additional comparisons with more recent or diverse methods could strengthen the empirical validation.
> >
> > We thank the reviewer for their thoughtful comment. Our experiments include comparisons with 11 diverse baselines, covering random PEs, eigen-based PEs, and structural PEs. If there are additional relevant works that the reviewer believes should be included in our comparisons, we would greatly appreciate their recommendations and will consider incorporating them into our analysis.
> >
> > >  Ablation Studies: While ablation studies are presented, further exploration of the impact of different initialization strategies and pooling functions on performance could provide more insights.
> >
> > We thank the reviewer for their valuable comment. In response, we have included additional ablation studies in the revised manuscript. Specifically, we analyze the effect of varying the number of layers, model sizes, and the choice of K in the first layer. Additionally, we have conducted runtime comparisons to provide a comprehensive evaluation of the method's performance and efficiency.

---

> > > ### Author Response · Authors · 2024-11-25
> > > **Response to reviewer's questions**
> > >
> > > >While the theoretical advantages are well articulated, can the authors provide more detailed insights or case studies on how PEARL performs in real-world applications or large-scale industrial settings?
> > >
> > > To further empasize the practicability of our approach in real-world applications, we test the performance of the proposed PEARL on large-scale recommendation for Stack Exchange Q\&A Website Database. To that end we utilize the rel-stack dataset for the relational deep learning benchmak (RelBench) [2]. Rel-stack is a temporal and heterogeneous graph with approximately 38 million nodes. We consider two different industrial settings; i) user-post-comment, where we predict a list of existing posts that a user will comment in the next two years, and ii) post-post-related, where we predict a list of existing posts that users will link a given post to in the next two years. The results for the two tasks can be found in the following table.
> > >
> > > ### Table: Validation and Test Mean Average Precision (MAP) on Large-Scale RelBench Recommendation Tasks
> > >
> > > **PEARL** achieves an 11% improvement over the backbone model with no PE on the *user-post-comment* task and a 2% improvement on the *post-post-related** task.
> > >
> > > | **Task**                  | **Evaluation** | **No PE** | **SignNet-8L** | **SignNet-8S** | **B-PEARL (ours)** | **R-PEARL (ours)** |
> > > |---------------------------|----------------|-----------|----------------|----------------|----------------------|----------------------|
> > > | **User-post-comment**     | Val. MAP       | 15.20     | 15.33          | **15.47**      | 15.13               | 15.24               |
> > > |                           | Test MAP       | 12.47     | 13.76          | 13.77          | 13.80               | **13.87**           |
> > > | **Post-post-related**     | Val. MAP       | 8.10      | 7.90           | 7.70           | 8.00                | **8.40**            |
> > > |                           | Test MAP       | 10.73     | 10.39          | 10.86          | **10.94**           | 10.86               |
> > >
> > >
> > > The backbone model for this RelBench task a heterogeneous identity-aware GNN and all methods are trained with batch size $20$. From the above table we observe that PEARL has an 11\% benefit over the identity aware backbone model with no PE on the user-post-comment task and a 2\% benefit on the post-post-related task. PEARL works similarly to SignNet-8S but with lower complexity and, and similarly to SignNet-8L on the user-post-comment task, but 5\% better than SignNet-8L on the post-post-related task. SPE could not be included in the comparisons due to it's cubic computational complexity and quadratic memory complexity.
> > >
> > > [2] Robinson, J., Ranjan, R., Hu, W., Huang, K., Han, J., Dobles, A., Fey, M., Lenssen, J.E., Yuan, Y., Zhang, Z. and He, X., RelBench: A Benchmark for Deep Learning on Relational Databases. In  The Thirty-eight Conference on Neural Information Processing Systems Datasets and Benchmarks Track.
> > >
> > > >Specifically, what types of graphs or domains would benefit the most from the proposed method, and are there any observed limitations in practical deployments?
> > >
> > > The proposed method is generic and can be applied to any type of graph or domain. It is also scalable and can be applied to large-scale graphs. We believe it is particularly effective for graphs with rich structural features and large datasets. This is because our approach enables the model to learn directly from the data and identify critical substructures without introducing additional biases.
> > >
> > > However, a potential limitation of our method is that it is not explicitly tailored to specific graph types, such as heterogeneous graphs or knowledge graphs. As a result, specialized positional encodings that exploit the regularities and unique characteristics of these graph types could potentially enhance the performance of PEARL in such domains.

---

> > > > ### Comment · Reviewer_eV2C · 2024-11-26
> > > > **Official comment by eV2C**
> > > >
> > > > Thanks the authors for the reply. As I am not professional on this topic, I'll just leave it to other reviewers.

---

### Official Review · Reviewer_BGAo · 2024-11-03

**Soundness:** 3
**Presentation:** 3
**Contribution:** 3
**Rating:** 6
**Confidence:** 4

**Summary:**

This work aims to propose a new framework for graph positional encoding that is expressive, scalable, stable and generalizable. The proposed framework generates positional encoding as aggregated outputs of GNNs with random or basis inputs, which is then fed into downstream GNNs for performance evaluation. The proposed framework is accompanied with theoretical arguments and empirical verification on graph regression and classification tasks.

**Strengths:**

1. Good motivations: the topic of graph positional encoding is a significant direction, which requires sufficiency, efficiency and easiness for generalization.
2. The proposed method is easy for plug-in: it proposes positional encodings as aggregation of GNNs' outputs with random/basis inputs, which are quite simple with standard implementation.
3. The empirical performance of the proposed method is comparable with SPE methods in Table 2, but it is much more efficient than SPE in Table 1 for larger graphs.
4. Different choices of input with random or basis are good with a consideration of different scales of problems, with the sample size of random inputs examined in Figure 2.

**Weaknesses:**

Major:
1. A key assumption is missing in proposition 3.1: GSO $S$ is required to be symmetric if we want to use the decomposition in line 105. So  the random walk matrix in Eq(2) does not satisfy this condition.
2. in line 463, it might be a little sudden to claim ''SPE can be efficiently implemented by B-PEARL with lower computational and memory complexity'': the argument seems coming from the comparison of numbers in Table 2. Although the numbers of SPE and the proposed methods are close, it would be better to conduct more careful analysis to provide such a ''subset'' or ''equivalence'' relationship between the methods, or refer to remark 5.1 for theoretical insights.
3. in line 60 of introduction, the paper argues '' stability is particularly crucial for out-of-distribution generalization'', which is one of the main motivations for this work. However, such a benefit of size generalization is not provided with enough evidence in this work. Actually in Table 3, the OOD-size performance is not improved by stable methods like SPE and the proposed R-PEARL.
4. in line 397, a 9-layer MPNN is used for experiments, which is relatively deep in the practice of GNNs. Could you please reveal how depth will impact the empirical results? Meanwhile, such a deep method with random inputs is quite similar to power iterations for eigenvalues, so some discussion between these might be helpful.

Minor:
1. in line 133-134, the definition of $F_{l-1}$ is not clearly stated, and it should be $F_l$ for $X^{(l)}$.
2. in line 293, it seems to be ''a consequence of Prop 4.1'' instead of 4.2

**Questions:**

Please see the above major concerns.

---

> ### Author Response · Authors · 2024-11-25
> **Summary of our response**
>
> We sincerely thank the reviewer for their thoughtful feedback and for recognizing the contributions of our work. The reviewer raises two primary concerns: (i) the out-of-distribution (OOD) generalization performance of our method compared to less stable methods, and (ii) the impact of the number of layers on performance.
>
> To address (i), we conducted additional experiments with B-PEARL and compared its OOD-Test AUC to that of less stable methods such as SignNet and BasisNet. The results show that B-PEARL improves OOD-Test AUC by **9.5%** over BasisNet and **3.8%** over SignNet in size OOD generalization. Similarly, B-PEARL achieves a **4.6%** improvement in OOD-Test AUC over both SignNet and BasisNet in Scaffold OOD generalization.
>
> To address (ii), we performed an ablation study by varying the number of layers in B-PEARL. Our findings indicate that while B-PEARL is robust to changes in the number of layers, and its performance benefits from the use of multiple layers compared to fewer layers.

---

> ### Author Response · Authors · 2024-11-25
> **Response to Weaknesses (1/3)**
>
> >  A key assumption is missing in proposition 3.1: GSO  S  is required to be symmetric if we want to use the decomposition in line 105. So the random walk matrix in Eq(2) does not satisfy this condition.
>
> We thank the reviewer for catching this. We have adjusted Proposition 3.1 to include non-symmetric GSO. Proposition 3.1 now reads:
>
> ### Proposition: GNNs are Nonlinear Functions of Eigenvectors
> A GNN defined in Eq. (1) with $f^{(l)}$ being one of the functions in Eq. (2) and $g^{(l)}$ being an MLP, is a nonlinear function of the GSO eigenvectors, i.e., $\mathbf{X}^{(l)} = \texttt{MLP}\left(\mathbf{V}\right)$. The trainable parameters of the first MLP layer are not independent but depend on the eigenvalues {$\lambda_n$}$_{n=1}^N$ and eigenvectors $\mathbf{v_n}$, $n=1,\dots,N$ of the GSO, as well as the initial node features $\mathbf{X}$ of the graph:
>
> $\mathbf{X}^{(l)} = \texttt{MLP}\left(\mathbf{V}\right) = \texttt{MLP}^{(-1)}\left(\sigma\left(\mathbf{V} \mathbf{W}\right)\right)$,
>
> $\mathbf{W}\left[n, f\right] = \sum_{i=1}^{F_{l-1}}\sum_{k=0}^{K-1}\lambda_n^k\mathbf{H}_k^{(l)}[i,f]\langle \mathbf{\alpha}_n, \mathbf{X}^{(l-1)}\left[:,i\right]\rangle$,
>
> where $\mathbf{\alpha}_n = \mathbf{v}_n$ when the GSO is symmetric, and $\mathbf{\alpha}_n = \mathbf{V}^{-1}[n,:]$ when it is not. $\texttt{MLP}^{(-1)}$ denotes all the layers of the $\texttt{MLP}$ except the first layer.
>
>
> >  in line 463, it might be a little sudden to claim ''SPE can be efficiently implemented by B-PEARL with lower computational and memory complexity'': the argument seems coming from the comparison of numbers in Table 2. Although the numbers of SPE and the proposed methods are close, it would be better to conduct more careful analysis to provide such a ''subset'' or ''equivalence'' relationship between the methods, or refer to remark 5.1 for theoretical insights.
>
> We thank the reviewer for their comment. We agree and have now removed this comment from the revised manuscript.

---

> ### Author Response · Authors · 2024-11-25
> **Response to Weaknesses (2/3)**
>
> >  in line 60 of introduction, the paper argues '' stability is particularly crucial for out-of-distribution generalization'', which is one of the main motivations for this work. However, such a benefit of size generalization is not provided with enough evidence in this work. Actually in Table 3, the OOD-size performance is not improved by stable methods like SPE and the proposed R-PEARL.
>
> The reviewer is raising an interesting point. Stability is indeed important in out-of-distribution (OOD) generalization on Scaffold and size.and the benefit is clear in Table 3, especially when it .  We include Table 3 with additional experiments with B-PEARL to facilitate the discussion.
>
> The reviewer raises an very interesting point. The importance of stability for out-of-distribution (OOD) generalization is theoretically established, and the benefit of stable methods is shown in Table 3, particularly on Scaffold OOD and Size OOD. To facilitate this discussion, we have included Table 3, with additional experiments with B-PEARL that have been added in the revised manuscript.
>
> ### Table: AUROC Results (5 Random Seeds) on DrugOOD
>
> | **Domain**      | **PE Method**       | **ID-Val (AUC)**      | **ID-Test (AUC)**     | **OOD-Val (AUC)**     | **OOD-Test (AUC)**     |
> |------------------|---------------------|-----------------------|-----------------------|-----------------------|------------------------|
> | **Scaffold**    | No PE               | 96.56 ± 0.10          | 87.95 ± 0.20          | 79.07 ± 0.97          | 68.00 ± 0.60          |
> |                 | PEG                 | 95.65 ± 0.29          | 86.20 ± 0.14          | 79.17 ± 0.97          | 69.15 ± 0.75          |
> |                 | SignNet             | 95.48 ± 0.34          | 86.73 ± 0.56          | 77.81 ± 0.70          | 66.43 ± 1.06          |
> |                 | BasisNet            | 85.80 ± 3.75          | 78.44 ± 2.45          | 73.36 ± 1.44          | 66.32 ± 5.68          |
> |                 | SPE                 | 96.32 ± 0.28          | 88.12 ± 0.41          | 80.03 ± 0.58          | **69.64 ± 0.49**       |
> |                 | SPE-8S              | 96.44 ± 0.08          | 87.88 ± 0.45          | 79.34 ± 0.50          | 68.72 ± 0.63          |
> |                 | R-PEARL (ours)    | 96.09 ± 0.32          | 88.01 ± 0.43          | 78.72 ± 0.02          | 69.20 ± 1.00          |
> |                 | B-PEARL (ours)    | 96.06 ± 0.29          | 87.56 ± 0.81          | 79.86 ± 0.58          | 69.51 ± 0.62          |
> | **Size**        | No PE               | 93.78 ± 0.12          | 93.60 ± 0.27          | 82.76 ± 0.04          | **66.04 ± 0.70**       |
> |                 | PEG                 | 92.46 ± 0.35          | 92.67 ± 0.23          | 82.12 ± 0.49          | 66.01 ± 0.10          |
> |                 | SignNet             | 93.30 ± 0.43          | 93.20 ± 0.39          | 80.67 ± 0.50          | 64.03 ± 0.70          |
> |                 | BasisNet            | 86.04 ± 4.01          | 85.51 ± 4.04          | 75.97 ± 1.71          | 60.79 ± 3.19          |
> |                 | SPE                 | 92.46 ± 0.35          | 92.67 ± 0.23          | 82.12 ± 0.49          | 66.02 ± 0.49          |
> |                 | SPE-8S              | 93.68 ± 0.20          | 93.86 ± 0.12          | 83.04 ± 0.63          | 65.74 ± 2.20          |
> |                 | R-PEARL (ours)    | 93.32 ± 0.34          | 93.92 ± 0.20          | 82.09 ± 0.44          | 65.89 ± 1.30          |
> |                 | B-PEARL (ours)    | 93.18 ± 0.45          | 93.29 ± 0.46          | 83.14 ± 0.37          | **66.58 ± 0.67**       |
>
> The least stable methods in Table 3 are SignNet and BasisNet. We observe that B-PEARL improves the OOD-Test AUC by **9.5%** compared to BasisNet and **3.8%** compared to SignNet in size OOD generalization. Similarly, B-PEARL achieves a **4.6%** improvement in OOD-Test AUC over both SignNet and BasisNet in Scaffold OOD generalization.
>
> A comparable trend is observed when comparing the stable SPE with SignNet and BasisNet. Furthermore, B-PEARL demonstrates an advantage over SPE in size generalization. This improvement is likely attributed to the linear scaling of SPE's stability bound with graph size, compared to the square-root scaling of PEARL's stability bound, which enhances its size generalization capabilities.

---

> > ### Author Response · Authors · 2024-11-25
> > **Response to Weaknesses (3/3)**
> >
> > >  in line 397, a 9-layer MPNN is used for experiments, which is relatively deep in the practice of GNNs. Could you please reveal how depth will impact the empirical results?
> >
> > A multi-layer GNN is particularly effective in capturing complex structures and motifs that frequently appear in real-world networks, especially in the context of molecular graphs. For example, in molecular graphs such as those in the ZINC dataset, hexagons and pentagons are among the most common structural motifs. Additionally, combinations of these polygons with shared nodes play a significant role in determining molecular properties. GNNs with multiple layers are well-suited to model these intricate patterns with higher precision.
> >
> > To evaluate the impact of the number of layers on performance, we present an ablation study on the ZINC dataset using B-PEARL with varying numbers of layers:
> >
> > | **# GIN Layers**      | 3                 | 5                 | 7                 | 9                 |
> > |------------------------|-------------------|-------------------|-------------------|-------------------|
> > | **Mean Test Loss**     | $0.0701_{\pm 0.005}$ | $0.067_{\pm 0.003}$  | $0.069_{\pm 0.001}$  | $0.0644_{\pm 0.001}$ |
> >
> > **Table**: Ablation on the number of GIN layers in B-\method for the ZINC Dataset.
> >
> > We observe that the performance B-PEARL is pretty robust with varying layer number, but also benefits from multiple layers.
> >
> > > Meanwhile, such a deep method with random inputs is quite similar to power iterations for eigenvalues, so some discussion between these might be helpful.
> >
> > This is indeed a very interesting discussion, as there are notable similarities with power iterations, but also two key differences:
> >
> > 1.  **Processing the Spectrum**: Power iterations compute one eigenvector at a time, while our proposed approach processes the entire or a subset of the spectrum simultaneously. In this regard, our method is more closely related to block Krylov subspace methods than to traditional power iterations.
> >
> > 2.  **Convergence Dependence**: The convergence of power iterations is determined by the number of iterations (analogous to the number of layers in a GNN), whereas the convergence of our method depends on the number of independent samples. In essence, power iteration convergence benefits from increased depth, whereas the convergence of our method is independent of depth and instead improves with larger sample sizes.
> >
> > > Minor:
> >
> > We thank the reviewer for catching these typos which are fixed in the revised manuscript.

---

> > > ### Author Response · Authors · 2024-12-02
> > > **We would appreciate your feedback in our response and revised manuscript**
> > >
> > > Dear reviewer BGAo,
> > >
> > > We sincerely thank you for your valuable feedback and appreciation of our work.
> > >
> > > In the revised manuscript, we have updated Proposition 3.2 to include non-symmetric GSOs. We have also addressed your concerns regarding the OOD generalization performance of our method by conducting new experiments, which demonstrate the superiority of PEARL over less stable methods. Furthermore, we performed an ablation study by varying the number of layers in B-PEARL to provide additional insights.
> > >
> > > As a result, we would be very glad if you could share your thoughts on our response and revised manuscript. We would be happy to answer any further questions you may have.

---

### Official Review · Reviewer_J67b · 2024-11-04

**Soundness:** 2
**Presentation:** 2
**Contribution:** 1
**Rating:** 3
**Confidence:** 4

**Summary:**

The authors propose learning positional encodings by using spectral graph filters in the initial layers. They argue that this approach corresponds to learning non-linear functions of the eigenvectors. The spectral GNN is applied to the input graph without using node features; instead, random features are added to the nodes. Although this breaks permutation equivariance, the authors argue that using a sufficient number of samples and specific aggregation functions preserves permutation equivariance. The authors argue that this method is expressive, scalable and stable.

**Strengths:**

- Most other methods based on eigenvectors require computing the eigenbasis, leading to problems on large graphs. In contrast, the authors argue for a worst-case $N^2$ forward complexity without high preprocessing-complexity.
- The experimental results appear promising.

**Weaknesses:**

- The paper is not well-integrated into the existing literature. It is unclear what advantages it offers compared to other methods that count cycles. Although related work is mentioned, the authors do not clearly differentiate their approach or highlight their novel contributions.
- The theoretical results often seem incorrect or lack necessary details. Additionally, many of the results appear to be summaries of theoretical findings from related works. Specific examples are provided in the questions section.
- While the method is more expressive than the 1-WL test, this is not surprising. You can basically do anything to increase beyond 1-WL. For instance, simply adding the number of connected components as a feature would make the model more expressive than 1-WL. It would be beneficial to include quantitative results, such as whether the method is weaker than the 2-FWL test, and to compare it with other methods that have proven counting properties for example.
- A more significant issue is that, although the method is more expressive, it also breaks permutation symmetry. It is well known that adding unique node identifiers can make a model universal but breaks permutation symmetry.
- In Section 5, it is unclear what is meant by having a basis as an input. Additionally, Remark 5.1 is incorrect. In Huang et al., the $\alpha_i$ are always analytic and are sequence-to-sequence mappings, whereas in this work, they are always real-valued mappings that operate on each eigenvector independently. This makes the approach less expressive, at least in the frequency domain.

**Questions:**

- Equation (3) does not accurately represent all possible MPNNs, even when restricting $g$ to be an MLP and $f$ to result from multiplication with a GSO. Specifically, using an MLP from $\mathbb{R}^k$ to $\mathbb{R}^k$ does not trivially extend to a graph signal of dimension $\mathbb{R}^{n \times h}$. It seems that the authors are actually performing matrix multiplications from the right with learnable matrices, and using more than one. Similarly, in Proposition 3.1, the representation in Equation (12) is assumed before the proof begins, which is actually what the authors aim to prove.
- In Proposition 4.3, what is the variable $m$ summing over? The entire proposition lacks clarity, and it appears there is no novelty compared to the existing results from Gama et al.
- The authors claim that Equation (6) is permutation equivariant, which does not seem to be correct. It is only equivariant in expectation, similar to the random positional encodings introduced by Sato et al. (2020).
- What are the contributions to the stability and expressivity analysis? It seems that older results are merely being reused.
- Did the authors follow the standard experimental setup? For example, in the ZINC dataset, it is customary to use a parameter budget of 500K parameters.
- The authors argued that their method is more scalable. Could you a) test on larger graphs, and b) measure the improved speed?

## Additional Comment:

The primary novelty appears to be the use of random features as input for learning positional encodings, instead of typically using eigenvectors. It appears that PEARL is a specific variant of SPE where the equivariant sequence-to-sequence function is replaced by a point-wise function applied to eigenvalues. This point-wise function is simply a learnable polynomial, offering the clear advantage of not requiring eigen decomposition, thus making it faster and more memory-efficient. However, it is unclear whether this approach improves or decreases expressivity or leads to better stability or generalization bounds. Notably, the out-of-distribution (OOD) generalization in the experiments appears to be better for the SPE positional encodings. However, it is not clear why since SPEs are also stable. Finally, I want to direct the authors to "Spatio-Spectral Graph Neural Networks" by Geisler et al., which also uses spectral and spatial layers in GNNs.

Recommendation:
I recommend rejecting the paper in its current form and suggest that the authors improve the clarity of the manuscript (in terms of novely and also writing) and better integrate it into the existing literature.

---

> ### Author Response · Authors · 2024-11-24
> **Summary of our response (1/2)**
>
> We thank the reviewer for their valuable input. The main concerns of the reviewer can be summarized into the following points:
>
> 1.  The reviewer argues that PEARL violates **permution equivariance**, since it uses unique identifiers to initialize the node attributes.
> 2.  The reviewer claims that our **expressivity** results are not surprising, since one can basically do anything to increase expressivity beyond 1-WL, e.g., adding the number of connected components. The reviewer also argues that our work is not as expressive as SPE in Huang et al. ?
> 3.  The reviewer is questioning the contribution of our work in terms of **stability** analysis, and wonders how it compares the stability properties of other methods.
> 4.  The reviewer is asking for additional experiments with larger graphs to showcase the **scalability** of our approach.
>
> **Our short answers are as follows:**
>
> **A1:**
> The reviewer's arguments may stem from a misunderstandings regarding a key component of our architecture. We kindly draw the reviewer's attention to the pooling function $\rho$ in Equations (5), (6), and (10), which might have been missed. The pooling function $\rho$ ensures that our model remains permutation equivariant, even while utilizing unique node identifiers. **Given the reviewer's concern about this aspect, we hope our explanation clarifies the matter and encourages the reviewer to reassess their evaluation in light of this information.**
>
> **A2:**
>
> The reviewer is correct that surpassing the limitations of 1-WL is conceptually straightforward. As mentioned, this can be achieved, for example, by simply adding the number of connected components as a feature. However, this observation highlights a critical point: to surpass 1-WL expressivity, most approaches rely on a **non-neural algorithm** to provide additional knowledge to the GNN, e.g., count the number of connected components and add them to the GNN as extra features.
>
> In contrast, our work does not introduce any additional bias or external features. We enhance the expressivity of **equivariant message-passing GNNs** without relying on extra features or complex operations. This represents a significant departure from prior approaches, as we achieve this increased expressivity solely within the framework of a message-passing GNN architecture. This is important, since we can achieve increased expressivity while maintaining **genericness, low computational complexity and stability.**
>
> In the revised manuscript we prove that PEARL is a universal approximator of basis invariant functions.
>
> **Theorem [Basis Universality]**
> *Let* $\mathcal{G}$ *be a graph with GSO* $\mathbf{S} = \mathbf{V} \mathbf{\Lambda} \mathbf{V}^T$, *and* $f$ *be a continuous function such that* $f(\mathbf{V}) = f(\mathbf{V} \mathbf{Q})$, $\mathbf{Q} \in \mathcal{O}\left({diag}\left(\mathbf{\Lambda}\right)\right)$, *for any eigenvalues* $\mathbf{\Lambda}$. *Then there exist GNN* $\Phi$ *and a continuous pooling function* $\rho$, *such that*:
>
> $$
> f(\mathbf{V}) = \rho\left[ \Phi\left( \mathcal{G}, \mathbf{q}^{(1)} \right), \dots, \Phi\left( \mathcal{G}, \mathbf{q}^{(M)} \right) \right].
> $$
>
> **The above Theorem adds novelty to the expressivity analysis of our work and shows that PEARL is as expressive as SPE in Huang et al. in the frequency domain.**
>
> **A3:**
> The reviewers point is based on the fact that the stability properties of PEARL are directly inhereted by the stability of GNNs. While the stability properties are derived as corollaries of GNN stability, we do not see why this should be considered a limitation.
> Our contribution does not merely analyzes the stability an existing architecture but instead introduces a novel PE framework. Since our architecture is a GNN, it is unavoidable to inherit the stability properties from existing work. However, these foundational results reinforce the theoretical validity of our framework and enable us to extend and apply them in innovative ways. A direct comparison of the stability bounds of our approach with those of SPE further highlights the advantages of our framework. Specifically:
> - The stability bound of SPE scales linearly with the graph size.
> - In contrast, the stability bound of PEARL scales only with the square root of the graph size.
>
> This significant difference provides PEARL with a substantial stability advantage over SPE, especially for larger graphs.

---

> > ### Author Response · Authors · 2024-11-24
> > **Summary of our response (2/2)**
> >
> > **A4:**
> > In the submitted manuscript we showcase the scalability of our approach with the two REDDIT datasets. Notably SPE runs out-of-memory, while PERL achieves the best performance. To further empasize the scalability of our approach we test the performance of the proposed PEARL on large-scale link prediction for Stack Exchange Q\&A Website Database. To that end we utilize the rel-stack dataset for the relational deep learning benchmak (RelBench) [2]. Rel-stack is a temporal and heterogeneous graph with approximately 38 million nodes. We consider two different tasks; i) user-post-comment, where we predict a list of existing posts that a user will comment in the next two years, and ii) post-post-related, where we predict a list of existing posts that users will link a given post to in the next two years. The results for the two tasks can be found in the following table.
> >
> > ### Table: Validation and Test Mean Average Precision (MAP) on Large-Scale RelBench Recommendation Tasks
> >
> > **PEARL** achieves an 11% improvement over the backbone model with no PE on the *user-post-comment* task and a 2% improvement on the *post-post-related** task.
> >
> > | **Task**                  | **Evaluation** | **No PE** | **SignNet-8L** | **SignNet-8S** | **B-PEARL (ours)** | **R-PEARL (ours)** |
> > |---------------------------|----------------|-----------|----------------|----------------|----------------------|----------------------|
> > | **User-post-comment**     | Val. MAP       | 15.20     | 15.33          | **15.47**      | 15.13               | 15.24               |
> > |                           | Test MAP       | 12.47     | 13.76          | 13.77          | 13.80               | **13.87**           |
> > | **Post-post-related**     | Val. MAP       | 8.10      | 7.90           | 7.70           | 8.00                | **8.40**            |
> > |                           | Test MAP       | 10.73     | 10.39          | 10.86          | **10.94**           | 10.86               |
> >
> >
> > The backbone model for this RelBench task a heterogeneous identity-aware GNN and all methods are trained with batch size $20$. From the above table we observe that PEARL has an 11\% benefit over the identity aware backbone model with no PE on the user-post-comment task and a 2\% benefit on the post-post-related task. PEARL works similarly to SignNet-8S but with lower complexity and, and similarly to SignNet-8L on the user-post-comment task, but 5\% better than SignNet-8L on the post-post-related task. SPE could not be included in the comparisons due to it's cubic computational complexity and quadratic memory complexity.
> >
> > [2] Robinson, J., Ranjan, R., Hu, W., Huang, K., Han, J., Dobles, A., Fey, M., Lenssen, J.E., Yuan, Y., Zhang, Z. and He, X., RelBench: A Benchmark for Deep Learning on Relational Databases. In  The Thirty-eight Conference on Neural Information Processing Systems Datasets and Benchmarks Track.
> >
> > **Detailed responses to these points are provided below.**

---

> ### Author Response · Authors · 2024-11-24
> **Response to Weaknesses (1/3)**
>
> >  The paper is not well-integrated into the existing literature. It is unclear what advantages it offers compared to other methods that count cycles. Although related work is mentioned, the authors do not clearly differentiate their approach or highlight their novel contributions.
>
> We thank the reviewer for their comment. Regarding methods that count cycles the contribution of our work is fourfold:
> 1. **Performance:** Our work appears to outperform methods than explicitly count cycles even in molecular datasets where it is well known that 5-node and 6-node cycles are the most important ones
> 2. **Expressivity:** PEARL is not limited to pre-defined motifs, such as cycles or cliques. It can compute other potentially important substructures, such as dense subgraphs, chordal cycles, or combinations of motifs, that explicit counting methods might omit simply because they are not pre-specified. Notably, the number of possible motifs in a graph grows combinatorially, highlighting the flexibility and breadth of PEARL.
> 3. **Complexity:** Explicitly counting high-order motifs, especially at the node level, can be computationally expensive. PEARL bypasses this challenge by learning to capture these structures implicitly, making it more scalable to large and complex graphs.
> 4. **Bias:** Predetermining which motifs to count introduces bias into the model. For example, molecular graphs often benefit from detecting cycles, while social networks emphasize cliques or dense subgraphs. PEARL is task-agnostic and allows the data to guide which motifs are most relevant, adapting to the specific requirements of the application. On the flip side, when the training data have small sizes, learning can benefit by specific biases that structural PEs admit.
>
> **We add this discussion and clarify our contribution in the revised manuscript.**
>
>
> >The theoretical results often seem incorrect or lack necessary details.
>
> As we explain in our upcoming answers none of our results are incorrect and there might be a misunderstanding from the reviewer's side. We kindly ask the reviewer to re-assess their evaluation which we believe is due to a misunderstanding. We thank the reviewer for pinpointing some details which could be useful to be present in the theorems. We have added them in the revised manuscript
>
> >Additionally, many of the results appear to be summaries of theoretical findings from related works.
>
> While the stability properties and some expressivity results are derived as corollaries of related work, we do not see why this should be considered a limitation. Our contribution does not merely analyzes the stability or expressivity of an existing architecture but instead introduces a novel PE framework. Since our architecture is a GNN, it is unavoidable to inherit some properties from existing work. However, these foundational results reinforce the theoretical validity of our framework and enable us to extend and apply them in innovative ways.
>
> **We would also like to highlight two key theoretical contributions of our work: our sample complexity analysis, which was omitted from this discussion, and the newly included Basis Universality Theorem, which has been added to the revised version.**

---

> > ### Author Response · Authors · 2024-11-24
> > **Response to Weaknesses (2/3)**
> >
> > >  While the method is more expressive than the 1-WL test, this is not surprising. You can basically do anything to increase beyond 1-WL. For instance, simply adding the number of connected components as a feature would make the model more expressive than 1-WL. It would be beneficial to include quantitative results, such as whether the method is weaker than the 2-FWL test, and to compare it with other methods that have proven counting properties for example.
> >
> > The reviewer is correct that surpassing the limitations of 1-WL is conceptually straightforward. As mentioned, this can be achieved, for example, by simply adding the number of connected components as a feature. However, this observation highlights a critical point: to surpass 1-WL expressivity, most approaches rely on a **non-neural algorithm** to provide additional knowledge to the GNN, e.g., count the number of connected components and add them to the GNN as extra features.
> >
> > In contrast, our work does not introduce any additional bias or external features. We enhance the expressivity of **equivariant message-passing GNNs** without relying on extra features or complex operations. This represents a significant departure from prior approaches, as we achieve this increased expressivity solely within the framework of a message-passing GNN architecture. This is important, since we can achieve increased expressivity while maintaining genericness, low computational complexity and stability.
> >
> > PEARL is not inherently weaker than the 2-FWL test. By drawing unique identifiers from a structurally aware distribution—such as one that accounts for the positions of triangles within the graph—PEARL can be shown to count substructures like 8-node cycles and 4-node cliques, which the 2-FWL test cannot. However, in our experiments, we did not find this addition to be effective, and thus it lies beyond the scope of this paper.
> >
> > To further clarify the capabilities of the proposed PEARL framework, we also prove that PEARL is a universal approximator of basis-invariant functions. This theoretical result has been included as an additional quantitative contribution in the revised manuscript, further solidifying the expressivity of our approach.
> >
> > >  A more significant issue is that, although the method is more expressive, it also breaks permutation symmetry. It is well known that adding unique node identifiers can make a model universal but breaks permutation symmetry.
> >
> > We kindly draw the reviewer's attention to the pooling function $\rho$ in Equations (5), (6), and (10). The pooling function $\rho$ ensures that our model remains permutation equivariant, even while utilizing unique node identifiers. **Given the reviewer's concern about this aspect, we hope our explanation clarifies the matter and encourages the reviewer to reassess their evaluation in light of this information.**
> >
> > >  In Section 5, it is unclear what is meant by having a basis as an input.
> >
> > We thank the reviewer for this comment. We actually mean standard basis vectors. We have now rephrased the line 338 to read as:
> >
> > In these cases, we propose using standard basis vectors {$\mathbf{e_m}$} $_{m=1}^{N}$ as the initial node attributes, where $\mathbf{e}_m [m]=1$ and $\mathbf{e}_m [i\neq m]=0$

---

> > > ### Author Response · Authors · 2024-11-24
> > > **Response to Weaknesses (3/3)**
> > >
> > > > Additionally, Remark 5.1 is incorrect. In Huang et al., the  $\alpha_i$  are always analytic and are sequence-to-sequence mappings, whereas in this work, they are always real-valued mappings that operate on each eigenvector independently. This makes the approach less expressive, at least in the frequency domain.
> > >
> > > **In the revised manuscript we prove that PEARL is a universal approximator of basis invariant functions, which makes it as expressive as SPE in Huang et al. in the frequency domain.**
> > >
> > > Regarding the details of Remark 5.1, we respectfully disagree with the reviewer's interpretation. In fact, the situation is quite the opposite. In Huang et al., the functions $\alpha_i: \mathbf{R}^d \to \mathbf{R}^d$ are **Lipschitz continuous** and not necessarily differentiable, and therefore not necessarily analytic (**see Eq. 2 and Assumption 3.1 in Huang et al.**), where \(d\) denotes the number of eigenvalues. The SPE method in Huang et al. is both expressive and stable when $d = N$.
> > >
> > > To prove basis universality, the $\alpha_i$ functions are assumed to be **elementwise real-valued functions** (see Eq. 88 in Huang et al.). The only requirement for this proof is that the $\alpha_i$ functions can discriminate eigenvalues, a condition that can be satisfied by polynomials as well. We demonstrate this explicitly in our new universality proof.
> > >
> > > Furthermore, the implementation of SPE in Huang et al. selects $\alpha_i$ to be either elementwise MLPs or DeepSets functions, which involve elementwise operations combined with summation or max pooling. It is straightforward to show that PEARL can compute both the sum and the maximum of eigenvalues. However, such computations are unnecessary since elementwise MLPs are **at least as expressive as DeepSets**.
> > >
> > > As a final note regarding the reviewer's comment, we want to emphasize that we do not explicitly apply any \(\alpha_i\) or real-valued mappings to either the eigenvalues or the eigenvectors. **We do not compute eigenvalues or eigenvectors**. Instead, we rely on a message-passing GNN where $f^{(l)}$ performs summation and $g^{(l)}$ is an MLP.  According to Proposition 3.1, this design **implicitly acts as a set of analytic functions on the eigenvalues**, which are then further processed by subsequent MLP layers. This approach achieves the desired expressivity without explicitly handling eigenvalues or eigenvectors.

---

> > > > ### Author Response · Authors · 2024-11-24
> > > > **Response to Questions (1/2)**
> > > >
> > > > >   Equation (3) does not accurately represent all possible MPNNs, even when restricting  g  to be an MLP and  f  to result from multiplication with a GSO. Specifically, using an MLP from  $R^k$  to  $R^k$  does not trivially extend to a graph signal of dimension  $R^{n×h}$. It seems that the authors are actually performing matrix multiplications from the right with learnable matrices, and using more than one.
> > > >
> > > > The reviewer is correct that Equation (3) does not represent all possible MPNNs, as we explicitly state in Equation (2) and the accompanying discussion. However, Equation (3) does represent the theoretically most powerful GNN when g is an MLP and f is summation, as shown in [1].
> > > >
> > > > It is important to note that Equation (3) is used solely for theoretical analysis. We never instantiate adjacency powers in our implementation. Instead, we exclusively perform standard message-passing GNN operations, ensuring computational efficiency and simplicity.
> > > >
> > > > [1] Xu, K., Hu, W., Leskovec, J. and Jegelka, S., 2018, September. How Powerful are Graph Neural Networks?. In International Conference on Learning Representations.
> > > >
> > > > > Similarly, in Proposition 3.1, the representation in Equation (12) is assumed before the proof begins, which is actually what the authors aim to prove.
> > > >
> > > >
> > > > The reviewer is correct that Equation (12) is assumed before the proof begins. Equation (12) represents a GNN defined in Eq. (1), where $f^{(l)}$ is one of the functions described in Eq. (2), and $g^{(l)}$ is an MLP, as explicitly stated in the first sentence of Proposition 3.1.
> > > >
> > > > However, our goal is not to prove Equation (12) itself. Instead, we aim to demonstrate that standard message-passing GNNs inherently compute nonlinear functions of eigenvectors and eigenvalues, as described by the equations within the proposition. This insight highlights a critical aspect of GNN theory that is often overlooked by the broader GNN community.
> > > >
> > > > >  In Proposition 4.3, what is the variable  m  summing over? The entire proposition lacks clarity, and it appears there is no novelty compared to the existing results from Gama et al.
> > > >
> > > > The variable $m$ sums over the number of samples, as defined in Equations (5), (6), and (8). We have added this clarification to the revised manuscript for completeness.
> > > >
> > > > The results of Proposition 4.3 are indeed derived from the stability analysis of GNNs presented in Gama et al. This reliance is unavoidable, as our method builds upon message-passing GNNs, which have been proven stable in multiple prior works. However, our contribution is not a new stability bound. Instead, we present a novel PE framework that is provably jointly stable, expressive, generic, and generalizable**. To the best of our knowledge, PEARL is the first PE framework to make such claims. While the stability analysis itself is not original, it serves to support and reinforce the foundational claims of our framework.
> > > >
> > > > To better align with its role, we have renamed the proposition to Corollary in the revised manuscript.
> > > >
> > > > >   The authors claim that Equation (6) is permutation equivariant, which does not seem to be correct. It is only equivariant in expectation, similar to the random positional encodings introduced by Sato et al. (2020).
> > > >
> > > > Equation (6) is **permutation equivariant**, as it represents an **unbiased estimator of an expectation**. This distinguishes it from the random PEs introduced by Sato et al. (2020), which are only equivariant in expectation.
> > > >
> > > > The random PEs proposed by Sato et al. (2020) are **neither permutation equivariant nor unbiased estimators** of a permutation equivariant function. Being equivariant in expectation is a much weaker property, as it only guarantees that the expectation itself is equivariant—a trivial outcome since expectation is inherently an equivariant operation.

---

> > > > > ### Author Response · Authors · 2024-11-24
> > > > > **Response to Questions (2/2)**
> > > > >
> > > > > >   What are the contributions to the stability and expressivity analysis? It seems that older results are merely being reused.
> > > > >
> > > > > As highlighted in previous comments, our work does not merely analyze an existing architecture but instead introduces a novel PE framework. Developing a new framework requires a detailed description of its properties to ensure both clarity and rigor. While the stability and some expressivity properties are derived as corollaries of earlier results, we do not see why this should be considered a limitation. These foundational results reinforce the theoretical validity of our framework and enable us to extend and apply them in innovative ways.
> > > > >
> > > > > **Stability Comparison:**
> > > > > A direct comparison of the stability bounds of our approach with those of SPE further highlights the advantages of our framework. Specifically:
> > > > > - The stability bound of SPE scales linearly with the graph size.
> > > > > - In contrast, the stability bound of PEARL scales only with the square root of the graph size.
> > > > >
> > > > > This significant difference provides PEARL with a substantial stability advantage over SPE, especially for larger graphs.
> > > > >
> > > > > **Expressivity Analysis:**
> > > > > In the revised manuscript, we have added a novel expressivity analysis, demonstrating that PEARL is a universal approximator of basis-invariant functions. This new result further underscores the theoretical strength and versatility of our proposed framework.
> > > > >
> > > > >
> > > > > >   Did the authors follow the standard experimental setup? For example, in the ZINC dataset, it is customary to use a parameter budget of 500K parameters.
> > > > >
> > > > > We used the following parameter counts for our experiments:
> > > > >
> > > > > -   **PEARL:** 644k
> > > > > -   **SignNet-8:** 662k
> > > > > -   **SignNet-full:** 631k
> > > > > -   **SPE-full:** 650k
> > > > > -   **SPE-8:** 635k
> > > > >
> > > > > Additionally, for completeness, we include a 487k-parameter version of both **R-PEARL** and **B-PEARL** in the results table below. Despite having fewer parameters and lower complexity, **B-PEARL** still outperforms the baselines, demonstrating its efficiency and effectiveness.
> > > > >
> > > > >
> > > > > ### Table: Test/Training MAE and Generalization Gap Results on ZINC
> > > > >
> > > > > | PE Method              | #PEs | #Params (k) | Test MAE                | Training MAE           | Generalization Gap      |
> > > > > |------------------------|------|-------------|-------------------------|------------------------|-------------------------|
> > > > > | No PE                 | N/A  | 575         | $0.1772_{\pm 0.0040}$   | $0.1509_{\pm 0.0086}$  | **$0.0263_{\pm 0.0113}$** |
> > > > > | PEG                   | 8    | 512         | $0.1444_{\pm 0.0076}$   | $0.1088_{\pm 0.0066}$  | $0.0382_{\pm 0.0100}$  |
> > > > > | PEG                   | Full | 512         | $0.1878_{\pm 0.0127}$   | $0.1486_{\pm 0.0191}$  | $0.0342_{\pm 0.0206}$  |
> > > > > | SignNet               | 8    | 631         | $0.1034_{\pm 0.0056}$   | $0.0418_{\pm 0.0101}$  | $0.0602_{\pm 0.0112}$  |
> > > > >  SignNet               | Full | 662         | $0.0853_{\pm 0.0026}$   | $0.0349_{\pm 0.0078}$  | $0.0502_{\pm 0.0103}$  |
> > > > > | BasisNet              | 8    | 442         | $0.1554_{\pm 0.0068}$   | $0.0513_{\pm 0.0053}$  | $0.1042_{\pm 0.0063}$  |
> > > > > | BasisNet              | Full | 513         | $0.1555_{\pm 0.0124}$   | $0.0684_{\pm 0.0202}$  | $0.0989_{\pm 0.0258}$  |
> > > > > | SPE                   | 8    | 635         | $0.0736_{\pm 0.0007}$   | **$0.0324_{\pm 0.0058}$** | $0.0413_{\pm 0.0057}$  |
> > > > > | SPE                   | Full | 650         | $0.0693_{\pm 0.0040}$ | $0.0334_{\pm 0.0054}$  | $0.0359_{\pm 0.0087}$  |
> > > > > | R-PEARL (ours)        | N/A  | **644**         | $0.0699_{\pm 0.002}$      | $0.0366_{\pm 0.006}$      | $0.0333_{\pm 0.007}$      |
> > > > > | B-PEARL (ours)        | N/A  | **644**         | $0.0644_{\pm 0.001}$  | $0.0290_{\pm 0.003}$  | $0.0353_{\pm 0.002}$  |
> > > > > | R-PEARL (ours)-500k   | N/A  | **487**         | $0.0721_{\pm 0.0045}$     | $0.0399_{\pm 0.001}$      | $0.0306_{\pm 0.0015}$      |
> > > > > | B-PEARL  (ours)-500k  | N/A  | **487**         | $0.0679_{\pm 0.0026}$     | $0.0336_{\pm 0.008}$      | $0.0322_{\pm 0.004}$       |
> > > > >
> > > > >
> > > > > >   The authors argued that their method is more scalable. Could you a) test on larger graphs, and b) measure the improved speed?
> > > > >
> > > > > We have included experiments with Rel-stack in the revised manuscript, which is a temporal and heterogeneous graph with approximately 38 million nodes. Please see "Summary of our response (2/2)" for more details. The time required for each method to operate per-epoch is: R-PEARL: 1minute, B-PEARL: 5 minutes, SignNet-8s: 26 minutes,  SignNet-8L: 8 minutes.

---

> > > > ### Comment · Reviewer_J67b · 2024-11-25
> > > >
> > > > Dear Authors,
> > > >
> > > > Thank you for your response. I would like to make a quick remark: Any analytic function $f:\mathbb{R}\to \mathbb{R}$ of individual eigenvalues is inherently a permutation equivariant Lipschitz continuous function $\tilde{f}:\mathbb{R}^n \to \mathbb{R}^n$ over the entire spectrum of eigenvalues by setting $\tilde{f}(x_1, \ldots, x_n)  := ( f(x_1), \ldots, f(x_n) )$. Consequently, this suggests that your formulation is less general than the SPE approach proposed by Huang et al. To see that SPE is strictly more general: The permutation equivariant function $(x_1, x_2) \mapsto (x_1+x_2, x_1+x_2)$ can not be realized by any function acting separately on each entry.
> > > >
> > > > I also agree that achieving basis universality does not require permutation-equivariant sequence-to-sequence mappings. Pointwise functions of eigenvalues are indeed sufficient for this purpose. But at the same time, breaking permutation symmetry by introducing random node features or (not permutation equivariant) unique node identifiers is unnecessary for achieving basis universality. Note that finding permutation equivariant unique node identifiers is NP-hard.
> > > >
> > > > Best regards,

---

> ### Author Response · Authors · 2024-11-24
> **Response to additional comment (1/2)**
>
> > However, it is unclear whether this approach improves or decreases expressivity or leads to better stability or generalization bounds.
>
> Our new expressivity results demonstrate that PEARL is at least as expressive as SPE. Furthermore, PEARL exhibits a lower stability bound that scales sublinearly with graph size, in contrast to the linear scaling of SPE.
>
> This significant difference in stability bounds suggests that PEARL might have better generalization properties compared to SPE, making it a more robust and scalable framework for a variety of graph-based tasks.
>
> > Notably, the out-of-distribution (OOD) generalization in the experiments appears to be better for the SPE positional encodings. However, it is not clear why since SPEs are also stable.
>
> We believe the reviewer is referring to the DrugOOD experiments, which explicitly assess out-of-distribution (OOD) generalization. In the revised manuscript, we have included experiments with B-PEARL that demonstrate a significant advantage of B-PEARL over SPE in terms of size generalization. This improvement is likely due to the linear scaling of SPE with graph size, compared to the square-root scaling of PEARL, which provides better size generalization capabilities. The results can be found below:
>
> ### Table: AUROC Results (5 Random Seeds) on DrugOOD
>
>
> | **Domain**      | **PE Method**       | **ID-Val (AUC)**      | **ID-Test (AUC)**     | **OOD-Val (AUC)**     | **OOD-Test (AUC)**     |
> |------------------|---------------------|-----------------------|-----------------------|-----------------------|------------------------|
> | **Assay**       | No PE               | 92.92 ± 0.14          | 92.89 ± 0.14          | 71.02 ± 0.79          | 71.68 ± 1.10          |
> |                 | PEG                 | 92.51 ± 0.17          | 92.57 ± 0.22          | 70.86 ± 0.44          | 71.98 ± 0.65          |
> |                 | SignNet             | 92.26 ± 0.21          | 92.43 ± 0.27          | 70.16 ± 0.56          | **72.27 ± 0.97**       |
> |                 | BasisNet            | 88.96 ± 1.35          | 89.42 ± 1.18          | 71.19 ± 0.72          | 71.66 ± 0.05          |
> |                 | SPE                 | 92.84 ± 0.20          | 92.94 ± 0.15          | 71.26 ± 0.62          | **72.53 ± 0.66**       |
> |                 | SPE-8S              | 92.36 ± 0.18          | 92.62 ± 0.10          | 70.71 ± 0.47          | 71.72 ± 0.71          |
> |                 | R-PEARL (ours)    | 92.71 ± 0.10          | 92.92 ± 0.12          | 70.57 ± 0.72          | 72.24 ± 0.30          |
> |                 | B-PEARL (ours)    | 90.54 ± 0.89          | 90.81 ± 0.79          | 70.53 ± 0.67          | 71.22 ± 0.42          |
> | **Scaffold**    | No PE               | 96.56 ± 0.10          | 87.95 ± 0.20          | 79.07 ± 0.97          | 68.00 ± 0.60          |
> |                 | PEG                 | 95.65 ± 0.29          | 86.20 ± 0.14          | 79.17 ± 0.97          | 69.15 ± 0.75          |
> |                 | SignNet             | 95.48 ± 0.34          | 86.73 ± 0.56          | 77.81 ± 0.70          | 66.43 ± 1.06          |
> |                 | BasisNet            | 85.80 ± 3.75          | 78.44 ± 2.45          | 73.36 ± 1.44          | 66.32 ± 5.68          |
> |                 | SPE                 | 96.32 ± 0.28          | 88.12 ± 0.41          | 80.03 ± 0.58          | **69.64 ± 0.49**       |
> |                 | SPE-8S              | 96.44 ± 0.08          | 87.88 ± 0.45          | 79.34 ± 0.50          | 68.72 ± 0.63          |
> |                 | R-PEARL (ours)    | 96.09 ± 0.32          | 88.01 ± 0.43          | 78.72 ± 0.02          | 69.20 ± 1.00          |
> |                 | B-PEARL (ours)    | 96.06 ± 0.29          | 87.56 ± 0.81          | 79.86 ± 0.58          | 69.51 ± 0.62          |
> | **Size**        | No PE               | 93.78 ± 0.12          | 93.60 ± 0.27          | 82.76 ± 0.04          | **66.04 ± 0.70**       |
> |                 | PEG                 | 92.46 ± 0.35          | 92.67 ± 0.23          | 82.12 ± 0.49          | 66.01 ± 0.10          |
> |                 | SignNet             | 93.30 ± 0.43          | 93.20 ± 0.39          | 80.67 ± 0.50          | 64.03 ± 0.70          |
> |                 | BasisNet            | 86.04 ± 4.01          | 85.51 ± 4.04          | 75.97 ± 1.71          | 60.79 ± 3.19          |
> |                 | SPE                 | 92.46 ± 0.35          | 92.67 ± 0.23          | 82.12 ± 0.49          | 66.02 ± 0.49          |
> |                 | SPE-8S              | 93.68 ± 0.20          | 93.86 ± 0.12          | 83.04 ± 0.63          | 65.74 ± 2.20          |
> |                 | R-PEARL (ours)    | 93.32 ± 0.34          | 93.92 ± 0.20          | 82.09 ± 0.44          | 65.89 ± 1.30          |
> |                 | B-PEARL (ours)    | 93.18 ± 0.45          | 93.29 ± 0.46          | 83.14 ± 0.37          | **66.58 ± 0.67**       |

---

> > ### Author Response · Authors · 2024-11-24
> > **Response to additional comment (2/2)**
> >
> > > Finally, I want to direct the authors to "Spatio-Spectral Graph Neural Networks" by Geisler et al., which also uses spectral and spatial layers in GNNs.
> >
> > We thank the reviewer for their suggestion and we have added the aforementioned paper in the relevant work discussion. Note however that our work is fundamentally different since we are not directly using any spectral operation as the suggested work.

---

> ### Comment · Reviewer_J67b · 2024-11-25
> **Reply**
>
> Dear Authors,
>
> Thank you for addressing my questions. The experimental results presented in your manuscript are promising, and I appreciate that your approach is distinct from prior work, offering advantages such as scalability and potentially greater stability in terms of generalization. By introducing less complexity compared to methods like SPE, your method shows practical utility.
>
> However, there are significant issues with the theoretical formulations, proofs, and the overall scope of the work. First, the claim that your method is permutation equivariant because it takes the empirical mean of random variables (all with the same expectation) is incorrect. Augmenting graphs with random node features is a known technique that inherently breaks permutation equivariance but achieves universality. This phenomenon is not surprising and is analogous to applying an MLP (or any universal architecture) to the flattened adjacency matrix. While this achieves universality, allowing the method to separate all graphs, it sacrifices permutation equivariance in the process.
>
> Section 3 and Proposition 3.1 also present significant issues. Many notations are undefined, making the section difficult to understand. For example, in Proposition 3.1, you use $\mathrm{MLP}$ in the formulation of the result, but the proof indicates that $\mathrm{MLP}$ corresponds to the update function $g^{(l)}$ of a layer. This interpretation is problematic due to dimensional inconsistencies. According to your manuscript, $\mathrm{MLP}$ operates on each node independently to maintain permutation equivariance, mapping from $\mathbb{R}^{F_l}$ to $\mathbb{R}^{F_{l+1}}$. Applying $\mathrm{MLP}$ to $\mathbf{V} \in \mathbb{R}^{n \times n}$ is therefore invalid. The proof, as written, does not appear correct.
>
> Similar issues arise in Theorem 3.1. For example, Equation 29 claims that $X^{(K)}[:, :, f] \in \mathbb{R}^{n \times n}$, which is clearly incorrect. These errors undermine the credibility of the theoretical results and need to be addressed comprehensively.
>
> Beyond these specific issues, the manuscript overall suffers from a lack of rigor in its use of notation and definitions, which makes the theoretical results difficult to understand. Every notation must be clearly defined, and all results and proofs need thorough verification to ensure correctness and clarity.
>
> While the experimental results and scalability of your approach are promising, the theoretical contributions are incremental and currently marred by significant issues in rigor and correctness. This is not inherently problematic for publication at ICLR, provided these concerns are addressed. However, I strongly recommend another round of revisions where the authors carefully verify all theoretical results and proofs and improve the overall clarity of the manuscript.
>
> For these reasons, I maintain my recommendation for rejection at this stage, with the hope that the authors will submit an improved and rigorously verified version in the future.

---

> ### Author Response · Authors · 2024-11-26
> **Addressing reviewer's remaining concerns (1/4)**
>
> We thank the reviewer for their prompt response and the opportunity to engage in a constructive and meaningful discussion. We greatly appreciate the reviewer’s recognition of the **experimental results** and the **novelty** of our approach, as well as their acknowledgment of the **scalability, stability, and practical utility** of our method.
>
> The reviewer raised three primary concerns regarding our work:
>
> 1.  *The permutation equivariance of our approach.*
> 2.  *The ability to achieve both universality and equivariance simultaneously.*
> 3.  *The clarity of Theorem 3.1 and Proposition 3.2.*
>
> We provide a brief summary of our responses below, followed by more detailed explanations.
>
> ----------
>
> ### **Summary of Responses (TL;DR)**
>
> **A1:** The general PEARL framework is introduced in Section 3.2, and we propose two practical implementations:
>
> -   **R-PEARL** (Section 4), which uses $M$ random samples to initialize node attributes.
> -   **B-PEARL** (Section 5), which uses the $N$ standard basis vectors to initialize node attributes.
>
> B-PEARL is deterministically **equivariant** and **universal**. R-PEARL, as an empirical expectation, can approximate the true expectation with arbitrary accuracy. Theorem 4.3 provides a bound on the number of samples required as a function of accuracy. While the expectation itself is equivariant, R-PEARL is **approximately equivariant** with increasing accuracy as the number of samples grows. For a very small number of samples, R-PEARL is not equivariant; however, in practice, sufficient samples (20-200 for any graph size) render it effectively equivariant.
>
> **A2:** The proof of the **universality** theorem does not rely on randomness or random samples. Instead, the initial node encodings use standard basis vectors. Thus, **universality is achieved deterministically, while preserving equivariance**.
>
> **A3:** We have revised the manuscript to enhance the clarity and presentation of **Proposition 3.2** and the proof of **Theorem 3.1**.
>
> ----------
>
> We look forward to further feedback from the reviewer and hope our clarifications address their concerns.

---

> ### Author Response · Authors · 2024-11-26
> **Addressing reviewer's remaining concerns (2/4)**
>
> > First, the claim that your method is permutation equivariant because it takes the empirical mean of random variables (all with the same expectation) is incorrect.
>
> In PEARL, we draw $M$ samples of a random vector $\mathbf{q}$ to initialize the node attributes. Since this random vector is processed using an equivariant GNN $\Phi(\mathcal{G}, \cdot)$, the resulting distribution of $\Phi(\mathcal{G}, \mathbf{q})$ is permutation equivariant. Consequently, the expected value $\mathbb{E}\left[\Phi(\mathcal{G}, \mathbf{q})\right]$ is also permutation equivariant.
>
> According to the law of large numbers and concentration inequalities, the empirical mean approximates the true mean with arbitrary accuracy, where the level of accuracy depends on the number of samples. R-PEARL computes the empirical mean $\hat{\mathbb{E}}\left[\Phi(\mathcal{G}, \mathbf{q})\right]$, enabling it to approximate $\mathbb{E}\left[\Phi(\mathcal{G}, \mathbf{q})\right]$ with high precision. In essence, R-PEARL achieves equivariance with increasing accuracy as the number of samples $M$ grows.
>
> The equivariance of R-PEARL depends on the number of samples $M$, with Theorem 4.3 providing a bound on the number required to achieve a specified accuracy. For very small sample sizes, R-PEARL is not equivariant; however, in practice, a modest number of samples (50–200) is sufficient to render it effectively equivariant. This is evident in our experiments. In Tables 1 and 2, the "GIN + rand id" model corresponds to R-PEARL with a single sample, which is not equivariant. We observe that "GIN + rand id" performs significantly worse—more than four times worse in logP prediction for molecular graphs—compared to R-PEARL.
>
> **We have added this discussion in the revised manuscript**
>
> To further validate this empirically, we measured the effect of randomness on the output of R-PEARL for graphs in the ZINC dataset. Specifically, we conducted five independent runs of R-PEARL, each with 100 samples and different random seeds. To evaluate consistency, we calculated the relative difference between the PEs generated for each node across these runs, defined as
>
> $$d_{ij}=\frac{|\mathbf{P}[\text{run}_i]-\mathbf{P}[\text{run}_j]|}{|\mathbf{P}[\text{run}_i]|}$$.
>
> Our findings revealed that $d_{ij}$ was consistently on the order of $10^{-2}$, while the difference in PEs between different nodes within the same run was on the order of $10^{-1}$.
>
> These results demonstrate that the variability introduced by randomness is negligible compared to the inherent differences in node features, supporting our claim that R-PEARL is practically equivariant in real-world applications.
>
> >Augmenting graphs with random node features is a known technique that inherently breaks permutation equivariance but achieves universality. This phenomenon is not surprising and is analogous to applying an MLP (or any universal architecture) to the flattened adjacency matrix. While this achieves universality, allowing the method to separate all graphs, it sacrifices permutation equivariance in the process.
>
> We completely agree with the reviewer. However, this is not how PEARL operates or how we establish universality in our work. The proof of the universality theorem does not involve randomness or rely on random samples. Instead, the initial node encodings are constructed using standard basis vectors. This approach ensures that universality is achieved deterministically while maintaining equivariance.

---

> ### Author Response · Authors · 2024-11-26
> **Addressing reviewer's remaining concerns (3/4)**
>
> >Section 3 and Proposition 3.1 also present significant issues. Many notations are undefined, making the section difficult to understand. For example, in Proposition 3.1, you use  MLP  in the formulation of the result, but the proof indicates that  MLP  corresponds to the update function  g(l)  of a layer. This interpretation is problematic due to dimensional inconsistencies. According to your manuscript,  MLP  operates on each node independently to maintain permutation equivariance, mapping from $\mathbb{R}^{F_l}$ to $\mathbb{R}^{F_{l+1}}$. Applying  MLP  to  $\mathbf{V}\in\mathbb{R}^{n×n}$  is therefore invalid. The proof, as written, does not appear correct.
>
> This is an interesting point raised by the reviewer. A GNN layer is a function $g\circ f : \left(\mathcal{G},\mathbb{R}^{F_{l-1}}\right)\rightarrow \mathbb{R}^{F_l}$, where $\mathcal{G}$ represents a graph with $N$ nodes ($N$ can be arbitrary). Let $f$ be one of the functions in Eq. (2) and $g$ be a multi-layer perceptron. According to Eq. (3) the degrees of freedom in the first perceptron layer are $K F_l F_{l-1}$.
>
> According to Proposition 3.1, the update for node $v$, is still a mapping $g\circ f : \left(\mathcal{G},\mathbb{R}^{F_{l-1}}\right)\rightarrow \mathbb{R}^{F_l}$ and $g\circ f$ can be **interpreted** as a nonlinear function of eigenvectors
>
> $$\mathbf{x}_v^{(l)} =\texttt{MLP}\left(\mathbf{v}^{(v)}\right)=\texttt{MLP}^{(-1)}\left(\sigma\left(\mathbf{W}^T\mathbf{v}^{(v)}\right)\right),~\mathbf{v}^{(v)}=\mathbf{V}[v,:]^T$$
>
> The degrees of freedom in the first MLP layer are $K F_l F_{l-1}$, since $\mathbf{W}[n,f]$ are not independent but they are themselves functions of node attributes and graph, i.e., $\mathbf{W}[n,f] : \left(\mathcal{G},\mathbb{R}^{F_{l-1}}\right)\rightarrow \mathbb{R}^{1}$. So the claim of Proposition 3.1 is not that a GNN layer just takes $\mathbf{v}^{(v)}$ as an input and applies a $\mathbb{R}^{N}\rightarrow \mathbb{R}^{F_l}$ mapping to it. Instead the claim is that the $\left(\mathcal{G},\mathbb{R}^{F_{l-1}}\right)\rightarrow \mathbb{R}^{F_l}$ GNN layer mapping can be viewed as a $\left(\mathbf{V},\mathbf{\Lambda},\mathbb{R}^{F_{l-1}}\right)\rightarrow \mathbb{R}^{F_l}$ mapping and the whole operation can be interpreted as a nonlinear mapping of the $v-$th row of the eigenvector matrix $\mathbf{V}[v,:]^T$, but the weights of this mapping are dependent with $K F_l F_{l-1}$ degrees of freedom.
>
> We updated Proposition 3.1 and added additional discussion that now read:
>
>
> ### Proposition: GNNs as Nonlinear Functions of Eigenvectors
> A GNN defined in Eq. (1) with  $f^{(l)}$ being one of the functions in Eq. (2) and $g^{(l)}$ being a multi-layer perceptron, operates as a nonlinear function of the GSO eigenvectors, i.e.,  $\mathbf{x}_v^{(l)} = \texttt{MLP}\left(\mathbf{v}^{(v)}\right),~\mathbf{v}^{(v)}=\mathbf{V}[v,:]^T$.
>
> The trainable parameters of the first MLP layer are not independent but depend on the eigenvalues $\{\lambda_n\}_{n=1}^N$, eigenvectors $\mathbf{v}_n$ $n=1,\dots,N$ of the GSO, and the node features $\mathbf{X}$ of the graph:
>
> $$\mathbf{x}_v^{(l)} = \texttt{MLP}\left(\mathbf{v}^{(v)}\right) = \texttt{MLP}^{(-1)}\left(\sigma\left(\mathbf{W}^T\mathbf{v}^{(v)}\right)\right)$$
>
>
> $$\mathbf{W}[n, f] = \sum_{i=1}^{F_{l-1}}\sum_{k=0}^{K-1}\lambda_n^k\mathbf{H}_k^{(l)}[i,f]\langle \mathbf{\alpha}_n, \mathbf{X}^{(l-1)}[:,i]\rangle,$$
>
> where $\mathbf{\alpha}_n = \mathbf{v}_n$ when the GSO is symmetric, and $\mathbf{\alpha}_n = \mathbf{V}^{-1}[n,:]$ when it is not. $MLP^{(-1)}$ denotes all the layers of the $\texttt{MLP}$ except the first layer.
>
> ---
> The proof is provided in Appendix A. According to Proposition 1, the update for node $v$, which is a function $g \circ f : \mathbb{R}^{F_{l-1}} \rightarrow \mathbb{R}^{F_l}$, can be viewed as a function $\texttt{MLP} : \mathbb{R}^N \rightarrow \mathbb{R}^{F_l}$ operating on $\mathbf{V}[v,:]$. The degrees of freedom in the first layer of $\texttt{MLP}$ are $K F_l F_{l-1}$ (as described in Eq. (3)), and not $F_l N$, which would have been the case for independent weights $\mathbf{W}$.
>
>
> The proof is provided in Appendix B. \blue{According to Proposition 3.1, the update for node $v$, defined by the function $g \circ f : \left(\mathcal{G},\mathbb{R}^{F_{l-1}}\right) \rightarrow \mathbb{R}^{F_l}$, can be interpreted as a nonlinear mapping $\texttt{MLP}$ applied to $\mathbf{V}[v, :]$, but the weights of the first layer of this mapping are also mappings, i.e., $\mathbf{W}[n,f] : \left(\mathcal{G},\mathbb{R}^{F_{l-1}}\right)\rightarrow \mathbb{R}^{1}$. The degrees of freedom in the first layer of $\texttt{MLP}$ are $K F_l F_{l-1}$ (as described in Eq. (5), rather than $F_l N$, which would be the case for independent weights $\mathbf{W}$. Furthermore, the dot product $\langle \mathbf{\alpha}_n, \mathbf{X}^{(l-1)}[:, i] \rangle$ depends on the eigenvectors and, for the update of node $v$, it only involves the components $\mathbf{X}^{(l-1)}[u, i],~u\in\mathcal{N}_v$.

---

> ### Author Response · Authors · 2024-11-26
> **Addressing reviewer's remaining concerns (4/4)**
>
> > Similar issues arise in Theorem 3.1. For example, Equation 29 claims that  $X^K [:,:,f]∈R^{N×N}$, which is clearly incorrect. These errors undermine the credibility of the theoretical results and need to be addressed comprehensively.
>
> We thank the reviewer for their comment. To clarify, there is no inaccuracy here, only a slight abuse of notation. We have updated $X^K [:,:,f]$ to $X^K [:,f,:]$ and have provided additional details for clarity. The text now reads:
>
> Since we independently feed $\mathbf{e}_1,\dots, \mathbf{e}_N$ to the PEARL architecture, we will have $N$ output samples for each output feature, i.e., $N$ samples for $\mathbf{X}^{(K)}[:,f]$. We will represent the $m-$th sample as $\mathbf{X}^{(K)}[:,f,m]$ and for the $f-$th output feature will have the following samples:
>
> $$\mathbf{X}^{(K)}[:,f,:]=[\mathbf{V_{\mu_f}}\mathbf{V_{\mu_f}}^T\mathbf{e}^1,\dots, \mathbf{V_{\mu_f}}\mathbf{V_{\mu_f}}^T\mathbf{e}^N]=\mathbf{V_{\mu_f}}\mathbf{V_{\mu_f}}^T$$
>
>
> >Beyond these specific issues, the manuscript overall suffers from a lack of rigor in its use of notation and definitions, which makes the theoretical results difficult to understand. Every notation must be clearly defined, and all results and proofs need thorough verification to ensure correctness and clarity.
>
> We appreciate the reviewer’s feedback and would greatly value more specific suggestions or detailed comments to help us improve the quality of our paper.

---

> ### Author Response · Authors · 2024-11-26
> **Following up on reviewer's quick remark**
>
> > I would like to make a quick remark: Any analytic function  f:R→R  of individual eigenvalues is inherently a permutation equivariant Lipschitz continuous function  $f~:R^n→R^n$  over the entire spectrum of eigenvalues by setting  $f(x1,…,xn):=(f(x1),…,f(xn))$. Consequently, this suggests that your formulation is less general than the SPE approach proposed by Huang et al. To see that SPE is strictly more general: The permutation equivariant function  $(x1,x2)\rightarrow(x1+x2,x1+x2)$  can not be realized by any function acting separately on each entry.
>
> We agree with the reviewer that an analytic function  $a^{(1)}:R→R$  of individual eigenvalues is less general than a function $a^{(2)}:R^n→R^n$ jointly operating on the whole spectrum. However, this does nor necessarily mean that
>
> $f_1= \rho_1\circ a^{(1)}=\rho_1\left(\left[\mathbf{V}\text{diag}\left(\alpha^{(1)}_1\left(\mathbf{\Lambda}\right)\right)\mathbf{V}^T,\dots,\mathbf{V}\text{diag}\left(\alpha^{(1)}_F\left(\mathbf{\Lambda}\right)\right)\mathbf{V}^T\right]\right),$
>
> is less general than
>
> $f_2= \rho_2\circ a^{(2)}=\rho_1\left(\left[\mathbf{V}\text{diag}\left(\alpha_1^{(2)}\left(\mathbf{\Lambda}\right)\right)\mathbf{V}^T,\dots,\mathbf{V}\text{diag}\left(\alpha_F^{(2)}\left(\mathbf{\Lambda}\right)\right)\mathbf{V}^T\right]\right)$.
>
> The interesting question is how general are $f_1= \rho_1\circ a^{(1)}$ and $f_2= \rho_2\circ a^{(2)}$. $f_1$ and $f_2$ are functions operating on a basis $\mathbf{V}$ and for $f_1$ and $f_2$ to be universal,  what matters is how distinctive $\alpha^{(1)}$ or $\alpha^{(2)}$ are, not how expressive. Having said that we can also view $f_1$ and $f_2$ as invariant functions at the graph level and in that case we would be interested in testing whether $f_1$ and $f_2$ are univarsal functions of eigenvalues. To that end we prove that PEARL or SPE with analytical functions of individual eigenvalues $a_i^{(1)}:R→R$, are universal functions of eigenvalues.
>
> The proof can be found in Appendix D of the revised manuscript.

---

> > ### Author Response · Authors · 2024-12-02
> > **We would like to hear your thoughts on our response and revised manuscript**
> >
> > Dear Reviewer J67b,
> >
> > Thank you for your thoughtful comments and constructive feedback. Your insights have significantly contributed to improving our paper.
> >
> > In our revised manuscript, we have enhanced the clarity and presentation of Proposition 3.2 and Theorem 3.1. Additionally, we have carefully addressed your concerns regarding the permutation equivariance of our approach and the ability to achieve both universality and equivariance simultaneously.
> >
> > We would greatly appreciate it if you could share your thoughts on our response and the updated manuscript. Please let us know if you have any further questions or feedback—we would be happy to address them.
> >
> > Thank you once again for your time and effort in reviewing our work.

---

> > > ### Comment · Reviewer_J67b · 2024-12-02
> > >
> > > Dear Authors,
> > >
> > > Thank you for your response. However, I must reiterate that **your proposed method is not permutation equivariant and universal simultaneously**. As you mentioned in your reply, you employ either random features or "basis vectors" to augment node features. Both approaches inherently break permutation equivariance.
> > >
> > > There is no polynomial-time algorithm known that can augment node features with basis vectors or any other unique node identifiers. Specifically, concatenating basis vectors to an arbitrary graph $G$ within its equivalence class (of isomorphic graphs) is already not permutation equivariant. For further details, please refer to the discussion on graph canonization: https://en.wikipedia.org/wiki/Graph_canonization.
> > >
> > > Thank you for clarifying the other points.
> > >
> > > Best regards,

---

> ### Author Response · Authors · 2024-12-02
> **Key misunderstanding in reviewer's comment. Please see our response.**
>
> Dear Reviewer J67b,
>
> Thank you for acknowledging that we have addressed your other concerns. Regarding your final concern, we would like to point out a **key misunderstanding on your comment that seems to be causing confusion about our work.**
>
> **A function can have non permutation equivariant inputs, yet produce a permutation equivariant output.** For instance, SPE (Huang et al.), SignNet (Lim et al.), and BasisNet (Lim et al.) take as inputs the Laplacian eigenvectors, which are not permutation equivariant due to sign and basis ambiguities, but transform them into permutation-equivariant node features. Our approach follows a similar principle.
>
> In the case of B-PEARL, the inputs are standard basis vectors, and **the PEARL mechanism subsequently transforms them into equivariant node features.** This is distinct from merely augmenting the output with standard basis vectors, which would not be permutation-equivariant. In simple words, the average of standard basis vectors is permutation-equivariant, as is the average of GNN outputs when the nodes are initialized with standard basis vectors.
>
> Furthermore, our proposed **PEARL**—like SPE and BasisNet—is not a universal approximator of arbitrary graph functions. Rather, it is **a universal approximator for any functions that satisfy $f(\mathbf{V})=f(\mathbf{VQ})$**, i.e., basis-equivariant or invariant functions, as explained in **Theorem 3.1** and reiterated in our previous responses. **Please see Appendix C in the revised manuscript that proves how PEAL jointly achieves basis universality and equivariance.**
>
> We hope this clarification resolves your last concern and look forward to your response.

---

### Author Response · Authors · 2024-11-25
**Updated manuscript**

We would like to sincerely thank all reviewers for their thoughtful feedback and for appreciating the contributions of our work. We have carefully addressed all the reviewers' concerns and clarified misconceptions about our approach. Each response includes a summary to refresh the key aspects of our work, followed by detailed replies to each comment. We would also greatly appreciate the opportunity to engage further in a constructive scientific discussion.

### Summary of Revisions:

We have made several updates to the manuscript, with the most significant improvements outlined below:

-   **Universal Approximation Result**: We prove that PEARL is a universal approximator of basis-invariant functions. This new result provides substantial insights into the expressivity of the proposed PE framework. In summary, we establish that PEARL is a **generic** positional encoding framework that is **jointly expressive, stable, generalizable, and scalable**. To the best of our knowledge, this is the first PE method to satisfy all these properties simultaneously.

-   **Additional Experiments**: We have conducted experiments on three additional tasks—**Rel-stack user-post-comment prediction**, **Rel-stack post-post-related prediction**, and **CSL**:

    -   The **Rel-stack tasks** involve large-scale link prediction on the Stack Exchange Q&A Website Database, modeled as a temporal-heterogeneous graph with approximately 38 million nodes. PEARL achieves state-of-the-art performance, demonstrating its effectiveness in real-world applications and its scalability to very large graphs.
    -   On the **CSL benchmark**, which tests graph isomorphism capabilities, PEARL achieves **100% classification accuracy**.

Overall, **PEARL has now been evaluated across 10 different tasks spanning 5 distinct graph types**:

1.  **Molecular Graphs**:
    -   LogP prediction on ZINC
    -   Binding affinity prediction on DrugOOD (3 tasks)
2. **Long-range Molecular Graphs**
    - Multi-label graph regression based on the 3D structure of the peptides
3.  **Social Networks**:
    -   Subreddit classification on REDDIT-BINARY
    -   Subreddit classification on REDDIT-MULTI5K
4.  **Relational Databases**:
    -   Post-post-related recommendation on Stack Exchange
    -   User-post-comment recommendation on Stack Exchange
5.  **Synthetic Graphs**:
    -   Graph classification on CSL

Across all tasks, the results consistently demonstrate that PEARL outperforms baseline methods and, in a few instances, just achieves performance on par with the best-performing approaches. Notably, in many instances, PEARL achieves these results with significantly lower computational and memory complexity, further showcasing its efficiency and practicality. We also include several additional ablation studies and runtime comparisons in the revised manuscript.

---

> ### Author Response · Authors · 2024-11-25
> **Main new results**
>
> ## Basis Universality of PEARL
>
> **To further strengthen the theoretical analysis of our approach we prove, in the revised manuscript,  that PEARL is a universal approximator of basis invariant functions. The details are presented below.**
>
> **Theorem [Basis Universality]**
> *Let* $\mathcal{G}$ *be a graph with GSO* $\mathbf{S} = \mathbf{V} \mathbf{\Lambda} \mathbf{V}^T$, *and* $f$ *be a continuous function such that* $f(\mathbf{V}) = f(\mathbf{V} \mathbf{Q})$, $\mathbf{Q} \in \mathcal{O}\left({diag}\left(\mathbf{\Lambda}\right)\right)$, *for any eigenvalues* $\mathbf{\Lambda}$. *Then there exist GNN* $\Phi$ *and a continuous pooling function* $\rho$, *such that*:
>
> $$
> f(\mathbf{V}) = \rho\left[ \Phi\left( \mathcal{G}, \mathbf{q}^{(1)} \right), \dots, \Phi\left( \mathcal{G}, \mathbf{q}^{(M)} \right) \right].
> $$
>
>
>
> ## Large-Scale Prediction on RelBench
>
>
> To further empasize the scalability and effectiveness of our approach we test the performance of the proposed PEARL on large-scale link prediction for Stack Exchange Q\&A Website Database. To that end we utilize the rel-stack dataset for the relational deep learning benchmak (RelBench) [2]. Rel-stack is a temporal and heterogeneous graph with approximately 38 million nodes. We consider two different tasks; i) user-post-comment, where we predict a list of existing posts that a user will comment in the next two years, and ii) post-post-related, where we predict a list of existing posts that users will link a given post to in the next two years. The results for the two tasks can be found in the following table.
>
> ### Table: Validation and Test Mean Average Precision (MAP) on Large-Scale RelBench Recommendation Tasks
>
> **PEARL** achieves an 11% improvement over the backbone model with no PE on the *user-post-comment* task and a 2% improvement on the *post-post-related** task.
>
> | **Task**                  | **Evaluation** | **No PE** | **SignNet-8L** | **SignNet-8S** | **B-PEARL (ours)** | **R-PEARL (ours)** |
> |---------------------------|----------------|-----------|----------------|----------------|----------------------|----------------------|
> | **User-post-comment**     | Val. MAP       | 15.20     | 15.33          | **15.47**      | 15.13               | 15.24               |
> |                           | Test MAP       | 12.47     | 13.76          | 13.77          | 13.80               | **13.87**           |
> | **Post-post-related**     | Val. MAP       | 8.10      | 7.90           | 7.70           | 8.00                | **8.40**            |
> |                           | Test MAP       | 10.73     | 10.39          | 10.86          | **10.94**           | 10.86               |
>
>
> The backbone model for this RelBench task a heterogeneous identity-aware GNN and all methods are trained with batch size $20$. From the above table we observe that PEARL has an 11\% benefit over the identity aware backbone model with no PE on the user-post-comment task and a 2\% benefit on the post-post-related task. PEARL works similarly to SignNet-8S but with lower complexity and, and similarly to SignNet-8L on the user-post-comment task, but 5\% better than SignNet-8L on the post-post-related task. SPE could not be included in the comparisons due to it's cubic computational complexity and quadratic memory complexity.
>
> [2] Robinson, J., Ranjan, R., Hu, W., Huang, K., Han, J., Dobles, A., Fey, M., Lenssen, J.E., Yuan, Y., Zhang, Z. and He, X., RelBench: A Benchmark for Deep Learning on Relational Databases. In  The Thirty-eight Conference on Neural Information Processing Systems Datasets and Benchmarks Track.

---

### Meta-Review · Area_Chair_d1pK · 2024-12-22

**Metareview:**

Summary: This paper proposes a new way to incorporate positional encodings (or node-wise features) via pooling function. Two types of encodings, i.e., eigenvectors and random features, are considered. The authors also prove their method is universal assuming proper parameterization of the pooling function and neural networks.

Strength: This paper proposes a simple approach that consistently improves the performance of GNNs (at the cost of additional comparison). The performance improvement is consistent across the considered benchmarks.

Weakness: Theorem 3.1 lacks strong connection to rest of the ideas in the work. In the end, the proposed architecture is ensemble of GNNs over randomized features, similar to Equivariant Subgraph Aggregation Networks. An empirical comparison to neural network ensembles with similar inference complexity (not parameter complexity) would have been nice.

Overall, I recommend acceptance for this work.

**Additional Comments On Reviewer Discussion:**

The negative reviewer raised a concern on correctness of theorem 3.1. I have gone through the rebuttals and the appendix of the paper. To the best of my knowledge, the proof is correct. Other concerns also seem to have been alleviated by the author rebuttals.

---

### Decision · Program_Chairs · 2025-01-22

Accept (Poster)